# STI1 domain engages transient helices to mediate Dsk2 phase separation and proteasome condensation

Nirbhik Acharya [ID] [1,2], Emily A Daniel [ID] [3], Thuy P Dao[1,2], Jessica K Niblo [ID] [2], Erin O Mulvey[4], Shahar Sukenik [ID] [2,5], Daniel A Kraut [ID] [4], Jeroen Roelofs [ID] [3] & Carlos A Castañeda [ID] [1,2,5,6✉]

## Abstract

**Ubiquitin-binding shuttle proteins are important components of stress-induced biomolecular condensates in cells. Yeast Dsk2 scaffolds proteasome-containing condensates via multivalent interactions with proteasomes and polyubiquitinated substrates under stress conditions. Here, we identify the chaperone-binding STI1 domain as the main driver of Dsk2 self-association and phase separation. Using nuclear magnetic resonance (NMR) spectroscopy and computational simulations, we find that the STI1 domain interacts with three transient amphipathic helices within the intrinsically disordered regions of Dsk2. Removal of either the STI1 domain or these helices significantly reduces Dsk2's propensity to form condensates. In vivo, perturbing STI1-helix interactions, specifically removal of the transient helices, reduces the formation of azide stress-induced Dsk2/proteasome condensates, in line with our in vitro results. Modeling of Dsk2 STI1-helix interactions reveals a binding mode reminiscent of chaperone STI1/DP2 domains interacting with client helices. Our findings support a model whereby STI1-helix interactions important for Dsk2 condensate formation can be replaced by STI1-client interactions for downstream chaperone or other protein quality control outcomes.**

**Keywords** Transient Helices; STI1 Domain; Phase Separation; Proteasome Condensates; Ubiquilins
**Subject Categories** Post-translational Modifications & Proteolysis; Structural Biology; Translation & Protein Quality

## Introduction

Ubiquitin (Ub)-binding shuttle proteins and adapters are important drivers and components of biomolecular condensates for protein quality control, stress response, and immune system activation (Yasuda et al, 2020; Uriarte et al, 2021; Waite et al, 2024; Zaffagnini et al, 2018; Sun et al, 2018; Rajendran and Castañeda, 2025; Goel et al, 2023; Du et al, 2022). Ubiquilins (UBQLNs), a subset of these Ub-binding shuttle proteins, form condensates with polyubiquitinated substrates (polyUb-substrates), proteasomes, and/or other PQC machinery under either physiological or stress-induced conditions (Dao et al, 2018; Alexander et al, 2018; Waite et al, 2024; Gerson et al, 2021; Mohan et al, 2022, 2025). The formation of distinct stress-induced proteasome-containing puncta, including cytoplasmic proteasome storage granules in yeast and nuclear proteasome condensates in mammalian cells, may be essential for proteostasis and resistance to stress (Yasuda et al, 2020; Uriarte et al, 2021; Laporte et al, 2008; Gu et al, 2017). In humans, dysregulation of UBQLNs is linked to neurodegeneration (Deng et al, 2011; Lin et al, 2022; Edens et al, 2017). In other organisms, including yeast and plants, removal of UBQLN orthologs negatively affects organismal stress responses (Waite et al, 2024; Nolan et al, 2017; Chuang et al, 2016; Jantrapirom et al, 2020). In yeast, Dsk2, the ortholog of human UBQLNs, and Rad23 are the two main shuttle factors of the ubiquitin–proteasome system (UPS) (Samant et al, 2018; Tsuchiya et al, 2017). Dsk2 and Rad23 are critical drivers of proteasome condensates in stressed yeast cells, and deletion of the genes encoding both proteins abrogates the formation of proteasome-containing condensates in yeast (Waite et al, 2024). However, the molecular underpinnings of this process are unknown.

Yeast Dsk2 (373 amino acids) and human UBQLNs share a similar domain architecture (Appendix Fig. S1). Each contains UBL (Ub-like) and UBA (Ub-associating) domains that interact with proteasome receptors and ubiquitinated substrates, respectively (Fig. 1A) (Funakoshi et al, 2002). Between the folded UBL and UBA domains are predicted intrinsically disordered regions (IDRs) and a single STI1 domain (DSK2) or two STI1 domains (UBQLNs). STI1 domains engage with chaperones and mediate interactions with client proteins, such as transmembrane domains of mitochondrial proteins (Itakura et al, 2016; Schmid et al, 2012; Lin et al, 2021; preprint: Onwunma et al, 2024). However, there is limited structural and dynamical information on STI1 domains and their

[1]Department of Biology, Syracuse University, Syracuse, NY 13244, USA. [2]Department of Chemistry, Syracuse University, Syracuse, NY 13244, USA. [3]Department of Biochemistry and Molecular Biology, University of Kansas Medical Center, Kansas City, KS 66160, USA. [4]Department of Chemistry and Biochemistry, Villanova University, Villanova, PA 19085, USA. [5]BioInspired Institute, Syracuse University, Syracuse, NY 13244, USA. [6]Interdisciplinary Neuroscience Program, Syracuse University, Syracuse, NY 13244, USA. ✉E-mail: cacastan@syr.edu

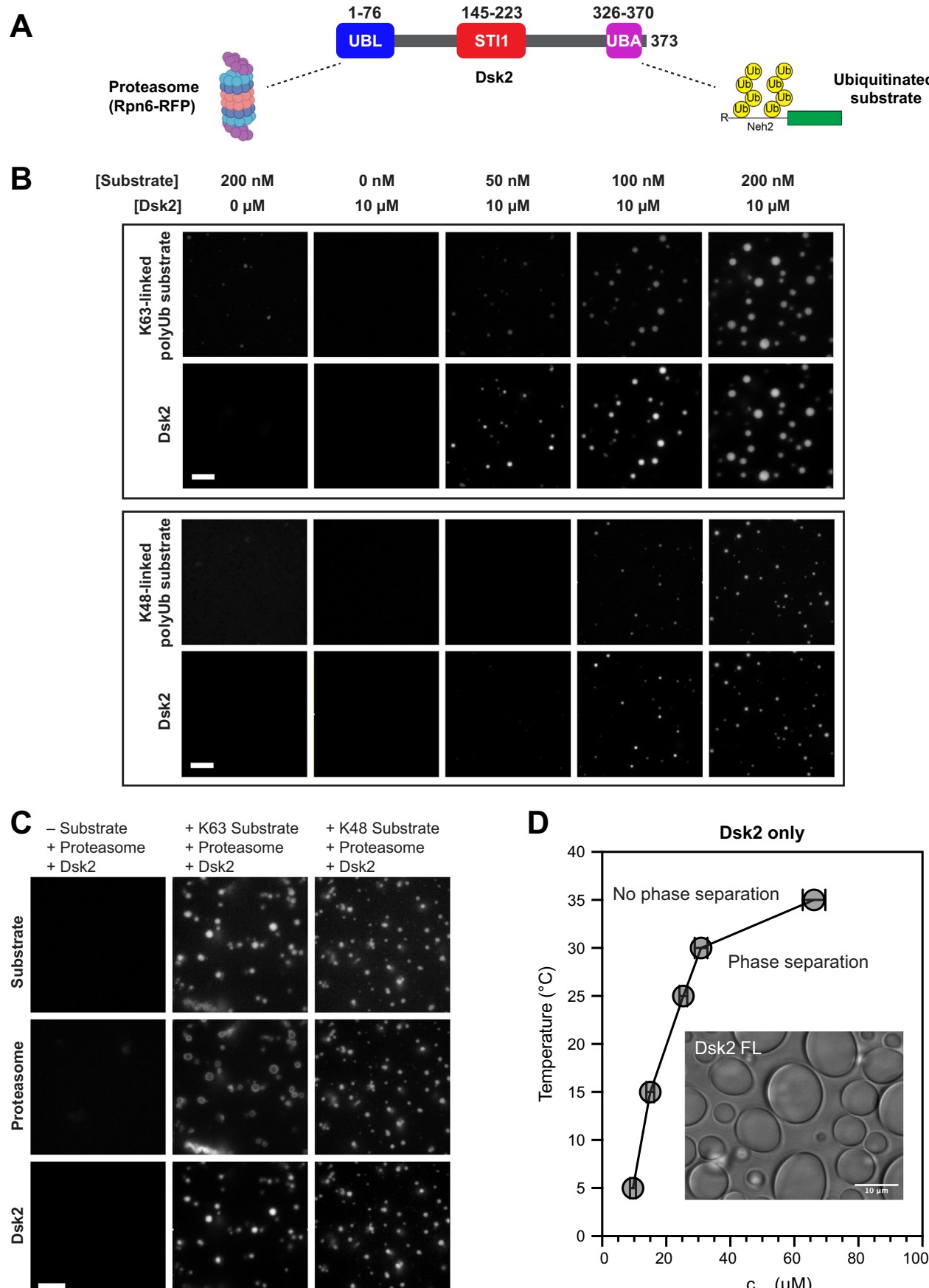

◀ **Figure 1. Dsk2 forms condensates with proteasomes and polyubiquitinated substrates.**

(A) Domain architecture of Dsk2, highlighting known interactions with proteasomes and polyUb-substrates. (B) Fluorescence microscopy showing phase separation of K63-linked or K48-linked R-Neh2Dual-sGFP polyUb-substrates (see "Methods") with Dsk2 (spiked with Alexa Fluor 647-Dsk2) in a concentration-dependent manner (3% PEG 8000, 18 °C, 0 or 10 μM Dsk2, 0–200 nM substrate). Scale bar, 5 μm. (C) Fluorescence microscopy of 10 μM Dsk2, 200 nM K63-linked or K48-linked polyUb-substrate, and 100 nM TagRFP-T-Rpn6 proteasome incubated at 18 °C in pH 7.5 buffer with 3% PEG 8000. Scale bar, 5 μm. (D) Temperature-concentration phase diagram of Dsk2 (alone) delineating the saturation concentration ($c_{sat}$) at different temperatures in buffer containing 20 mM NaPhosphate pH 6.8, 150 mM NaCl, 7.5% PEG 8000, 0.5 mM EDTA. Inset: bright-field microscopy of 100 μM Dsk2 droplets incubated at 25 °C under the same conditions. Scale bar, 10 μm. Values are the mean ± standard deviation (SD) over two or more protein preps in triplicate.

binding mechanisms (Fry et al, 2021). AlphaFold predictions of STI1 domains are generally of low confidence and show non-compact structures. Recent work from our lab demonstrated that the STI1-II domain of UBQLN2 is important for mediating both self-association and phase separation (Dao et al, 2018, 2024). Several ALS/FTD disease-linked mutations in UBQLN2 localize to the STI1 domains, but little is known about their underlying disease mechanisms (Renaud et al, 2019).

Here, we demonstrated that the STI1 domain is a major contributor to the self-association and phase separation of Dsk2 in vitro, in a similar role as the STI1-II domain of UBQLN2. Critically, we obtained residue-by-residue information using NMR (nuclear magnetic resonance) spectroscopy for >84% of full-length Dsk2. We identified three transient helical regions within the disordered regions of Dsk2 that interact with Dsk2's own STI1 domain. In vivo, we demonstrated that the deletions of the STI1 domain or the transient helices impact proteasome condensate formation in a stress-dependent manner. Our data suggest that the interaction between the STI1 domain and transient helices within Dsk2 represents a conserved feature among related UBQLNs and co-chaperone proteins containing STI1 domains (Sti1, HIP, SGTA, Tic40).

## Results

### Dsk2 phase separates under physiological conditions, enhanced by polyUb substrates

Ub-binding shuttle proteins Dsk2 and Rad23 are required for proteasome condensate formation under certain stress conditions in yeast (Waite et al, 2024). To determine the role of Dsk2 in proteasome condensates, we reconstituted a minimal system in vitro using only Dsk2, either K48-linked or K63-linked polyUb-substrates, and yeast proteasome (Fig. 1A). We chose K48-linked and K63-linked polyUb chains because they are the two most abundant polyUb chain types in cells and are both involved in maintaining proteostasis (Pickart and Fushman, 2004; Komander and Rape, 2012; Ohtake et al, 2016; Grumati and Dikic, 2018). The presence of polyUb-substrates is essential as multivalency of a polyUb chain makes it suitable for promoting phase separation of UBQLNs (Valentino et al, 2024; Dao et al, 2022; Galagedera et al, 2023), while monoUb dissolves UBQLN2 droplets (Dao et al, 2018). Using fluorescence microscopy, we tested and determined that Dsk2 phase separates with polyUb-substrates in a concentration-dependent manner (Fig. 1B), consistent with our recent results on UBQLN2 and polyUb-substrates (Valentino et al, 2024). Furthermore, we observed colocalization of proteasome, Dsk2, and K48-

linked or K63-linked polyUb-substrates in droplets (Fig. 1C), consistent with previous observations in yeast (Waite et al, 2024). In the absence of both the proteasome and polyUb-substrates, Dsk2 phase separates at concentrations reaching 10 μM at 5 °C in the presence of a higher amount of macromolecular crowder polyethylene glycol (PEG) (Fig. 1D). We obtained a phase diagram of Dsk2 by measuring the saturation concentration ($c_{sat}$) of Dsk2 as a function of temperature (see "Methods"). These data are consistent with Dsk2 following an upper critical solution temperature (UCST) phase transition, whereby decreasing temperature favors phase separation.

### Short helices exist within intrinsically disordered regions of Dsk2

To map out different structural elements within Dsk2 on a residue-by-residue basis, we used biomolecular NMR spectroscopy as previously employed in (Dao et al, 2018, 2022). We expressed and purified [15]N full-length Dsk2 (Dsk2 FL) protein ("Methods", Appendix Fig. S2 and Appendix Table S1). We collected [1]H–[15]N TROSY-HSQC NMR spectra (Fig. 2A) to monitor backbone amide resonances at 50 μM protein concentration, where Dsk2 is a monomer according to size-exclusion chromatography—multi-angle light scattering (SEC-MALS) experiments (Appendix Fig. S3). Consistent with the presence of intrinsically disordered regions (IDRs) in Dsk2, a majority of amide resonances are located in the middle of the spectrum (7.5–8.5 ppm in [1]H) (Dao et al, 2018; Burke et al, 2015). Peaks located on either side of this middle region correspond to residues in the well-folded UBL and UBA domains, as previously determined (Zhang et al, 2009). Using triple resonance NMR experiments with several domain-deletion variants of Dsk2, we obtained chemical shift assignments for 84% of all Dsk2 FL backbone amide resonances (see "Methods"). Most of the missing assignments correspond to resonances in the STI1 domain, suggesting increased dynamics within the STI1 domain.

We probed secondary structure content of Dsk2 using [13]C backbone $C_\alpha$-$C_\beta$ chemical shifts, where positive values indicate α-helical content and negative values represent β-strands (Fig. 2B). Our experimentally determined $C_\alpha$-$C_\beta$ values for Dsk2 correlate very well with predicted $C_\alpha$-$C_\beta$ values determined from the Alphafold-derived structure of Dsk2 (Fig. 2B,C; see "Methods") (Gu et al, 2024). The STI1 domain is mostly helical with the magnitude of α-helical propensity comparable to well-structured helices in the UBA domain. By contrast, the regions in between the UBL, STI1, and UBA domains are largely disordered. We identified three regions (114–134, 279–291, and 303–313) with low-moderate helical propensity, indicating the presence of putative α-helices. To corroborate these observations, we collected NOESY spectra that

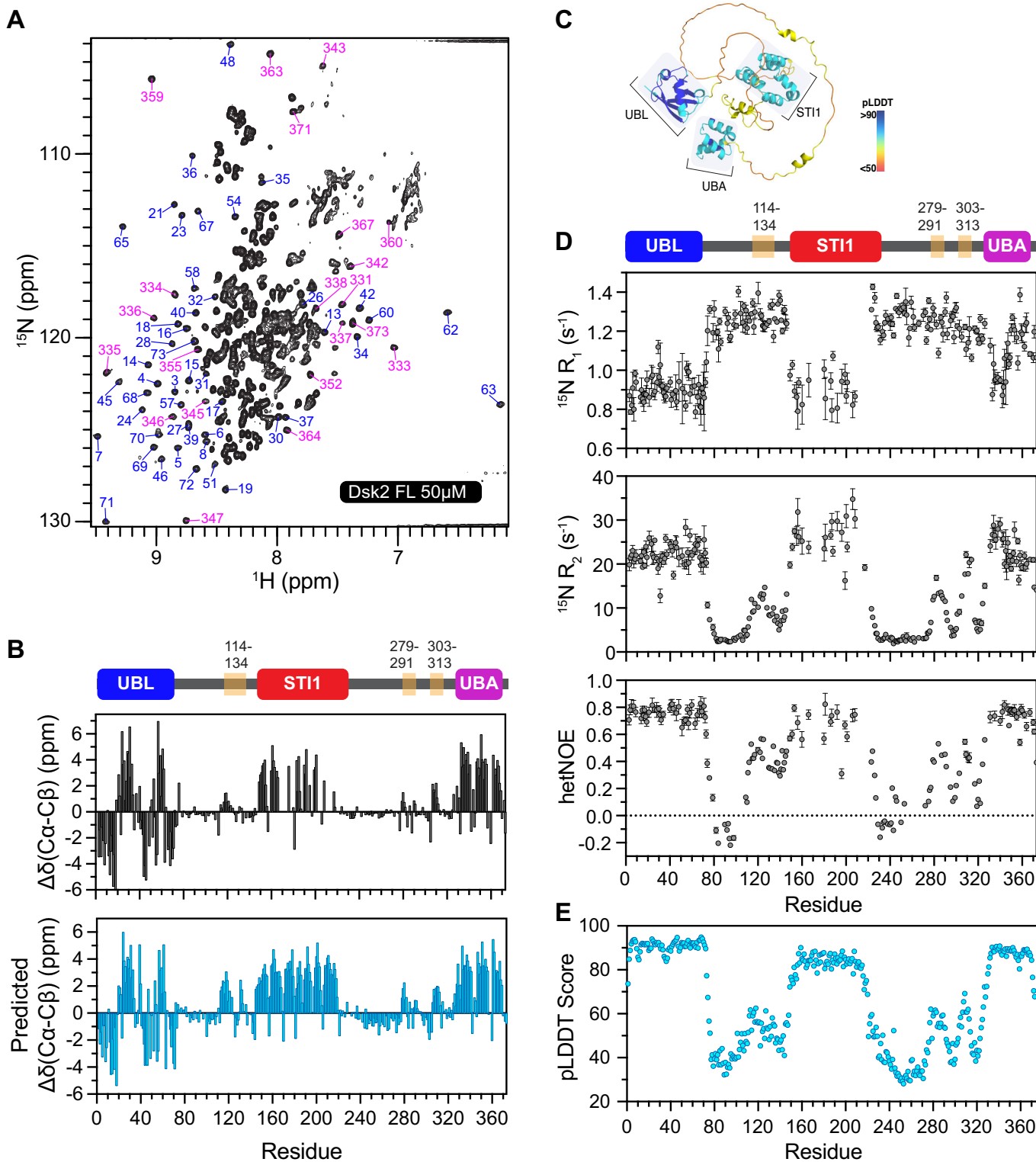

identify through-space H$^N$–H$^N$ interactions. We detected strong i ± 1 NOEs for some of these helical regions (Appendix Fig. S4), reflecting close spatial proximity between sequential residues within the helices (Eliezer et al, 2000).

To map backbone amide dynamics on a residue-by-residue level, we measured backbone $^{15}$N R$_1$, $^{15}$N R$_2$, and {$^1$H–$^{15}$N} heteronuclear

Overhauser enhancement (hetNOE) experiments (Fig. 2D). R$_1$ relaxation rates monitor nanosecond–picosecond (ns–ps) backbone dynamics, while R$_2$ relaxation rates are also sensitive to both ns–ps and slower millisecond–microsecond backbone dynamics. hetNOE reports on local dynamics at the ns–ps scale. Resonances within the UBL, STI1, and UBA domains exhibited high hetNOE values

**Figure 2.  Dsk2 is a multidomain protein with three transient helical regions in intrinsically disordered regions.**

(A) $^1$H–$^{15}$N TROSY-HSQC spectra of 50 µM Dsk2 FL at 25 °C in NMR buffer (20 mM NaPhosphate, 0.5 mM EDTA, 0.02% NaN3 pH 6.8). Resonances for UBL (blue) and UBA (magenta) domains are annotated. (B) Residue-level secondary structure comparison between experimentally determined $C_\alpha$-$C_\beta$ chemical shifts (top) and EFG-CS predicted $C_\alpha$-$C_\beta$ chemical shifts from Dsk2 AlphaFold structure (bottom; See "Methods"). Positive and negative $\Delta\delta C_\alpha$-$C_\beta$ values indicate α-helix and β-sheet content, respectively. (C) AlphaFold-predicted structure of Dsk2. (D) Amide backbone $^{15}$N $R_1$ relaxation rates, $^{15}$N $R_2$ relaxation rates, and {$^1$H}–$^{15}$N hetNOE values are plotted for Dsk2 FL at 50 µM; $n = 1$. For $R_1$ and $R_2$, values are the best model parameters from fit (see "Methods"). Error bars in $R_1$ and $R_2$ are the standard deviation from 500 Monte Carlo trials using RELAXFIT on MATLAB. Error bars in hetNOE measurements were determined using the standard error propagation formula (propagated uncertainties from spectral noise). (E) AlphaFold-predicted local distance difference test (pLDDT) score for Dsk2. Predicted data in (B, E) are colored skyblue for clarity.

(>0.6), consistent with these residues being part of structured regions in the protein. Elevated $R_2$ rates and lowered $R_1$ rates among these residues suggest similar rotational tumbling times, likely due to these domains being of similar sizes. In contrast, resonances within disordered regions have low $R_2$ relaxation rates (2–4 s$^{-1}$) and low hetNOE values (~0.1 or lower), similar to previously reported data for IDRs (Dao et al, 2018; Burke et al, 2015). Notably, the three putative helical regions have intermediate hetNOE values (~0.2–0.4), suggestive of their transient helical character (hereafter referred to as transient helices (TH)). Remarkably, the pattern of $R_2$ relaxation rates and hetNOE values closely matched the AlphaFold pLDDT scores across the entire protein (Fig. 2E). High pLDDT scores indicate strong confidence in the structural prediction, and these regions correspond to the well-folded domains in Dsk2. All three transient helices (hereafter 3TH when referring to all three regions together) exhibit intermediate pLDDT scores, consistent with their intermediate hetNOE values and $R_2$ relaxation rates. Together, these data show that the UBL, STI1, and UBA domains are well-folded, with the 3TH regions exhibiting transient helical propensity.

### Multivalent interactions drive Dsk2 self-association

Macromolecular phase separation can be driven by multivalent interactions among folded and/or disordered regions in proteins (Pappu et al, 2023; Martin et al, 2021). Using SEC-MALS, we determined that Dsk2 predominantly exists as a monomer at 150 µM, but we do observe a decrease in the elution volume as protein concentration is increased to 300 µM. This indicates that Dsk2 forms dynamic self-associating assemblies as protein concentration increases (Appendix Fig. S3). To assess the residue-level contribution of different domains/regions in Dsk2 self-association, we measured concentration-dependent effects by comparing $^1$H–$^{15}$N NMR spectra of FL Dsk2 between 50 µM and 400 µM, representing monomer and dynamic self-association, respectively (Fig. 3A). Changes to NMR spectra at 400 µM stem from both inter- and intramolecular interactions. Strikingly, at high protein concentration, many peaks for residues in the STI1 domain either disappeared or exhibited ~90% decrease in peak intensity (Fig. 3A, inset). The peak intensities in the UBL and UBA domains decreased by 50–70%. Intensity changes in UBL and UBA resonances stem from previously known UBL-UBA interactions common to Ub-binding shuttle proteins (Lowe et al, 2006; Zientara-Rytter and Subramani, 2019; Zheng et al, 2021). We validated these UBL-UBA interactions by comparing chemical shifts of the UBL or UBA domains in the absence and presence of the rest of the Dsk2 protein (Fig. EV1A–C). In addition, there was a moderate decrease in peak intensities and subtle chemical shift

perturbations (CSPs) for the 3TH regions (Fig. 3B,C). The concentration-dependent changes in $^{15}$N $R_2$ relaxation rates also point towards self-association involving multiple regions of Dsk2 (Fig. EV1D). Together, these data implicate the UBL, UBA, STI1, and the 3TH regions in Dsk2 self-association.

By far, the largest changes in peak intensity occur in STI1 resonances, suggestive of intermolecular STI1-STI1 interactions and/or STI1 interactions with other regions of Dsk2 at an intramolecular or intermolecular level. To test this, we monitored concentration-dependent changes in peak intensity for several domain-deletion variants of Dsk2, including ΔUBL and two constructs of Dsk2 lacking both UBL and UBA domains (Fig. EV1E). Resonances corresponding to residues from the STI1 domain consistently exhibited the largest decreases in peak intensity (70–90%) at high protein concentration, suggestive of STI1-STI1 self-association even in the absence of UBL and UBA domains. Our data indicate that the STI1 domain is a major contributor to Dsk2 self-association.

### STI1 domain interacts with transient helices in Dsk2

To investigate how the STI1 domain interacts with the rest of the Dsk2 protein, we made a Dsk2 ΔSTI1 (STI1 deletion) construct and compared $^1$H–$^{15}$N NMR spectra of Dsk2 FL and Dsk2 ΔSTI1 at identical protein concentrations (50 µM) (Fig. 4A). We observed that the only amide resonances to exhibit large changes in chemical shifts upon STI1 deletion are in the 3TH regions (Fig. 4B,C). To validate STI1–3TH interactions, we titrated unlabeled isolated Dsk2 STI1 domain into a $^{15}$N Dsk2 ΔSTI1 sample; this experiment resulted in shifts of 3TH resonances toward their Dsk2 FL positions (Fig. 4A; Appendix Fig. S5). Removing the STI1 domain also selectively impacted the backbone dynamics of 3TH (Fig. 4D). Strikingly, deletion of the STI1 domain resulted in a significant decrease of $R_2$ relaxation rates for all three helical regions. The STI1 deletion-induced changes in 3TH backbone dynamics could result from: (a) removal of STI1–3TH interactions, or (b) destabilization of the secondary structure of these transient helices. However, the latter is unlikely as the measured hetNOE values (Fig. 4D) and α-helix propensity (Fig. 4E) for the 3TH regions in ΔSTI1 were similar when compared with data for Dsk2 FL.

To model STI1–3TH interactions, we conducted molecular dynamics simulations for full-length Dsk2 using CALVADOS3$_{COM}$, a coarse-grained model for disordered and multidomain proteins where each residue is represented as a single bead (Tesei et al, 2021; Tesei and Lindorff-Larsen, 2023; Cao et al, 2024). In these simulations, the UBL, STI1, and UBA were treated as folded domains (based on the Dsk2 AlphaFold structure) held together by elastic constraints (Cao et al, 2024), with the rest of the protein

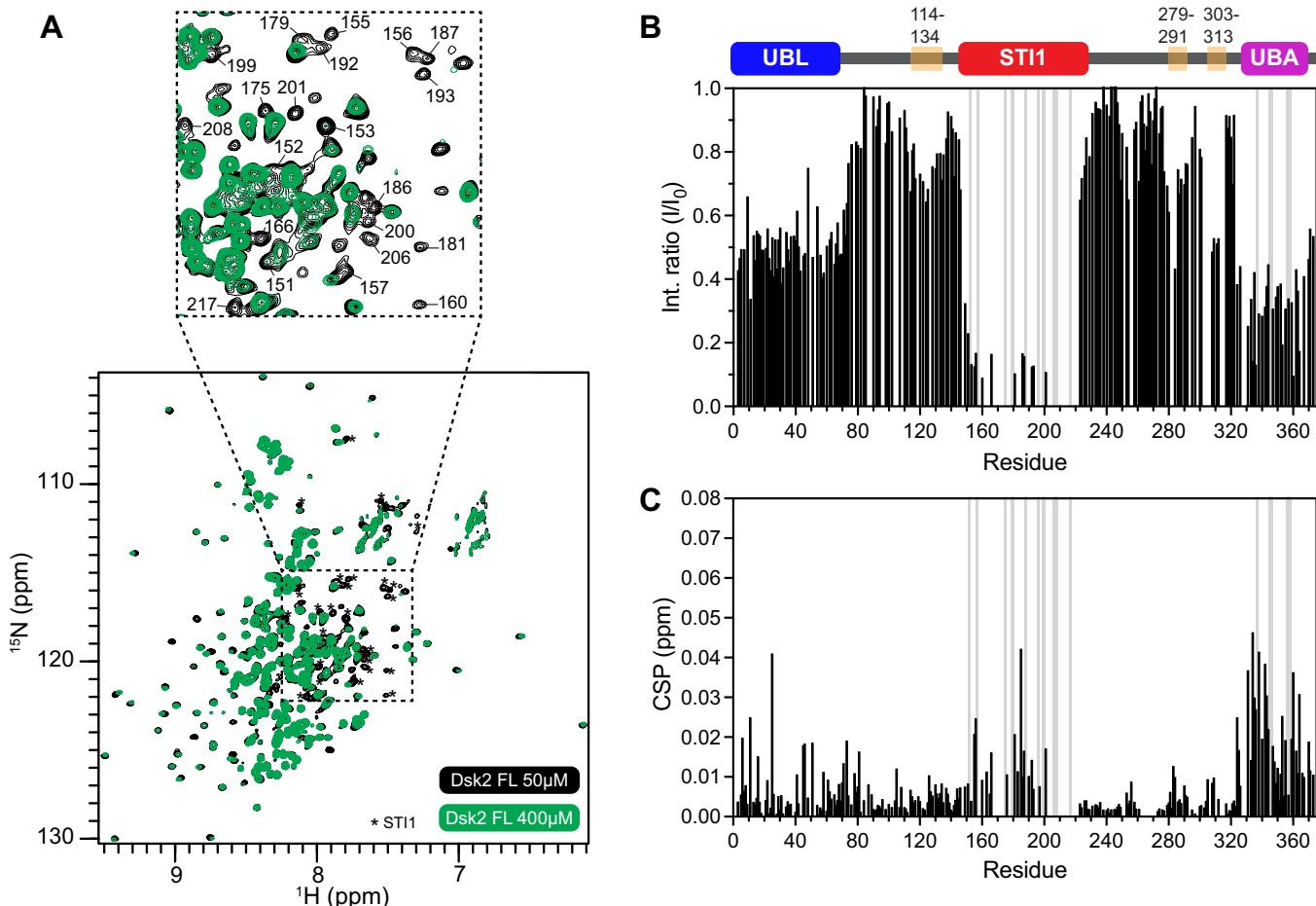

**Figure 3. NMR reveals multivalent interactions in Dsk2 self-association.**

(A) Comparison of $^1H$–$^{15}N$ TROSY-HSQC spectra of 50 μM and 400 μM full-length (FL) Dsk2. The inset shows significant line broadening for select amide resonances at 400 μM (green), primarily in the STI1 domain. Spectra were collected under identical acquisition parameters ("Methods"). (B, C) Residue-specific intensity ratio ($I/I_0$) between 400 μM (I) and 50 μM ($I_0$) resonances or chemical shift perturbation (CSPs) of amide resonances for the same concentration range. Gray bars represent resonances with no observable peak at 400 μM protein concentration.

being disordered. From these single-protein simulations, we noted that the disordered regions of Dsk2 transiently occupied the STI1 groove (Movie EV1). We quantified the ensemble-averaged probability of the disordered regions occupying the STI1 groove, and specifically observed an increased preference of all three TH regions to interact with STI1 (shaded orange regions in Fig. 4F). However, in validating these simulations, we found that the average predicted radius of gyration ($R_g$) value for these simulations ($47.0 \pm 0.2$ Å) was larger than the experimental $R_g$ value ($37.9 \pm 0.7$ Å) obtained from small angle X-ray scattering (SAXS) data for full-length Dsk2 (Fig. 4G; Appendix Fig. S3 and Appendix Table S2). We hypothesized that this difference stemmed from the known UBL–UBA association (Lowe et al, 2006) that was also observed in our NMR data (Fig. EV1). To test this hypothesis, we placed elastic constraints that held together the UBL and UBA domains in the same orientation as the previously solved crystal structure 2BWE of the two isolated domains (see "Methods", Fig. EV1C). Bound UBL–UBA simulations showed quantitative agreement with experimental SAXS data for full-length Dsk2 (Fig. 4G; Movie EV2). In addition, keeping the UBL and UBA in a

bound conformation resulted in an increased preference for TH2 and TH3 regions to interact with the STI1 domain (Fig. 4F), in line with what we observed by NMR (Fig. 4B). Overall, the probabilities for the TH regions to occupy the STI1 groove are significantly higher compared to what is expected in the absence of specific chemical interactions (roughly 50-, 75-, and 150-fold higher for TH1, TH2, and TH3, respectively, Appendix Fig. S6). Together, these data suggest that the STI1 domain has an increased probability of interacting with the 3TH regions in the IDRs of Dsk2 (Movie EV2; Fig. 4H).

## STI1 domain and transient helices are major contributors to Dsk2 phase separation

NMR analysis and simulations revealed distinct multivalent interactions within Dsk2, including UBL-UBA, STI1-STI1, and STI1–3TH intramolecular and intermolecular interactions. We assessed the differential contributions of these interactions in Dsk2 phase separation, which depends on intermolecular interactions. We obtained phase diagrams of different domain-deletion and

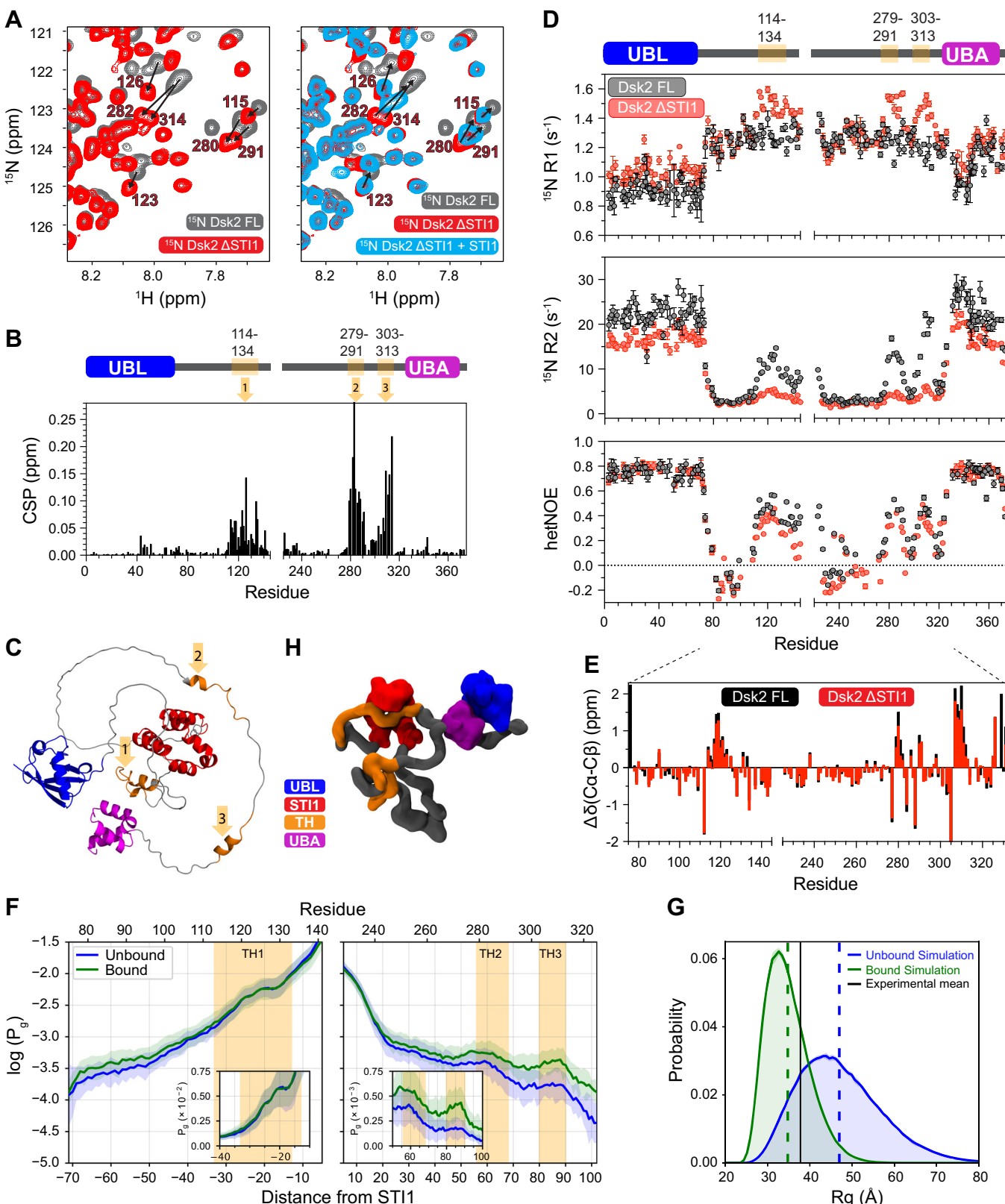

◀ **Figure 4. STI1 domain interacts with transient helices present in the IDRs of Dsk2.**

(A) Comparison of $^1H$–$^{15}N$ TROSY-HSQC spectra of 50 μM Dsk2 FL (black) and Dsk2 ΔSTI1 (red) at 25 °C focused on amide resonances from the three transient helical regions (3TH) in Dsk2 ΔSTI1. (Right) Addition of 8X isolated unlabeled STI1 domain to $^{15}N$ Dsk2 ΔSTI1 moved amide resonances of 3TH regions towards their respective positions in Dsk2 FL. See Appendix Fig. S5 for full spectra. (B, C) Backbone amide CSPs between Dsk2 FL and Dsk2 ΔSTI1 are shown in (B), with the largest chemical shift differences for residues from 3TH regions in the IDRs of Dsk2 highlighted by orange arrows and mapped onto Dsk2 AlphaFold structure in (C). (D) $^{15}N$ $R_1$, $^{15}N$ $R_2$ relaxation rates, and hetNOE values are compared for Dsk2 FL (black) and Dsk2 ΔSTI1 (red) at 50 μM 25 °C; $n = 1$. For $R_1$ and $R_2$, values are the best model parameters from fit (see "Methods"). Error bars in $R_1$ and $R_2$ are the standard deviation from 500 Monte Carlo trials using RELAXFIT on MATLAB. Error bars in hetNOE measurements were determined using the SE propagation formula (propagated uncertainties from spectral noise). (E) Secondary structure content in the IDRs of Dsk2 FL and Dsk2 ΔSTI1 using $C_\alpha$-$C_\beta$ chemical shift differences. (F) Log of the probability (inset: linear probability) of the disordered region residues occupying the STI1 groove for CALVADOS simulations with the UBL and UBA unbound (blue) or bound (green). TH regions are colored orange. The shaded region represents the differences among the replicates. The normalized excess probability plot corrected for excluded volume is shown in Appendix Fig. S6 (see "Methods"). (G) Simulation-derived probability distribution of $R_g$ values compared between the UBL-UBA unbound (blue) and bound (green) forms of Dsk2 with average $R_g$ values represented as dashed lines. Experimental $R_g$ from SAXS is shown as black solid line. (H) A single snapshot from bound UBL-UBA simulations showing a transient helix (orange) occupying the STI1 groove (red). See Movie EV2 for the full STI1 groove occupancy time course.

mutant constructs of Dsk2 (Fig. 5A) and used bright-field microscopy to monitor droplet formation (Fig. 5B). To disrupt UBL-UBA interactions, we expressed and purified Dsk2 ΔUBL and Dsk2 mutUBA (G343A/F344A double mutant) (Waite et al, 2024; Sasaki et al, 2005). For both Dsk2 ΔUBL and Dsk2 mutUBA constructs, the loss of UBL-UBA interactions reduced phase separation propensity, as higher protein concentrations were required to induce phase separation (rightward shift in their phase diagrams compared to Dsk2 FL) (Fig. 5A).

Removal of the three transient helical regions (Dsk2 Δ3TH) or the STI1 domain (Dsk2 ΔSTI1) increased $c_{sat}$ values for phase separation compared to the FL, ΔUBL, and mutUBA constructs (Fig. 5A). We observed phase separation for Dsk2 Δ3TH and Dsk2 ΔSTI1, but only at substantially higher protein concentrations (Fig. 5B). We propose that the reduction in phase separation propensity stems from the loss of multivalent interactions when all three transient helical regions are removed, impacting STI1-TH1, STI1-TH2, and STI1-TH3 interactions. The removal of the STI1 domain, due to eliminating STI1-STI1 interactions as well, led to the largest decrease in phase separation propensity of all constructs tested. Given that the STI1 domain mediates multiple interactions in Dsk2 self-association, including STI1-STI1 and STI1–3TH interactions (STI1-TH1, STI1-TH2, and STI1-TH3) (Fig. 5C,D), we conclude that the STI1 domain is the major contributor to both self-association and phase separation in vitro.

## Deletion of the Dsk2 STI1 domain or 3TH regions impacts proteasome condensate formation in vivo

In yeast, the formation of proteasome condensates induced by sodium azide (mitochondrial inhibition) or prolonged growth (gradual carbon starvation) depends on the presence of poly-ubiquitinated substrates and the shuttle factors Dsk2 and Rad23 (Waite et al, 2024). We hypothesized that changes in Dsk2 phase-separation propensity, specifically the disruption of STI1–3TH interactions, would also impact proteasome condensate formation in vivo (hereafter, proteasome condensates refer to Dsk2/proteasome condensates unless specified otherwise). We used CRISPR/Cas9 to delete the genomic sequences in yeast corresponding to either the STI1 domain (*DSK2 ΔSTI1*) or the 3TH regions (*DSK2 Δ3TH*). These deletions were introduced in *rad23Δ* cells (RAD23 deletion strain) expressing GFP-tagged proteasomes (Rpn1-GFP); the *rad23Δ* background was selected to eliminate redundancy from Rad23 in proteasome condensate formation (Waite et al, 2024).

Cells were subjected to either sodium azide stress (Fig. 6A–F) or prolonged growth stress (3 days in rich medium; Fig. 6G–L) (Fig. EV2A for schematic).

Contrary to expectations, deletion of the STI1 domain increased the percentage of yeast cells with at least one proteasome punctum under both stress conditions (Fig. 6A,B,G,H). As noted from our in vitro observations, increased protein concentrations can drive phase separation of Dsk2 constructs (Δ3TH and ΔSTI1) despite their low phase separation propensity (Fig. 5B, bottom panels). To monitor Dsk2 protein levels in response to stress, we generated yeast strains expressing ALFA-tagged Dsk2 (Dsk2[ALFA]) and determined Dsk2 abundance by immunoblotting (Fig. 6C,I; Appendix Fig. S7). Deletion of the STI1 domain resulted in a roughly 2.5-fold higher Dsk2[ALFA] level (compared to WT) under both stress conditions, consistent with the expectation that higher protein concentration promotes condensate formation. Furthermore, we observed an increase in K48-linked polyUb levels in *DSK2 ΔSTI1* cells under both stress conditions (Fig. 6D,E,J,K); similarly, K48-linked polyUb also contributes favorably to the condensate formation in vitro (Fig. 1B,C). Deletion of the STI1 domain had no detectable change in colocalization of Dsk2 and proteasome puncta (Fig. EV2D,K). Together, these data suggest that deletion of the STI1 domain of Dsk2 promotes proteasome condensate formation in vivo, driven in part by higher Dsk2 protein levels and the accumulation of polyubiquitinated proteins.

In line with our in vitro phase separation results (Fig. 5), removing the three transient helical regions (Δ3TH) significantly reduced azide-induced proteasome puncta/condensate formation compared to WT (Fig. 6A,B). Notably, the effect of 3TH deletion on condensate formation significantly differed from ΔSTI1 (both compared to WT). *Δ3TH* cells did not exhibit elevated Dsk2[ALFA], K48-linked polyUb levels, or proteasome (Rpn1-GFP) levels (Fig. 6C–F). Under prolonged growth stress, *Δ3TH* cells displayed increased proteasome condensates compared to WT (Fig. 6G,H), but we believe this stems from (a) *Δ3TH* cells having >twofold higher Dsk2[ALFA] levels compared to WT (Fig. 6I) and (b) slightly elevated K48-linked polyUb levels and proteasome (Rpn1-GFP) levels that together could promote condensate formation relative to WT (Fig. 6J–L). Similar trends and stress-dependent effects were observed in the yeast strains expressing endogenous Rad23 (Fig. EV2), although the impact on proteasome puncta formation was largely masked, presumably due to Rad23's redundancy in proteasome condensate formation. Overall, our data are consistent with our hypothesis that removal of the transient helices reduces

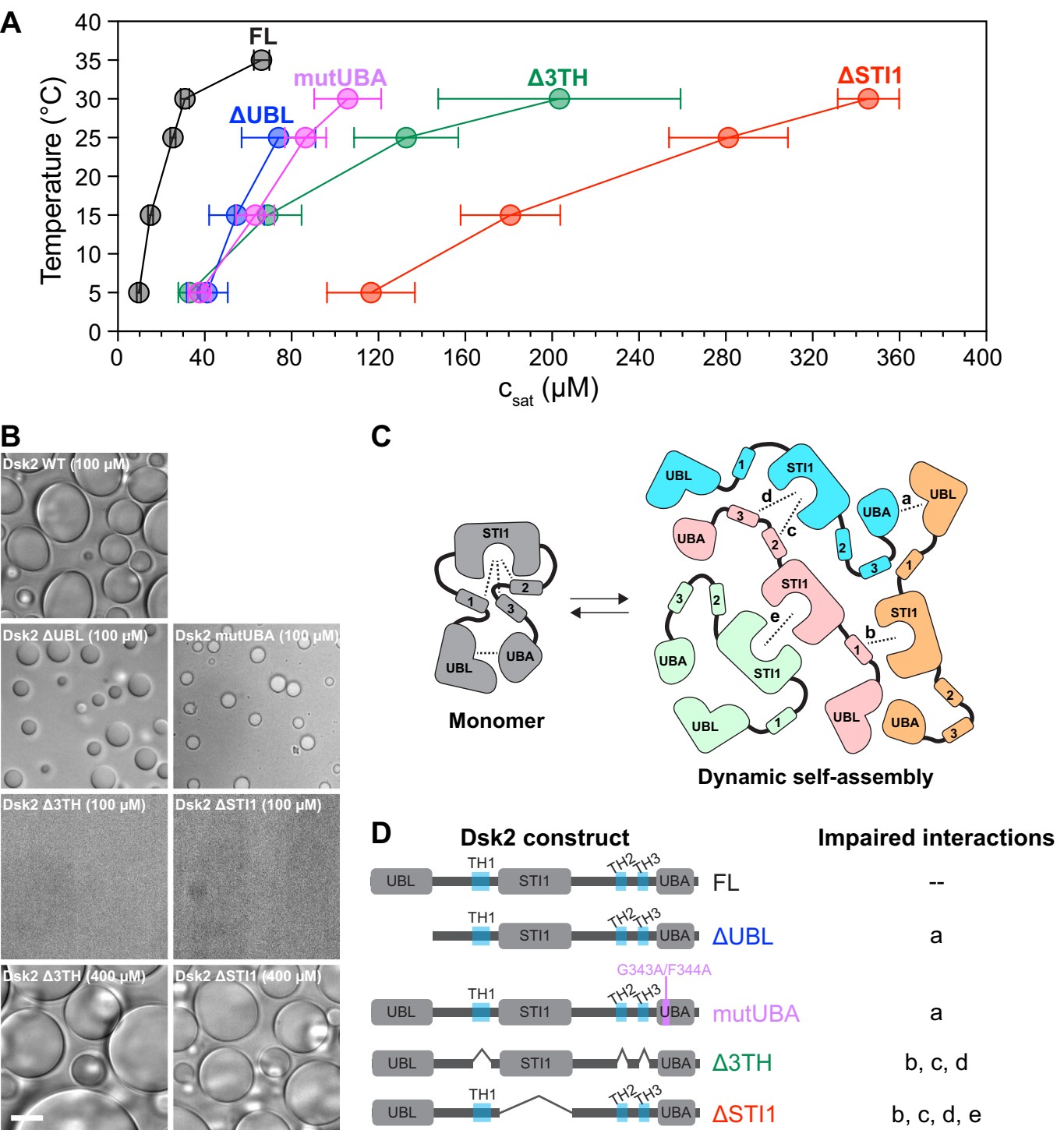

**Figure 5. STI1 domain and transient helical regions are important drivers of Dsk2 phase separation.**

(A) Phase diagrams of Dsk2 constructs compared against Dsk2 FL. Values are the mean ± standard deviation (SD) over two or more protein preps in triplicate. (B) Bright-field microscopy of Dsk2 constructs at 100 μM protein concentration (unless otherwise specified) incubated at 25 °C for 20 min. Scale bar: 10 μm. (C) Representative model depicting interactions among different regions of Dsk2 (represented by dashed lines and further described in panel (D) important for Dsk2 self-association and phase separation. Monomeric units are shown in different colors where no more than two interactions among domains/regions are highlighted between monomers for clarity. (D) Domain architectures of Dsk2 constructs alongside impaired interactions.

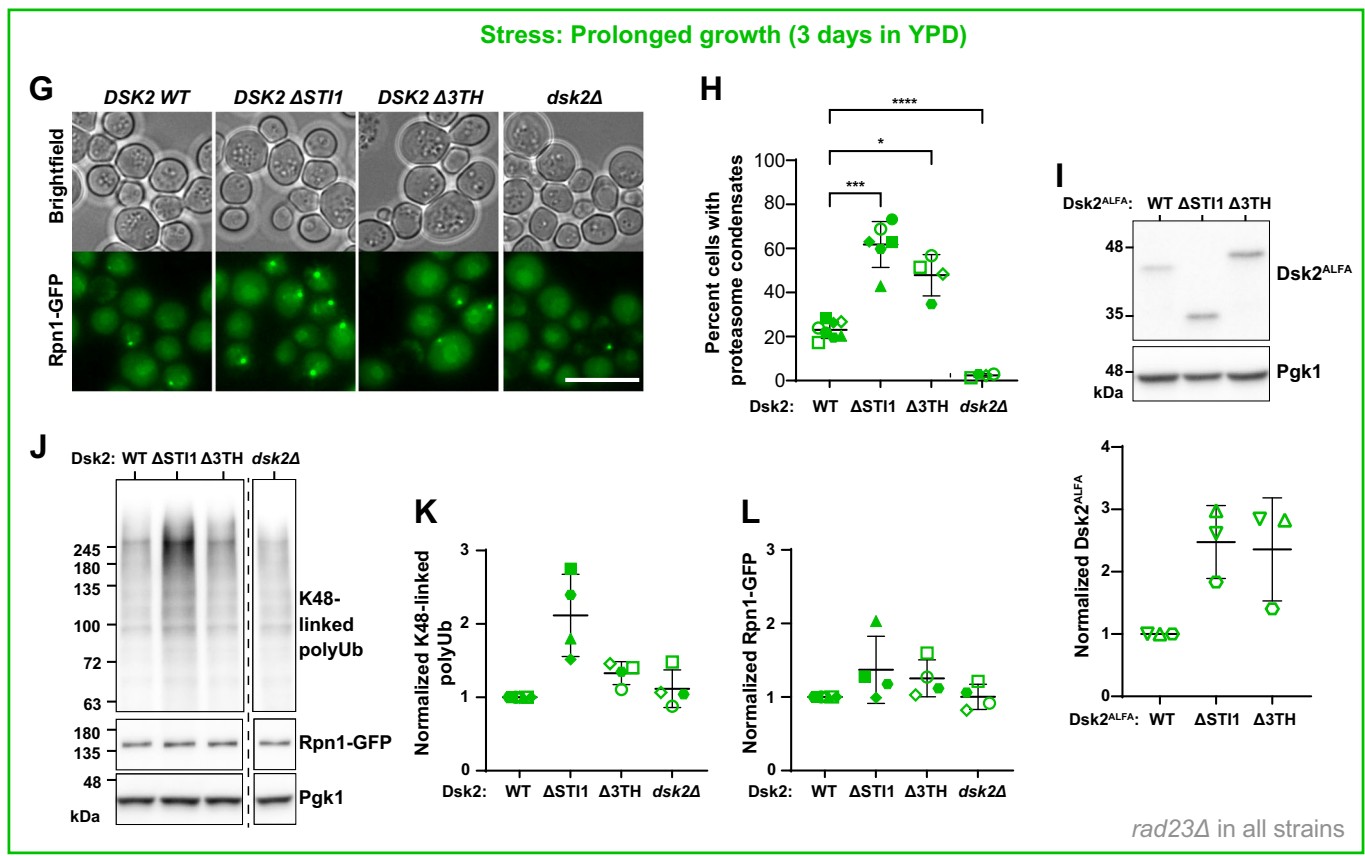

**Figure 6. Deletion of the Dsk2 STI1 domain and 3TH regions impacts stress-induced proteasome puncta formation.**

*S. cerevisiae* yeast strains (in *rad23Δ* background) were subjected to (**A–F**) ~ 24 h azide stress or (**G–L**) prolonged growth stress (3 days in YPD). (**A, G**) Representative bright-field and extended-depth-of-field epifluorescence images (GFP channel) showing proteasome puncta (endogenous Rpn1-GFP) in strains expressing Dsk2 variants. Scale bar: 10 μm. (**B, H**) Percentage of cells with ≥1 punctum after stress. (**B**) **$P = 0.0023$, ***$P = 0.001$, ****$P = 0.000048$; one-way ANOVA with Dunnett's post-hoc test ($α = 0.05$), $n ≥ 4$. (**H**) *$P = 0.0301$, ***$P = 0.0007$, ****$P = 0.000003$; one-way ANOVA with Dunnett's post-hoc test ($α = 0.05$), $n ≥ 4$. (**C, I**) Representative immunoblot of whole-cell lysates showing relative levels of endogenous Dsk2$^{ALFA}$ variants normalized to Pgk1 and scaled to Dsk2$^{ALFA}$ WT. $n = 3$. (**D–F, J–L**) Representative immunoblot of whole-cell lysates from Dsk2 variant strains showing (**E, K**) K48-linked polyubiquitin and (**F, L**) Rpn1-GFP levels, normalized to Pgk1 and scaled to Dsk2 WT. $n = 4$. Each biological replicate is denoted by a single symbol wherever applicable. On plots, the horizontal line and error bars represent the mean and SD, respectively. Full blots in Appendix Figs. S8 and S9.

Dsk2 condensate formation in vitro and in vivo, with discrepancies explained by elevated Dsk2 and/or polyUb levels.

## Discussion

Here, we comprehensively mapped multivalent interactions across both folded and intrinsically disordered domains of full-length Dsk2 on a residue-by-residue basis using NMR spectroscopy (Fig. 2). No other UBQLN family member has been examined at this level as many resonances are significantly broadened beyond detection, specifically STI1 domains in human UBQLNs (Dao et al, 2018; Zheng et al, 2021). Among the identified multivalent interactions in Dsk2, the STI1 domain and three transient helices in disordered regions emerged as the major contributors to both Dsk2 self-association and phase separation. We further reconstituted Dsk2 condensates with colocalized proteasomes and polyUb-substrates, showcasing how increasing concentration of PQC components, particularly long polyUb chains, can drive condensation of Dsk2 (Fig. 1B).

Consistent with prior work on Dsk2 and human ortholog UBQLN2, our NMR and SAXS data indicate that the UBL and UBA domains interact with each other (Figs. EV1A–C and 4). The CSPs for the UBL and UBA domains (when comparing full-length Dsk2 at monomeric concentration with the isolated domains) map to the previously characterized UBL-UBA interface (Lowe et al, 2006). Simulations of the Dsk2 monomer (where the UBL and UBA domains are unbound) resulted in a calculated effective concentration for these domains of $413.5 ± 8.8\ μM$ (Sørensen and Kjaergaard, 2019; González-Foutel et al, 2022). This value is higher than the reported $K_d$ of 80 μM for the UBL-UBA interaction using isolated domains (Lowe et al, 2006), further supporting that UBL and UBA domains interact intramolecularly at low protein concentrations. However, these UBL-UBA interactions become intermolecular at higher protein concentrations, promoting self-association, as evidenced by the decrease in peak intensity observed in both UBL and UBA domains with increased protein concentration (Fig. 3B). Unlike UBQLN2, where deletion of the UBL domain enhances phase separation (Zheng et al, 2021), the removal of the UBL domain in Dsk2 reduces phase separation (Fig. 5A,B). This opposite behavior may stem from: (a) the deletion of UBQLN2 UBL also included N-terminal disordered residues 1–33 which regulate UBQLN2 phase separation (Dao et al, 2024) but are not present in Dsk2, (b) differences in oligomerization propensity for Dsk2 (monomer) and UBQLN2 (dimer), or (c) potentially different interaction patterns of the UBL domain with the rest of either Dsk2 or UBQLN2 (Zheng et al, 2021). Modulation of UBL-UBA

interactions by PQC components further changes Dsk2 phase separation propensity. As the concentration of polyUb-substrates is increased, Dsk2 phase separation is enhanced (Fig. 1B). This result agrees with proteasome condensate formation in yeast, where UBL and UBA domains of Rad23 and Dsk2 contribute significantly to condensation via their interactions with proteasomes and poly-ubiquitin chains (Waite et al, 2024).

Our NMR results and simulations point towards the existence of STI1-helix interactions (Fig. 4), which also contribute to Dsk2 self-association as protein concentration increases. Functionally, STI1 domains of co-chaperone proteins (Sti1, HIP, SGTA, Tic40) and adapters of the ubiquitin-proteasome system (Dsk2, UBQLNs, KPC2) are implicated in helix-mediated substrate interaction and/or activation (Fry et al, 2021; Schmid et al, 2012; Itakura et al, 2016; Lin et al, 2021). For co-chaperone Hop (an ortholog of yeast Sti1 protein), the DP2 domain (also known as STI1-II domain) directly interacts with the glucocorticoid receptor (GR) client protein via a GR amphipathic helix binding to the hydrophobic groove of the Hop DP2 domain (Fig. 7A) (Wang et al, 2022). Similarly, recent work revealed that a transmembrane helix binds to the hydrophobic groove of the Dsk2 STI1 domain from *M. bicuspidata* (Fig. 7B) (preprint: Onwunma et al, 2024). Structural modeling using AlphaFold2 multimer predictions ((Mirdita et al, 2022) and "Methods") suggests that the three amphipathic transient helices in Dsk2 each interact with the hydrophobic groove of the STI1 domain through their hydrophobic faces (Figs. 7C–E and EV3), reminiscent of previously characterized STI1-helix interactions in Hop and *M. bicuspidata* Dsk2 (Fig. 7A,B). These AlphaFold predictions are consistent with our CSP data (Figs. 4B and EV3A). In addition, STI1–3TH interactions in Dsk2 do not contribute to any significant local secondary structure stabilization in either region, as indicated by unchanged hetNOE values for 3TH regions following STI1 domain deletion (Fig. 4D), or for STI1 domain following 3TH deletion (Appendix Fig. S10).

The STI1 domain of the co-chaperone Sgt2 interacts with transmembrane domain (TMD) client helices with varying affinities depending on helix length and distribution of hydrophobic residues along the helix (Lin et al, 2021). Highly hydrophobic helices with >11 residues were strong binders to the STI1 domain. Similarly, the three transient helices in Dsk2 vary in helix length, number, and distribution of hydrophobic residues (Appendix Table S3), suggesting they may bind the STI1 domain with varying affinities. We speculate that each of these helices transiently occupies the STI1 hydrophobic groove (Fig. 7), as observed in our simulations (Fig. 4F; Movies EV1 and EV2). Such a mechanism of dynamic binding between STI1 and multiple helices could explain the weak or missing amide resonances of the Dsk2 STI1 domain even in

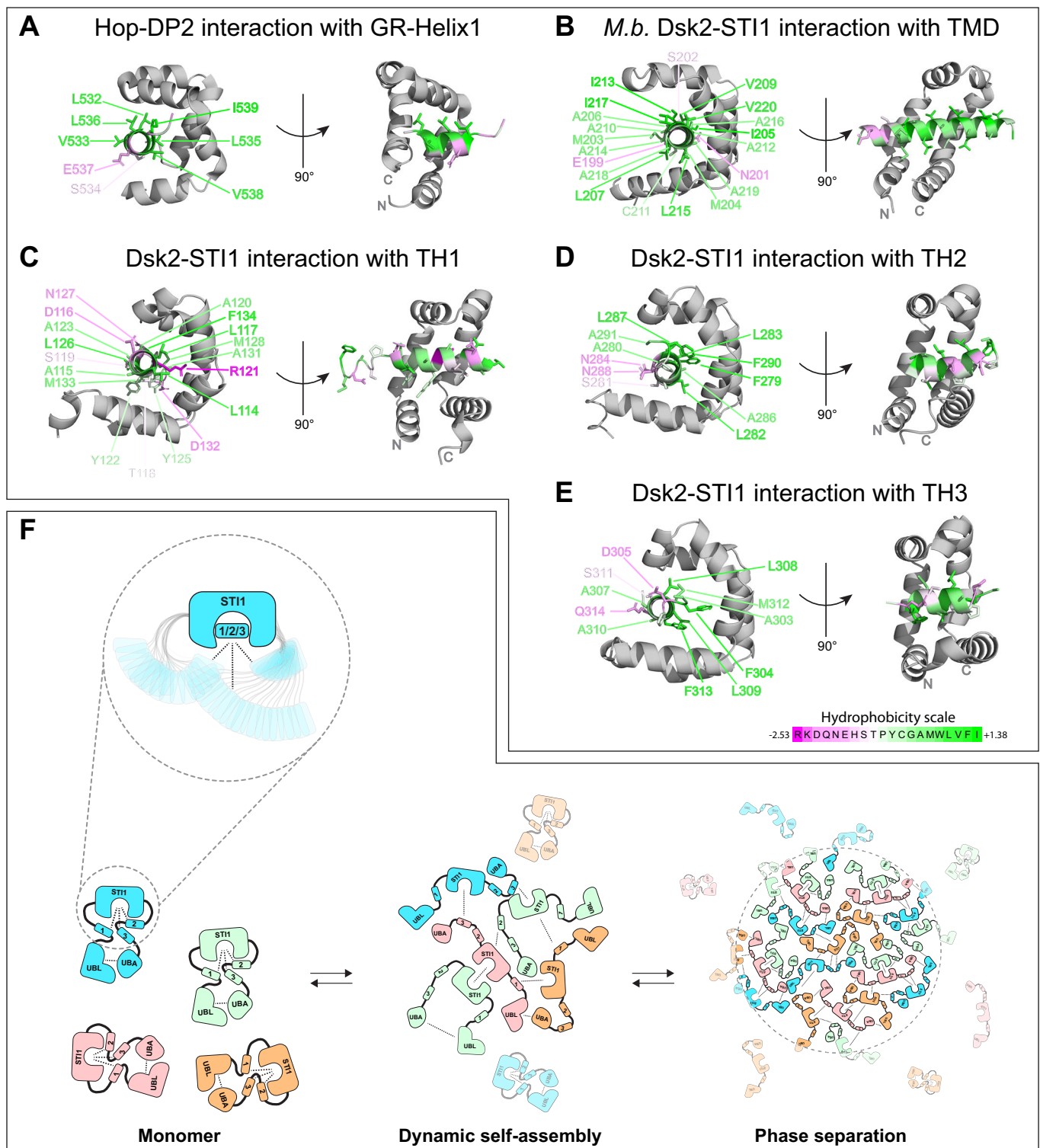

**A** Hop-DP2 interaction with GR-Helix1

**B** *M.b.* Dsk2-STI1 interaction with TMD

**C** Dsk2-STI1 interaction with TH1

**D** Dsk2-STI1 interaction with TH2

**E** Dsk2-STI1 interaction with TH3

Hydrophobicity scale
-2.53 R K D Q N E H S T P Y C G A M W L V F I +1.38

**F**

Monomer          Dynamic self-assembly          Phase separation

Dsk2 truncation constructs (Fig. EV1E). In the cell, we predict that the STI1–3TH interactions are displaced by STI1 substrates, as hypothesized in recent work (preprint: Onwunma et al, 2024). In addition, the transient helices of Dsk2 may bind to E3 ligases to facilitate ubiquitination and proteasomal degradation of certain STI1-bound Dsk2 client proteins, as suggested for UBQLN1

interactions with mitochondrial membrane proteins (Itakura et al, 2016). TH3 of Dsk2 exhibits some sequence similarity to the transient helix adjacent to the UBA domain (UBAA) in human UBQLNs (Appendix Fig. S1) that interacts with the AZUL domain in the E3 ligase E6AP to potentially recruit substrates for ubiquitination (Buel et al, 2023).

**Figure 7. Amphipathic or hydrophobic helices bind to the hydrophobic groove of STI1 domains.**

(A) Structure of STI1-II (DP2) domain of Hop (gray) bound to helix-1 of the glucocorticoid receptor client protein (PDB ID 7KW7). (B) Crystal structure of the STI1 domain (gray) of *M. bicuspidata* (*M.b.*) Dsk2 bound to a transmembrane helix (TMD) (PDB ID 9CKX); only relevant domains from the structures are shown in (A, B). (C–E) Representative AlphaFold2 multimer models are shown for the interaction between the STI1 domain of Dsk2 (gray) with transient helical regions TH1 (C), TH2 (D), and TH3 (E). Only relevant short segments of transient helices are shown for clarity (TH1: residue 114–134, TH2: 278–291, TH3: 303–315). See Fig. EV3A for full models. All helices in (A–E) are colored according to hydrophobicity (Eisenberg et al, 1984). (F) Proposed model for how multivalent interactions in Dsk2 drive its transition from monomer to dynamic self-assembly and macromolecular phase separation. The dynamic interactions between the transient helices within the disordered regions of Dsk2 and the STI1 domain are highlighted in the inset.

To assess the broader applicability of STI1-transient helix interactions, we generated a chimeric construct (Dsk2-Chimera), in which the STI1 domain of Dsk2 was replaced with the STI1-II domain of human UBQLN2 (residues 379–462 of UBQLN2), and compared NMR spectra against Dsk2 FL and Dsk2 ΔSTI1 (Fig. EV4). Notably, the chemical shifts of 3TH residues in Dsk2-Chimera closely resemble those of Dsk2 FL rather than Dsk2 ΔSTI1. This observation supports a general mechanism where the transient helices of UBQLNs engage with their STI1 domain, suggesting that such interactions may be conserved across STI1 domain-containing co-chaperone proteins.

Our NMR data support the existence of STI1-STI1 interactions that drive Dsk2 self-association (Figs. 3 and EV1E). In UBQLN2, we previously noted that STI1-II dimerization plays a pivotal role in driving UBQLN2 phase separation (Dao et al, 2018, 2024). Under identical protein concentrations and experimental conditions (50 μM, pH 6.8 buffer), Dsk2 prefers the monomeric state, unlike UBQLN2 (Appendix Fig. S3). However, we do observe concentration-dependent dynamic self-assembly of Dsk2 at higher protein concentrations (Fig. 3). Furthermore, a recent crystal structure of a fungal Dsk2 shows STI1 capable of forming a dimer holding two transmembrane helices in the middle (preprint: Onwunma et al, 2024).

Though multivalent interactions among different domains and regions potentiate self-assembly in Dsk2 (Figs. 3, 5, and 7F), the STI1 domain (STI1-II domain in case of UBQLN2) remains the key component driving the self-association and phase separation for both UBQLN2 (Dao et al, 2018) and Dsk2 (this study). The STI1 domain and its substrates are likely to regulate cellular Dsk2/proteasome condensate formation. Indeed, STI1 domain deletion in Dsk2 increased proteasome condensate formation in stressed yeast cells (Fig. 6A,B,G,H) in contrast to its in vitro phase separation behavior. In yeast, the STI1 domain of Dsk2 likely chaperones substrates, with failed substrates directed for polyUb-mediated degradation pathways (Itakura et al, 2016; Kurlawala et al, 2017). Consequently, deletion of the STI1 domain likely impaired Dsk2's chaperone activity on STI1-binding client proteins, leading to compensatory Dsk2 upregulation, and/or accumulation of impacted polyUb-substrates (Fig. 6C–E,I–K). Improper clearance of polyUb-substrates can lead to proteasome dysfunction and promote proteasome condensation, given the propensity for ubiquitinated substrates to drive condensation (Rajendran and Castañeda, 2025; Vamadevan et al, 2022; Goel et al, 2023; Valentino et al, 2024; Waite et al, 2024). Unlike *DSK2 ΔSTI1*, the *Δ3TH* strain preserves the STI1 domain and showed reduced proteasome condensation under azide stress (Fig. 6A,B), consistent with our in vitro phase separation data (Fig. 5). Under prolonged growth, *Δ3TH* exhibited increased proteasome condensate formation relative to WT, but this is accompanied by elevated Dsk2 levels

(Fig. 6G–I). We speculate that the absence of placeholder TH regions may enhance off-target interactions of the STI1 domain specifically over prolonged growth, leading to effects similar to those observed in *ΔSTI1* cells due to STI1 dysfunction, and distinguishing it from the azide stress response. Together, our findings suggest that the enhanced Dsk2/proteasome condensate formation observed in vivo in the *ΔSTI1* strain results from impaired STI1 activity, while the azide-treated *Δ3TH* strain, with a functional STI1 domain, shows reduced condensation in vivo in agreement with in vitro observations.

In summary, our work identifies and highlights the crucial role of multivalent interactions among different regions of Dsk2, particularly the STI1 domain and transient helices, in driving self-association and phase separation. Notably, interactions between the STI1 domain and transient helices closely resemble STI1-client helix interactions observed in similar STI1-containing proteins, suggesting interplay between the two sets of interactions. We speculate that different Dsk2 substrates can modulate Dsk2 inter- and/or intramolecular interactions, leading to promotion or inhibition of phase separation, both directly and indirectly, in vitro and in vivo. Putting together these observations, we propose that the STI1 domain and transient helices of Dsk2, key drivers of Dsk2 phase separation in vitro, are also key regulators of condensate formation in vivo.

## Study limitations

Our NMR results are collected in the absence of additional NaCl or PEG crowding agent; these conditions do not match our in vitro phase separation assays. NMR experimental conditions were collected as such to improve sensitivity following our previous study on UBQLN2 (Dao et al, 2018). A limitation of our proteasome condensate reconstitution experiments is that the concentration of Dsk2 used is an order of magnitude more than the endogenous concentration, which is estimated at ~0.3 μM (Ho et al, 2018), though stress conditions can alter protein levels. CALVADOS simulations are coarse-grained, and no helicity is assigned to transient helical regions. For in vivo experiments, we made ALFA-tagged Dsk2 strains to assess the protein level of wild-type or modified Dsk2 due to the unavailability of a suitable Dsk2 antibody for yeast lysates. Additionally, as tagging Dsk2 with a C-terminal fluorescent protein inhibits in vivo proteasome condensation formation (Waite et al, 2024), we used fluorescently tagged Dsk2 only to establish colocalization of Dsk2 with proteasome condensates. Stabilization or destabilization of Dsk2 resulting from the genetic or protein manipulations could affect proteasome condensate formation directly via altered protein-protein interaction landscape or indirectly via changes in proteostasis.

# Methods

### Reagents and tools table

| Reagent/resource | Reference or source | Identifier or catalog number |
|---|---|---|
| **Experimental models** | | |
| *S. cerevisiae* strains (Background: BY4742/BY4743) | | |
| sJR1255, *rpn1::RPN1-GFP (HIS3)* | Waite et al, 2024 | Appendix Table S5 |
| sJR2659, *rpn1::RPN1-GFP (HIS3), dsk2::DSK2-S145–N223del* | This study | Appendix Table S5 |
| sJR2689, *rpn1::RPN1-GFP (HIS3), dsk2::DSK2-L114–F134del, F279–A291del, M301–Q314del* | This study | Appendix Table S5 |
| sJR2691, *rpn1::RPN1-GFP (HIS3), dsk2::DSK2-E246_G247insALFA* | This study | Appendix Table S5 |
| sJR2746, *rpn1::RPN1-GFP (HIS3), dsk2::DSK2-S145–N223del, E246_G247insALFA* | This study | Appendix Table S5 |
| sJR2748, *rpn1::RPN1-GFP (HIS3), dsk2::DSK2-L114–F134del, E246_G247insALFA, F279–A291del, M301–Q314del* | This study | Appendix Table S5 |
| sJR1127, *rpn1::RPN1-GFP (HIS3), rad23Δ::KanMX* | Waite et al, 2024 | Appendix Table S5 |
| sJR2660, *rpn1::RPN1-GFP (HIS3), rad23Δ::KanMX, dsk2::DSK2-S145–N223del* | This study | Appendix Table S5 |
| sJR2690, *rpn1::RPN1-GFP (HIS3), rad23Δ::KanMX, dsk2::DSK2-L114–F134del, F279–A291del, M301–Q314del* | This study | Appendix Table S5 |
| sJR2692, *rpn1::RPN1-GFP (HIS3), rad23Δ::KanMX, dsk2::DSK2-E246_G247insALFA* | This study | Appendix Table S5 |
| sJR2747, *rpn1::RPN1-GFP (HIS3), rad23Δ::KanMX, dsk2::DSK2-S145–N223del, E246_G247insALFA* | This study | Appendix Table S5 |
| sJR2749, *rpn1::RPN1-GFP (HIS3), rad23Δ::KanMX, dsk2::DSK2-L114–F134del, E246–_G247insALFA, F279–A291del, M301–Q314del* | This study | Appendix Table S5 |
| sJR1323, *scl1::SCL1-mCherry (HYG), dsk2::DSK2-GFP (HIS3)* | This study | Appendix Table S5 |
| sJR2662, *scl1::SCL1-mCherry (HYG), dsk2::DSK2-S145–N223del-GFP (HIS3)* | This study | Appendix Table S5 |
| sJR1123, rpn1::RPN1-GFP (HIS3), dsk2Δ::KanMX | Waite et al, 2024 | Appendix Table S5 |
| sJR1203, rpn1::RPN1-GFP (HIS3), rad23Δ::KanMX dsk2Δ::natMX4 | Waite et al, 2024 | Appendix Table S5 |
| Rosetta 2 (DE3) pLysS | Novagen | 70956 |
| **Recombinant DNA** | | |
| Plasmids for CRISPR-Cas9 in yeast targeting *DSK2* | This study | Appendix Table S6 |
| His-SUMO-Dsk2 in pE-SUMO construct | This study | |
| His-SUMO-Dsk2 UBL | This study | |

| Reagent/resource | Reference or source | Identifier or catalog number |
|---|---|---|
| His-SUMO-Dsk2 UBA | This study | |
| His-SUMO-Dsk2 mutUBA | This study | |
| His-SUMO-Dsk2 ΔSTI1 | This study | |
| His-SUMO-Dsk2 ΔUBL | This study | |
| His-SUMO-Dsk2 Δ3TH | This study | |
| His-SUMO-Dsk2 IDR + STI1 | This study | |
| His-SUMO-Dsk2 STI1 + IDR | This study | |
| pCDB327 | Addgene #113671 | |
| **Antibodies** | | |
| α-GFP | ChromoTek/ Proteintech | 3h9 |
| α-Pgk1 | Invitrogen | 459250 |
| α-K48-linked ubiquitin | Cell Signaling | 8081 |
| α-ALFA (E1R1A) | Cell Signaling | 54963 |
| **Oligonucleotides and other sequence-based reagents** | | |
| Duplex DNA repair fragments for insertion/deletion of regions in *DSK2*. | Integrated DNA Technologies | Appendix Table S6 |
| Primers used for tagging *DSK2* and *SCL1* with C-terminal FP tags | Integrated DNA Technologies | Appendix Table S6 |
| **Chemicals, enzymes, and other reagents** | | |
| Yeast extract | IBI Scientific | IB49161 |
| Peptone | Research Products International | P20250 |
| Dextrose | VWR/Avantor | 0188 |
| Phosphate-buffered saline tablet | Midwest Scientific | KCP32080 |
| Luria-Bertani Broth, Miller | Fisher bioreagents | BP1426-2 |
| Kanamycin | GoldBio | K-120-100 |
| Chloramphenicol | GoldBio | C-105-100 |
| Ammonium chloride | Fisher Chemicals | A661-500 |
| Dextrose | Fisher Chemicals | D16-1 |
| Ammonium chloride ($^{15}$N, 99%) | Cambridge Isotope Laboratories | NLM-467-10 |
| D-Glucose (U-$^{13}$C$_6$, 99%) | Cambridge Isotope Laboratories | CLM-1396-10 |
| Sodium phosphate monobasic | Fisher Chemicals | S397-500 |
| Sodium phosphate dibasic | Fisher Chemicals | S374-3 |
| Sodium chloride | Fisher Chemicals | S271-3 |
| Magnesium chloride | Fisher Chemicals | M33-500 |

| Reagent/resource | Reference or source | Identifier or catalog number |
|---|---|---|
| Imidazole | Fisher Chemicals | O3196-500 |
| EDTA | Sigma-Aldrich | 03609-250 G |
| PMSF | GoldBio | P-470-25 |
| Sodium azide | Fisher bioreagents | BP922I-500 |
| Pierce universal nuclease | Thermo Scientific | 88700 |
| HisPur Ni-NTA resin | Thermo Scientific | 88222 |
| HisPur Cobalt Resin | Thermo Scientific | 89965 |
| Polyethylene glycol 8000 (PEG) | Fisher bioreagents | BP233-1 |
| Urea | Fisher Chemicals | U15-3 |
| Glycerol | Fisher Chemicals | G33-1 |
| Adenosine triphosphate | GoldBio | A-081-25 |
| IPTG | GoldBio | I2481C100 |
| Dithiothreitol | GoldBio | DTT100 |
| Tris(2-carboxyethyl)phosphine, hydrochloride (TCEP-HCl) | GoldBio | TCEP25 |
| Bovine serum albumin (BSA) | Fisher bioreagents | BP1605-100 |
| Proteasome inhibitor cocktail | Valentino et al, 2024 | |
| Deuterium oxide | Sigma-Aldrich | 756822-1 |
| Alexa fluor 647 NHS Ester | Invitrogen | A20006 |
| **Software** | | |
| TopSpin | Bruker | 3.6.4 |
| NMRPipe | Delaglio et al, 1995 | 11.5rev2023.105.21.31 |
| CCPNMR | Vranken et al, 2005 | 2.5.2 |
| NMRBox | Maciejewski et al, 2017 | 2025.42.0 |
| ImageLab | Bio-Rad | 6.1 |
| Fiji | Schindelin et al, 2012 | 1.54 |
| GeneSys | Syngene (Synoptics Ltd.) | 1.6.9.0 |
| GraphPad Prism | GraphPad Software, LLC | 10 |
| BZ-X800 Viewer | Keyence Corporation | |
| NanoDrop2000 | Thermo | 1.6.198 |
| **Other** | | |
| Bruker Avance III 800 MHz | Bruker | |
| NGC 10 Chromatography System | Bio-Rad | |

| Reagent/resource | Reference or source | Identifier or catalog number |
|---|---|---|
| ONI Nanoimager | Oxford Nanoimaging Ltd | |
| Gel Doc EZ Imager | Bio-Rad | |
| G:box Mini | Syngene (Synoptics Ltd.) | |
| BZ-X810 Microscope | Keyence Corporation | |

## Protein expression and purification

Wild-type yeast Dsk2 was cloned from yeast genomic DNA into a pE-SUMO construct using Gibson assembly to create His-SUMO-Dsk2. Different domain deletion and mutant constructs of Dsk2 were prepared from the original plasmid using Phusion Site-Directed Mutagenesis Kit (Thermo Scientific) (Appendix Table S1). All Dsk2 constructs were expressed in *E. coli* Rosetta (DE3) cells in Luria-Bertani broth with 50 mg/L kanamycin and 35 mg/L chloramphenicol grown to $OD_{600}$ of 0.6, induced with 0.5 mM IPTG, and expressed overnight at at 18 °C for 24 h. NMR active ($^{15}N$ and $^{13}C/^{15}N$ labeled) protein samples were expressed in M9 minimal media as detailed elsewhere (Dao et al, 2018). Bacteria were pelleted, frozen, then lysed via freeze/thaw method in 50 mM sodium phosphate buffer pH 8.0 containing 300 mM NaCl, 25 mM imidazole, 0.5 mM EDTA, 1 mM PMSF, 1 mM $MgCl_2$, and 25 U of Pierce universal nuclease. All Dsk2 constructs were purified by $Ni^{2+}$ affinity chromatography. The N-terminal His-SUMO tag was cleaved by SUMO protease at room temperature overnight while dialyzing in a 20 mM sodium phosphate buffer, pH 7.2. To remove His-SUMO tag from the cleaved protein, the cleavage mix was passed through a $Ni^{2+}$ or $Co^{2+}$ column, and the flow-through containing purified protein was collected. To achieve a higher degree of purification, we performed anion exchange chromatography and concentrated all fractions containing purified protein using centrifugal concentrators. Protein purity was estimated by gel electrophoresis (Appendix Fig. S2). Protein concentrations were measured spectroscopically using respective theoretical molar extinction coefficients (Appendix Table S4), except for Dsk2 STI1 + IDR construct, which lacks any Y or W residues. For Dsk2 STI1 + IDR, the concentration was estimated by SDS-PAGE gel using concentration standards of similar molecular weight protein. Purified protein samples were buffer-exchanged into 20 mM sodium phosphate buffer at pH 6.8 containing 0.5 mM EDTA and 0.02% $NaN_3$, and stored at −80 °C.

## SEC-MALS-SAXS data collection and analysis

SAXS was performed at the SIBYLS beamline (beamline 12.3.1 at the Advanced Light Source, Berkeley, CA) with in-line size-exclusion chromatography (SEC) (Rosenberg et al, 2022; Classen et al, 2013; Putnam et al, 2007) to separate sample from aggregates and other contaminants thus ensuring optimal sample quality and multiangle light scattering (MALS) and refractive index measurement (RI) for additional biophysical characterization (SEC-MALS-

SAXS) (see Appendix Table S2). The samples were loaded on a Shodex Protein KW-803 column run by a 1260 series HPLC (Agilent Technologies) with a flow rate of 0.65 mL/min and a temperature of 25 °C. The flow passed through (in order) the UV detector (Agilent 1290 II Diode Array Detector), a MALS detector (18-angle DAWN Helios II, Wyatt Technologies), the SAXS sample cell, and finally an RI detector (Optilab T-rEX, Wyatt). Scattering intensity was recorded using a Pilatus X3 2 M (Dectris) detector which was placed 2.1 m from the sample giving access to a q-range of 0.011 Å$^{-1}$ to 0.47 Å$^{-1}$. 2.0 s exposures were acquired every 2 s during elution (~25 min), and data was reduced using BioXTAS RAW 2.1.1 (Hopkins et al, 2017). Buffer blanks were created by averaging regions flanking the elution peak and subtracted from exposures selected from the elution peak to create the I(q) vs q curves used for subsequent analyses. Peak deconvolution by evolving factor analysis (EFA) (Meisburger et al, 2016) was performed in BioXTAS RAW 2.1.1. Using BioXTAS RAW, Guinier fit and molecular weight analysis were performed (Rambo & Tainer, 2013; Piiadov et al, 2019). Molecular weights were also calculated from the MALS and RI data using the ASTRA 7 software (Wyatt).

## Phase diagram measurements

Protein stock samples were prepared in sodium phosphate buffer (pH 6.8, 20 mM NaPhosphate, 0.5 mM EDTA): 250–1050 μM for Dsk2 FL, Dsk2 ΔUBL, Dsk2 mutUBA, Dsk2 Δ3TH, and Dsk2 ΔSTI1. For the phase-separation assay, 10 μL protein sample from the stock was mixed with 10 μL of 2× phase separation buffer (pH 6.8, 20 mM NaPhosphate, 300 mM NaCl, 15% PEG 8000, 0.5 mM EDTA) and incubated at different/desired temperatures for 20 min, followed by centrifugation at 15,000× g for 5 min at the respective temperature. After centrifugation, 10 μL supernatant was immediately pipetted out without disrupting the pellet and mixed 1:1 in 8 M urea solution to quench any further phase separation due to temperature changes. The concentration of the dilute phase representing the saturation concentration or $c_{sat}$ (1:1 urea diluted supernatant) was measured spectroscopically on NanoDrop One (Thermo Scientific) using respective molar extinction coefficients ($c_{sat}$ = (A280/molar extinction coefficient) *2). Phase separation assays for each Dsk2 construct were performed with two or more protein preparations in triplicate.

## Microscopy

For Dsk2 and Ub-substrate concentration-dependent phase separation study (Fig. 1B), samples were prepared to contain 10 μM Dsk2 (spiked with 20 nM Dsk2 labeled with Alexa Fluor 647) and 0–200 nM of K63- or K48-linked polyubiquitinated substrates (K63-linkage: Rsp5-ubiquitinated R-Neh2Dual-sGFP, K48-linkage: Ubr1-ubiquitinated R-Neh2Dual-sGFP (Valentino et al, 2024)) in 50 mM Tris-Cl pH 7.5 buffer containing 5 mM MgCl$_2$, 5% glycerol, 1 mM ATP, 10 mM creatine phosphate, 0.1 mg/mL creatine phosphokinase, 2 mM DTT, 1 mg/mL BSA, and 1% DMSO, 3% PEG 8000. K63- and K48-linked polyubiquitinated substrates were prepared as described in (Valentino et al, 2024). Substrate was ubiquitinated in a 500 μL reaction, then purified via size-exclusion chromatography on a Superdex S200 column; fractions with the greatest extent of ubiquitination were pooled and concentrated

using centrifugal concentrators. For samples containing proteasome in addition to Dsk2 and polyUb-substrate (Fig. 1C), 100 nM TagRFP-T-Rpn6 containing proteasome (as described in (Valentino et al, 2024)) were added in above condition with 100 μM proteasome inhibitor cocktail (epoxomicin, bortezomib, and MG132). For bright-field imaging of phase separation (Figs. 1D and 5B), 50 μL samples were prepared as described for the phase separation assay from 200 μM protein stocks for all Dsk2 constructs. For Dsk2 Δ3TH and Dsk2 ΔSTI1, additional samples were prepared from 800 μM protein stock.

All samples were added onto Eisco Labs Microscope Slides, with a single Concavity, and covered with 5% BSA-coated coverslips to minimize potential droplet-surface interactions. Sample slides were incubated coverslip-side down at 25 °C for 20 min (or 18 °C for 1 h: Fig. 1B,C) prior to imaging. All samples were imaged on an ONI Nanoimager (Oxford Nanoimaging Ltd) equipped with a Hamamatsu sCMOS ORCA flash 4.0 V3 camera using an Olympus 100X/1.4 N.A. objective. Images were prepared using Fiji (Schindelin et al, 2012) and FigureJ (Mutterer and Zinck, 2013).

## NMR experiments

All NMR experiments were performed at 25 °C on a Bruker Avance III 800 MHz spectrometer equipped with TCI cryoprobe. Proteins were prepared in the NMR buffer: 20 mM sodium phosphate buffer (pH 6.8), 0.5 mM EDTA, 0.02% NaN$_3$, and 5% D$_2$O. All NMR data were processed on NMRBox (Maciejewski et al, 2017) using NMRPipe (Delaglio et al, 1995) and analyzed using CCPNMR 2.5.2 (Vranken et al, 2005).

## NMR spectra

$^1$H–$^{15}$N TROSY-HSQC experiments were acquired using spectral widths of 15 and 27 ppm in the direct $^1$H and indirect $^{15}$N dimensions, and corresponding acquisition times of 200 ms and 46 ms. Centers of frequency axes were ~4.7 and 117.5 ppm for $^1$H and $^{15}$N dimensions, respectively. $^1$H–$^{15}$N TROSY spectra were processed and apodized using a Lorentz-to-Gauss window function with 15 Hz line sharpening and 20 Hz line broadening in the $^1$H dimension, while $^{15}$N dimension was processed using a cosine squared bell function. All spectra were collected with 16 numbers of scans. Contour level and peak intensities were normalized to factor in concentration differences for fair and quantitative comparison across different concentrations wherever applicable. Chemical shift perturbations (CSPs) were quantified as follows: $\Delta\delta = [(\Delta\delta H)^2 + (\Delta\delta N/5)^2]^{1/2}$, where $\Delta\delta H$ and $\Delta\delta N$ are the differences in $^1$H and $^{15}$N chemical shifts, respectively.

## NMR chemical shift assignments

We determined backbone $^1$H, $^{13}$C, $^{15}$N assignments using traditional $^1$H-detect triple-resonance experiments (HN(CA)N, HN(COCA)N, HNCACB, CBCA(CO)NH) on $^{13}$C/$^{15}$N samples containing 250 μM Dsk2 FL or 400 μM Dsk2 UBL or 180 μM Dsk2 UBA or 400 μM Dsk2 ΔSTI1 proteins in pH 6.8 NMR buffer at 25 °C. All experiments used optimized parameter sets incorporating non-uniform sampling.

Assignments for the STI1 domain were obtained using a 100 μM sample of the Dsk2 IDR + STI1 construct (residues 76–223, inclusive of the IDR between the UBL and STI1, and the STI1

domain) with standard triple resonance experiments (HN(CA)N, HNCACB, CBCA(CO)NH, HNCO, and HN(CA)CO). Approximately 88% of backbone amide assignments were transferred by visual inspection to full-length Dsk2 NMR spectra.

Acquisition times for $^1$H-detect experiments were 20 ms, 16 ms, 6 ms, and 75–80 ms, in the indirect $^{15}$N dimensions, indirect $^{13}$CO, indirect $^{13}$Cα/Cβ dimensions, and direct $^1$H dimensions, respectively. Spectral widths were generally 16 ppm in indirect $^{13}$CO (for $^1$H-detect experiments), 28 ppm in indirect $^{15}$N (for $^1$H-detect experiments), and 70 ppm in indirect $^{13}$C$_\alpha$/C$_\beta$. Experiments were acquired with 12%–25% sampling using the Poisson Gap sampling method (Hyberts et al, 2010). Spectra were processed using NMRPipe and employed standard apodization parameters and linear prediction in the indirect dimensions, and analyzed using CCPNMR on NMRBox. Using these experiments, we successfully assigned backbone resonances (H, N, C$_\alpha$, C$_\beta$, CO) for ~84% of all residues.

## Secondary structure determination from NMR data

For C$_\alpha$ and C$_\beta$ secondary shift calculations, random coil chemical shifts for Dsk2 constructs were determined at https://spin.niddk.nih.gov/bax/nmrserver/Poulsen_rc_CS/ using default parameters at 25 °C sample temperature and pH 6.8 (Kjaergaard and Poulsen, 2011). We calculated $\Delta\delta$(C$_\alpha$-C$_\beta$) values (Fig. 2B top) by combining C$_\alpha$ and C$_\beta$ secondary shifts from several Dsk2 constructs (Dsk2 FL, Dsk2 UBL-only, Dsk2 UBA-only, Dsk2 IDR + STI1). The predicted $\Delta\delta$(C$_\alpha$-C$_\beta$) values of Dsk2, presented in Fig. 2B bottom, were obtained using the EFG-CS web server (https://biosig.lab.uq.edu.au/efg_cs/) (Gu et al, 2024). This web server employs machine learning-based models to predict chemical shifts from protein sequences and structures, including experimental structures and AlphaFold2 predictions. For this study, we used the AlphaFold2-predicted structure of Dsk2 (AF-P48510-F1-v4) as input to generate the predicted C$_\alpha$ and C$_\beta$ chemical shifts.

## $^{15}$N relaxation experiments

Longitudinal (R$_1$), transverse (R$_2$) backbone $^{15}$N relaxation rates, and {$^1$H}–$^{15}$N steady-state heteronuclear Overhauser enhancement (hetNOE) were measured for Dsk2 and its domain-deletion variants using established interleaved relaxation experiments and protocols (Castañeda et al, 2016; Hall and Fushman, 2003). Protein concentrations used were either 50 μM or 400 μM. Relaxation inversion recovery periods for R$_1$ experiments were 4 ms (×2), 600 ms (×2), and 1000 ms (×2) for Dsk2 FL and 4 ms (×2), 500 ms (×2), and 800 ms (×2) for Dsk2 ΔSTI1, using an interscan delay of 2.5 s. Total spin-echo durations for R2 experiments were 8 ms (×2), 32 ms, 48 ms, 64 ms (×2), 120 ms, and 160 ms (×2) for Dsk2 FL and Dsk2 ΔSTI1, and 4 ms (×2), 32 ms (×2), 48 ms, 64 ms (×2), 96 ms (×2), and 120 ms (×2) for Dsk2 Δ3TH using an interscan delay of 2.5 s. Heteronuclear NOE experiments were acquired with an interscan delay of 4.5 s. All relaxation experiments were acquired using spectral widths of 12 and 28 ppm in the $^1$H and $^{15}$N dimensions, respectively, with corresponding acquisition times of 90 ms (or 75 ms for 50 μM ΔSTI1 R$_1$) and 28 ms (or 26 ms for 50 μM ΔSTI1 R$_1$). Spectra were processed using squared cosine bell apodization in both $^1$H and $^{15}$N dimensions. Relaxation rates were derived by fitting peak heights to a mono-exponential decay using

RELAXFIT (Fushman et al, 1997). Errors in R$_1$ and R$_2$ were determined using 500 Monte Carlo trials using RELAXFIT on MATLAB. HetNOE values per residue are determined as a ratio of peak heights from corresponding saturation and reference experiments. Errors in hetNOE measurements were determined using the standard error (SE) propagation formula.

## 3D $^1$H–$^{15}$N HSQC-NOESY

Three-dimensional $^1$H–$^{15}$N HSQC-NOESY spectra for 175 μM Dsk2 STI1 + IDR were acquired using the standard Bruker pulse sequence with a mixing time of 120 ms and 16 scans. Spectral widths were set to 16 ppm in the direct $^1$H dimension, 25 ppm in the indirect $^{15}$N dimension, and 12 ppm in the indirect $^1$H dimension. Acquisition times were 12 ms, 8 ms, and 70 ms, in the indirect $^{15}$N dimensions, indirect $^1$H, and direct $^1$H dimensions, respectively.

## CALVADOS molecular dynamics simulations

We investigated the interactions of the transient helical (TH) regions with the STI1 groove by performing single-chain simulations of full-length Dsk2 using CALVADOS3$_{COM}$ (Tesei et al, 2021; Cao et al, 2024; Tesei and Lindorff-Larsen, 2023). In CALVADOS3$_{COM}$, each residue is represented as a single bead placed at the center of mass calculated from all atoms within the residue (Cao et al, 2024). All interactions were assigned as described, and the elastic network model was applied to the folded domains to restrain non-bonded pairs (UBL: residues 1–75, STI1: residues 147–223, and UBA: residues 327–373). Ten full-atom initial conformations of Dsk2 were generated using Modeller (Šali and Blundell, 1993) from the AlphaFold predicted structure (AF-P48510-F1-v4), which were then mapped to the CALVADOS3$_{COM}$ coarse-grained representation. Simulations were run at a temperature of 298.15 K, pH of 6.8, and ionic strength of 0.22 M for 70 ns, where the first 3.5 ns was discarded as equilibration, using Langevin dynamics. The drag coefficient was set to 0.01 ps$^{-1}$, and the timestep was set to 10 fs (Cao et al, 2024).

A second set of simulations was performed for full-length Dsk2 where the UBL and UBA domains were placed in a bound conformation following the crystal structure PDB ID 2BWE (Lowe et al, 2006). To generate a starting structure, the AlphaFold predicted structure was modified in Pymol such that the UBA was moved to an orientation overlapping that of PDB ID 2BWE. The 10 amino acids (residues 317–326) leading to the UBA domain were rebuilt using the Build tool within Pymol, before being energy-minimized using Relax_Amber (Mirdita et al, 2022). Modeller (Šali and Blundell, 1993) was then used to generate ten all-atom initial conformations. To ensure that the UBL and UBA remained bound within the simulations, the elastic bond network was applied to UBL and UBA as a group, and then applied to STI1 separately. Simulations were then performed following the same methodology as the unbound Dsk2 structure.

To quantify the occupancy of the STI1 groove, the amino acids of STI1 were labeled as either interior (being within the hydrophobic groove) or exterior. The center position of the STI1 groove was calculated by averaging the positions of the interior residues. Each IDR residue was then compared to the center position of STI1, and if it was closer to the center position than the

furthest interior residue, the residue was initially considered to be within the groove. For residues marked as within the groove, we further verified occupancy by checking that the residue was closer to an interior residue than an exterior residue. Lastly, to account for the groove's curvature, a principal component analysis was performed on the interior group to identify the dominant axis of the groove, and marked residues as within the groove that were similar to this axis. The presented probabilities are the averages from 10 independent simulations, with each trajectory being the probability of the IDR occupying the STI1 groove within the 133,000 frames of each trajectory. All trajectory data is available at https://zenodo.org/records/15333863, and the analysis code is available at https://github.com/sukeniklab/Dsk2_transient_helices.

For the excluded volume (EV) simulations, all interactions were assigned as described for the full simulations above, and the non-bonded scaling parameter and λ for all amino acids were set to 0. The elastic network model was applied to the folded domains to restrain non-bonded pairs (same as above). Ten full-atom initial conformations of Dsk2 were generated using Modeller from the AlphaFold predicted structure (AF-P48510-F1-v4), which were then mapped to the CALVADOS3$_{COM}$ coarse-grained representation. Simulations were run at a temperature of 298.15 K, pH of 6.8, and ionic strength of 0.22 M. EV simulations were run for 700 ns, where the first 3.5 ns was discarded as equilibration, using Langevin dynamics. Drag coefficient and timestep were set to the same values as noted above.

## AlphaFold2 multimer modeling for predicting interactions between the STI1 domain and transient helices of Dsk2

We used AlphaFold2 Multimer (ColabFold v1.5.5) (Mirdita et al, 2022) to model how the STI1 domain interacts with transient helices in Dsk2. We used the following residues of Dsk2 FL as input: 147-228 as the STI1 domain, 77-146 as TH1 (this is inclusive of the IDR between the UBL domain and the STI1 domain), 229-291 as TH2 (this is inclusive of the IDR between the STI1 domain and TH2 region), and 292-325 as TH3. The exact amino acid sequence for these regions is detailed in Appendix Table S1. Each multimer run resulted in five predicted models ranked 1 to 5. Overlays of all five models from each multimer run are shown in Fig. EV3A. For simplified representation, only one model from each multimer run is shown in Figs. 7C–E and EV3B (rank 2 for TH1; rank 3 for TH2; rank 1 for TH3—each chosen at random). Transient helices in the Dsk2 sequence were compared against human UBQLNs using a multiple sequence alignment generated with M-Coffee (Moretti et al, 2007; Wallace et al, 2006; Di Tommaso et al, 2011).

## Yeast strains and gene manipulations

S. cerevisiae strains used in this work are reported in Appendix Table S5. C-terminal fluorescent tags were introduced at the endogenous locus using standard PCR-based procedures (Waite et al, 2024; Janke et al, 2004; Goldstein and McCusker, 1999). We used a CRISPR-Cas9 plasmid-based approach to delete the Dsk2 STI1 domain, delete the region encoding the three transient helices (3TH), or introduce an internal ALFA Tag between Glu246 and Gly247 of Dsk2 (Laughery et al, 2015; Götzke et al, 2019).

Candidate guide RNAs were determined using the CRISPR/Cas9 target online predictor CCTop (Stemmer et al, 2015; Labuhn et al, 2018). Each guide RNA was cloned into the 2 μ Cas9- and sgRNA-expressing vector pML107, replacing the original gRNA sequence (Addgene plasmid # 67639, (Laughery et al, 2015)). The newly generated plasmids were introduced into yeast together with duplex repair DNA to generate strains with the desired genetic changes (see Appendix Tables S5 and 6 for details). Genotype nomenclature for CRISPR-generated alleles was adapted from (Mucelli & Huang, 2024). Mutations were confirmed by sequencing the genomic region beyond the repair DNA borders, and the CRISPR-Cas9 plasmid was evicted.

## Yeast growth conditions

S. cerevisiae yeast were grown with yeast extract peptone medium with 2% dextrose (YPD). Prior to experimentation, yeast strains were streaked from YPD-glycerol stocks onto YPD agar plates, which were used within 3 weeks. Cells were inoculated from plates into liquid YPD medium and grown overnight at 30 °C with shaking at 220 rpm. Cultures were then diluted to ~0.5–0.6 OD$_{600}$ in fresh YPD media and grown for 3 days (72–77 h). Alternatively, cells were grown for 4 h, sodium azide was added to a final concentration of 0.5 mM, and cells were grown for an additional 24–26 h. After 3 days in YPD, the optical densities were 43–59 OD$_{600}$; after ~25 h in azide, the optical densities were 4.7–8.9 OD$_{600}$.

## Yeast live-cell microscopy

Microscopy preparation and imaging were done at room temperature. In total, 200 μL of yeast culture was diluted with 800 μL phosphate-buffered saline (PBS) to reduce autofluorescence from the YPD. Cells were pelleted by centrifugation (2000×g, 2 min) and resuspended in supernatant (~10 μL after azide treatment or ~50 μL after three days in YPD) to obtain a good cell density for imaging. Three microliters of the resuspended cells were immobilized on a microscopy slide using a 1% agarose-PBS gel pad (1% agarose dissolved in 1× PBS) that was ~0.2-mm thick (modified from https://www.youtube.com/watch?v=ZrZVbFg9NE8, 2019). Images were acquired on a Keyence BZ-X810 microscope using a Nikon Plan Apo 100×/1.40 objective and a 2.8 megapixel monochrome CCD camera. Bright-field images were captured as single-plane images. Rpn1-GFP images of whole cells were captured with the Keyence BZ-X GFP filter cube (Ex 470/40 nm; Em 525/50 nm) using the Quick Full Focus (extended depth of field) setting in the BZ-X800 Viewer application. This setting captures in-focus light in steps of ~0.8 μm into a two-dimensional 8-bit image. Exposure time of the GFP channel was 1.2 s per step with no gamma adjustment. Images of two non-overlapping fields of view at least 400 μm apart were captured within 10 min of dilution of cells in PBS to avoid dissipation of granules.

## Quantification of proteasome puncta

To determine the percentage of cells with proteasome puncta, >200 green-fluorescing cells per biological replicate were manually scored in randomized, blinded images for the absence or presence of at least one Rpn1-GFP punctum. Statistical calculations were

performed in GraphPad Prism. If the Shapiro-Wilk test for normality was passed ($P > 0.05$), datasets were tested for equal variance. Brown–Forsythe and Welch one-way ANOVA with Dunnett's test for multiple comparisons was performed on datasets with unequal variance (determined via Bartlett's statistic). One-way ANOVA with Tukey's test for multiple comparisons was performed on datasets with equal variance determined by Levene's median test. Alpha values were always set to 0.05, and $P$ values less than 0.05 were considered significant.

## Colocalization of Dsk2 and proteasomes

Cells expressing Dsk2-eGFP and proteasome subunit a1-mCherry were used to determine colocalization of Dsk2 and proteasomes. After growth, 5 OD of cells were collected at $2000{\times}g$ at room temperature, incubated in 250 µL of 2% formaldehyde-PBS (pH 7.4) for 4 min, and centrifuged at $2000{\times}g$ at room temperature for 1 min to pellet cells. Fixed cells were resuspended in 250 µL of 1× PBS (pH 7.4), pelleted at $2000{\times}g$, and resuspended in approximately 10 µL of supernatant. Three microliters of this cell suspension were added to 1% agarose-PBS pads on glass slides. Images were acquired at room temperature on a Keyence BZ-X810 microscope using a Nikon Plan Apo 100X/1.40 objective. Single focal plane images were captured with exposure times of 1.5 seconds and no gamma adjustment using the Keyence BZ-X GFP filter cube (Ex 470/40 nm; Em 525/50 nm) and BZ-X TexasRed filter cube (Ex 560/40 nm; Em 630/75 nm).

## Cell lysis and immunoblotting analysis

The equivalent of 2 $OD_{600}$ cell amount was harvested by centrifugation and immediately lysed via alkaline lysis into sample buffer and frozen at $-80\,°C$ as reported previously (Kushnirov, 2000). Cells were pelleted at $17,000{\times}g$ and resuspended in 100 µL deionized water. Following the addition of 100 µL 200 mM NaOH, samples were incubated at room temperature for 5 min and centrifuged for 1 min at $17,000{\times}g$. Pellets were resuspended in 50 µL SDS-PAGE sample buffer (0.06 M Tris–HCl, pH 6.8, 5% glycerol, 2% SDS, 4% β-mercaptoethanol, 0.0025% bromophenol blue), boiled at 98 °C for 5 min, centrifuged for 1 min at $17,000{\times}g$, and the supernatant was collected and frozen at $-80\,°C$. Equal volumes of each lysate was separated by SDS-PAGE, transferred to PVDF membranes, and immunoblotted using antibodies against GFP (Chromotek, catalog no. 3h9, 1:1500), Pgk1 (Invitrogen, catalog no. 459250, 1:10,000), K48-linked polyubiquitin (Cell Signaling Technology, catalog no. 8081, 1:1000), or ALFA Tag (Cell Signaling Technology, catalog no. 8081, 1:1000) and HRP-conjugated secondary antibodies. Images were acquired with a G:Box imaging system with a 6.0 megapixel camera and GeneSys software (Syngene). Densitometry analyses were performed with the 16-bit TIFF images using the Gels tool in Fiji version 1.54 with manual background correction (Schindelin et al 2012). Plots were generated in GraphPad Prism.

## Data availability

The datasets and computer code produced in this study are available in the following databases: SAXS data: SASBDB - Full-length Dsk2 Accession SASDYT2. NMR chemical shift assignments: BMRB - Full-length Dsk2 BMRB ID 53439 https://bmrb.io/data_library/summary/?bmrbId=53439. All raw data for figures, including imaging data: Biostudies - https://www.ebi.ac.uk/biostudies/studies/S-BSST2319. CALVADOS simulation trajectory data: Zenodo (https://zenodo.org/records/15333863). Simulation analysis code: Github (https://github.com/sukeniklab/Dsk2_transient_helices).

The source data of this paper are collected in the following database record: biostudies:S-SCDT-10_1038-S44318-026-00696-1. This database record also lists individual authorship for specific figure panels/source data where available.

## Peer review information

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

## Acknowledgements

We acknowledge support from National Institute of Health (NIH) R01 GM136946 and R35 GM158070 to CAC, R35 GM149314 to JR, R35 GM137926 to SS, NSF grant MCB1935596 to DAK, postdoctoral support to NA and JKN from the Syracuse University Vice President of Research office, and a graduate fellowship to EAD from the Madison and Lila Self Graduate Programs at the University of Kansas. SS acknowledges support from the Alfred P. Sloan Foundation. Support for the Bruker 800 MHz spectrometer with TCI cryoprobe was provided by shared instrumentation NIH Grant 1S10OD012254. We appreciate support from Dr. Charlie Fry at the SUNY-ESF NMR facility. Additional NMR experiments were performed at the Johns Hopkins Biomolecular NMR facility with support from Dr. Ananya Majumdar. We thank Dr. Matthew Wohlever for his important feedback on our work. We thank Dr. Maxwell Watkins for the SEC-MALS-SAXS experiments performed at BioCAT/SIBYLS facility. This research used resources of the Advanced Light Source (ALS), a national user facility operated by Lawrence Berkeley National Laboratory on behalf of the Department of Energy (DOE), Office of Basic Energy Sciences, through the Integrated Diffraction Analysis Technologies (IDAT) program, supported by DOE Office of Biological and Environmental Research. Additional support comes from the NIH project ALS-ENABLE (P30 GM124169) and a High-End Instrumentation Grant S10OD018483. This research also used resources of the Advanced Photon Source, a U.S. DOE Office of Science User Facility operated for the DOE Office of Science by Argonne National Laboratory under Contract No. DE-AC02-06CH11357. BioCAT was supported by grant P30 GM138395 from the National Institute of General Medical Sciences (NIGMS) of the NIH. We thank Syracuse University's High Performance Computing cluster, OrangeZest, for computational resources and Daniel Jeski for help in running the molecular dynamics simulations. The content is solely the responsibility of the authors and does not necessarily reflect the official views of NIGMS or the NIH.

## Author contributions

**Nirbhik Acharya**: Conceptualization; Data curation; Software; Formal analysis; Supervision; Validation; Investigation; Visualization; Methodology; Writing—original draft; Writing—review and editing. **Emily A Daniel**: Conceptualization; Data curation; Formal analysis; Validation; Investigation; Visualization; Methodology; Writing—review and editing. **Thuy P Dao**: Formal analysis; Validation; Investigation; Visualization; Writing—review and editing. **Jessica K Niblo**: Resources; Data curation; Software; Formal analysis; Validation; Investigation; Visualization; Methodology; Writing—review and editing. **Erin O Mulvey**: Resources; Investigation. **Shahar Sukenik**: Data curation; Formal analysis; Supervision; Investigation; Methodology; Project administration; Writing—review and editing. **Daniel A Kraut**: Resources; Supervision; Investigation; Project administration; Writing—review and editing. **Jeroen Roelofs**: Data curation; Formal analysis; Supervision; Investigation; Methodology; Project administration; Writing—review and editing. **Carlos A Castañeda**: Conceptualization; Data curation; Formal analysis; Supervision; Funding acquisition; Investigation; Methodology; Writing—original draft; Project administration; Writing—review and editing.

Source data underlying figure panels in this paper may have individual authorship assigned. Where available, figure panel/source data authorship is listed in the following database record: biostudies:S-SCDT-10_1038-S44318-026-00696-1.

## Disclosure and competing interests statement

The authors declare no competing interests.

# Expanded View Figures

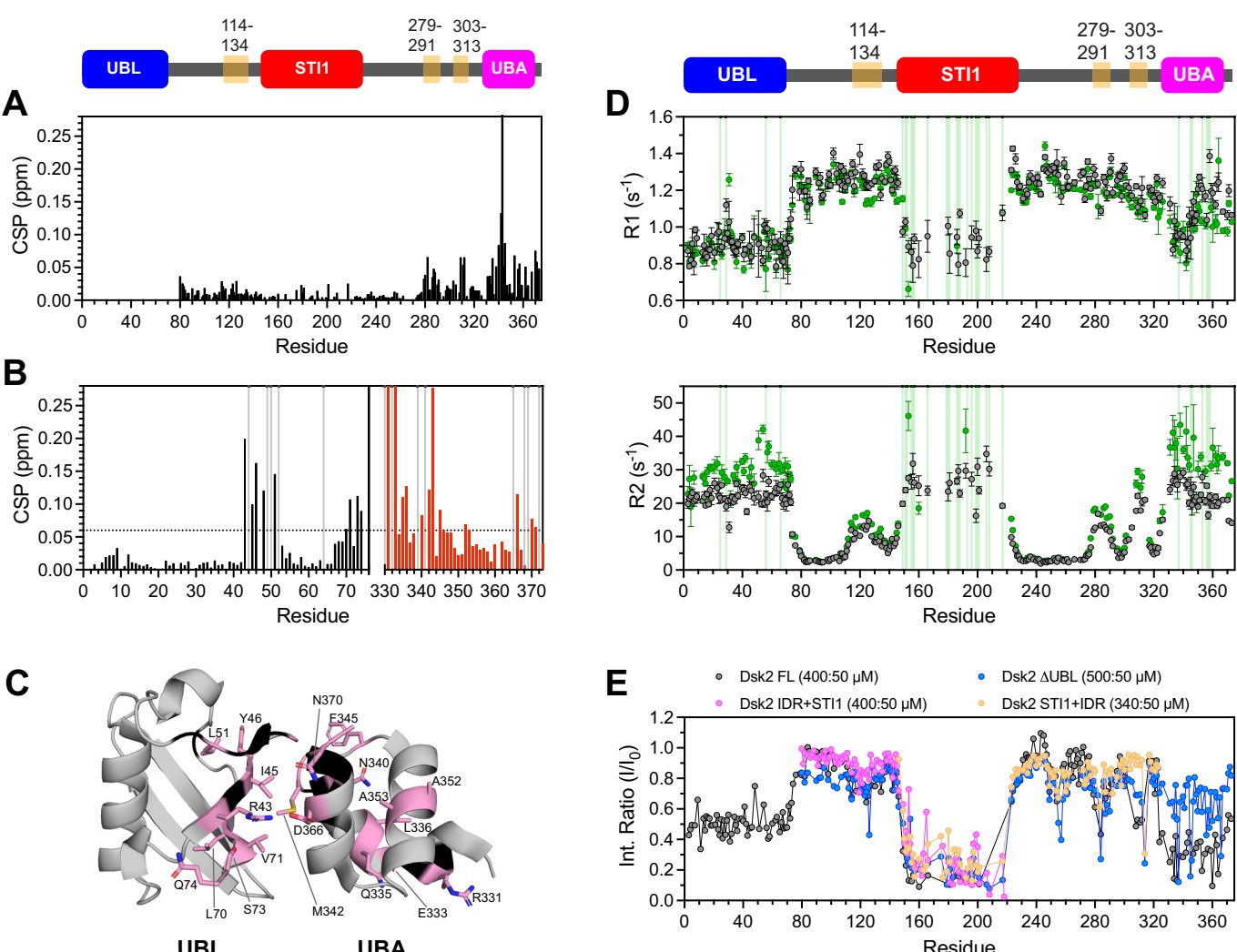

**Figure EV1.  Identification of UBL-UBA interactions and STI1 concentration-dependent interactions in Dsk2.**

(**A**) CSPs of backbone amide resonances between Dsk2 FL and Dsk2 ΔUBL show large CSPs for residues in the UBA domain of Dsk2 indicative of UBL-UBA interactions. (**B**) CSPs are shown between Dsk2 UBL-only and Dsk2 FL (black bars), and between Dsk2 UBA-only and Dsk2 FL (red bars). Gray bars represent residues for which no amide peaks were observed (or unassigned) for Dsk2 FL. (**C**) Residues with CSPs > 0.06 ppm (above dotted black line in panel B) are highlighted as pink sticks that map to the UBL-UBA interface as shown on the crystal structure of the bound form of isolated Dsk2 UBL and UBA domains (PDB: 2BWE). (**D**) $^{15}$N $R_1$ and $R_2$ relaxation rates are compared for Dsk2 at 50 μM (gray) and 400 μM (green); $n = 1$. $R_1$ and $R_2$ values are best model parameter from fit and error bars in relaxation rates are standard deviation from 500 Monte Carlo trials using RELAXFIT (see "Methods"). The concentration-dependent increase in $R_2$ relaxation rates (and corresponding decrease in $R_1$ relaxation rates) for resonances in the UBL and UBA domains suggest increased UBL-UBA intermolecular interactions with increased protein concentration. Green bars represent resonances for which there is no observable amide resonance at 400 μM; primarily affected are resonances corresponding to the STI1 domain, suggestive of intermolecular interactions involving the STI1 domain. (**E**) Concentration-dependent intensity ratios ($I/I_0$) of amide resonances are plotted between high ($I$) and low ($I_0$) protein concentrations of STI1 domain-containing Dsk2 variants (Appendix Table S1). Concentrations are noted in the legend above plot; intensity ratio is corrected for differences in protein concentration ("Methods"). Notably, residues within the STI1 domain of all Dsk2 constructs exhibit a similar decrease in peak intensities at higher protein concentration indicative of concentration-dependent STI1-STI1 interactions.

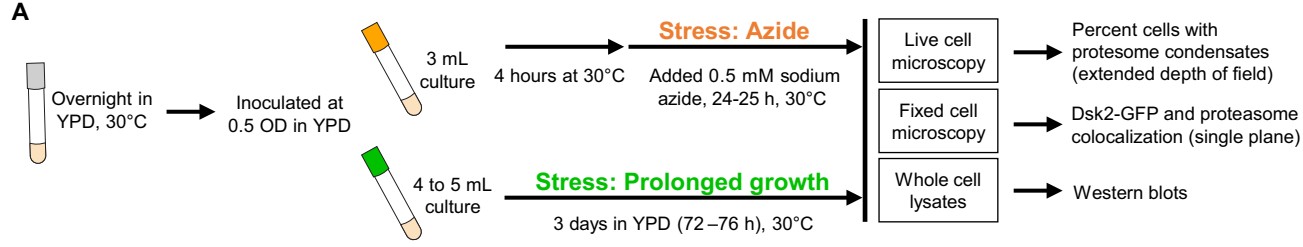

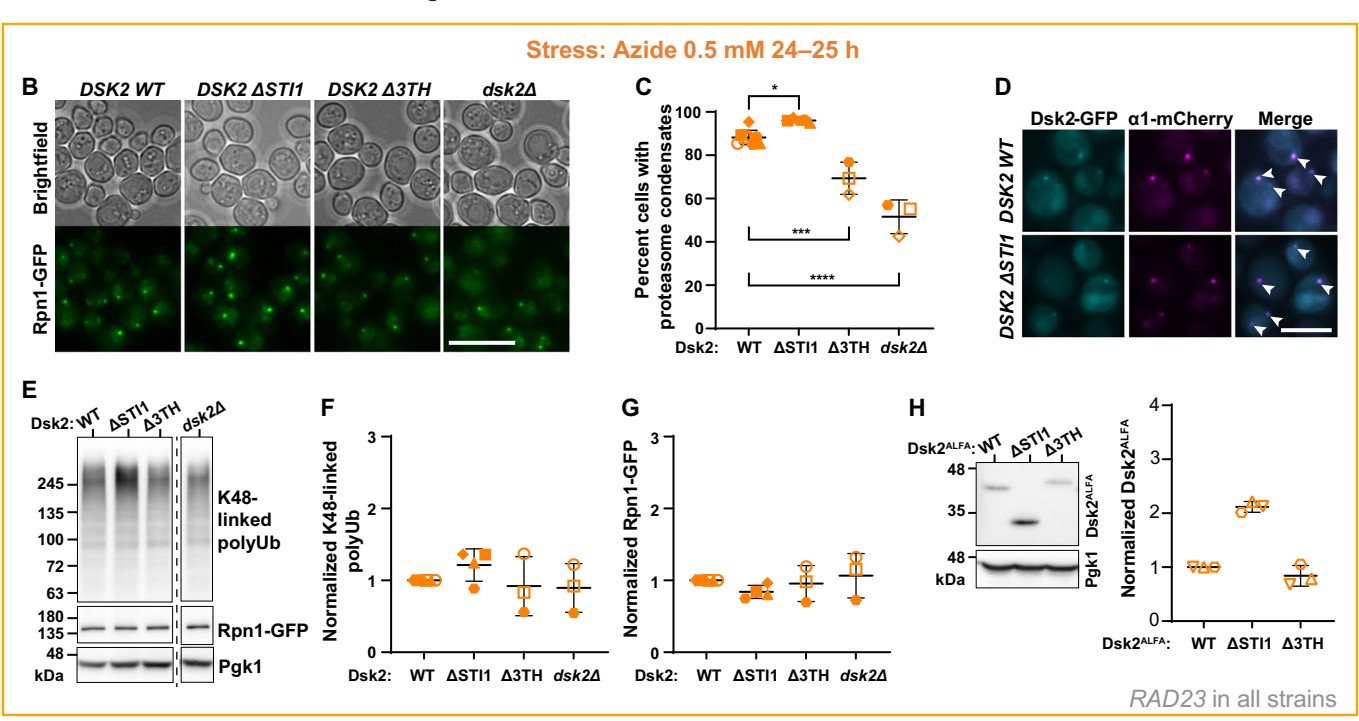

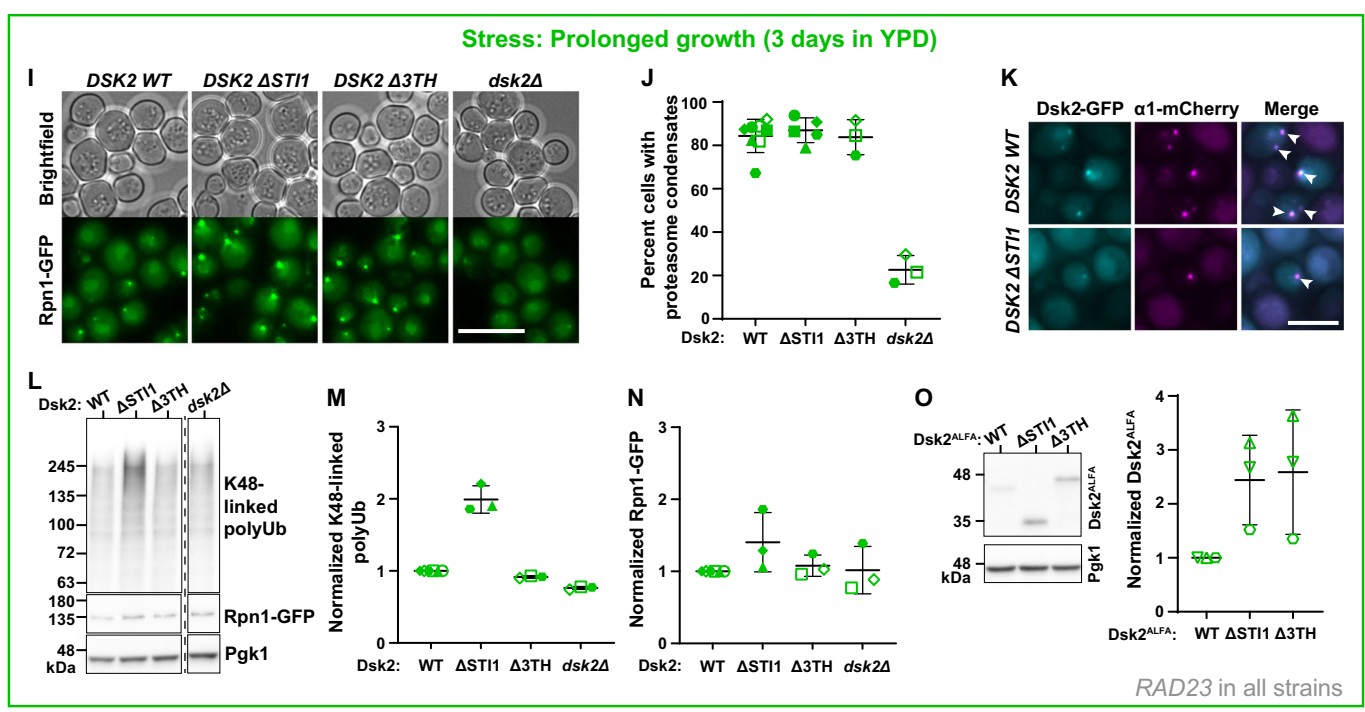

◄

**Figure EV2. Presence of shuttle factor Rad23 diminishes the impact of Dsk2 STI1 domain and 3TH deletions on proteasome condensate formation.**

(A) Workflow schematic for stress-induced proteasome condensate induction in *S. cerevisiae* and cell analysis methods. (B–O) All yeast strains (in the endogenous *RAD23* background) were subjected to (B–H) ~ 24 h azide stress or (I–O) prolonged growth stress (3 days in YPD). (B, I) Representative bright-field and extended-depth-of-field epifluorescence images (GFP channel) showing proteasome puncta (endogenous Rpn1-GFP) in strains expressing Dsk2 variants after (B) ~ 24 h azide stress or (I) prolonged growth stress. Scale bar: 10 μm. (C, J) Percentage of cells with ≥1 punctum after stress. Statistics for (C) are *$P = 0.0393$, ***$P = 0.0001$, ****$P = 0.00000003$; one-way ANOVA with Tukey's test ($\alpha = 0.05$), $n \geq 3$. (D, K) Single-plane epifluorescence images GFP-tagged endogenous Dsk (Dsk2-GFP: pseudocolored cyan) and mCherry-tagged α1 subunit of proteasome (α1-mCherry: pseudocolored magenta). Arrows indicate colocalization of Dsk2 with proteasome condensates in Dsk2 WT and Dsk2 ΔSTI1 cells. Scale bar: 5 μm. (E–G) Representative immunoblot of whole-cell lysates from Dsk2 variant strains after ~24 h azide stress showing (F) K48-linked polyubiquitin and (G) Rpn1-GFP levels, normalized to Pgk1 and scaled to Dsk2 WT. $n \geq 3$. (H) Representative immunoblot of whole-cell lysates after ~24 h azide stress (left), showing relative levels of endogenous Dsk2[ALFA] variants normalized to Pgk1 and scaled to Dsk2[ALFA] WT (right). $n = 3$. (L–O) Immunoblots as in (E–H) for prolonged growth stress. Each biological replicate is denoted by a single symbol wherever applicable. On plots, horizontal line and error bars represent mean and SD, respectively. Full blots in Appendix Figs. S8 and S9.

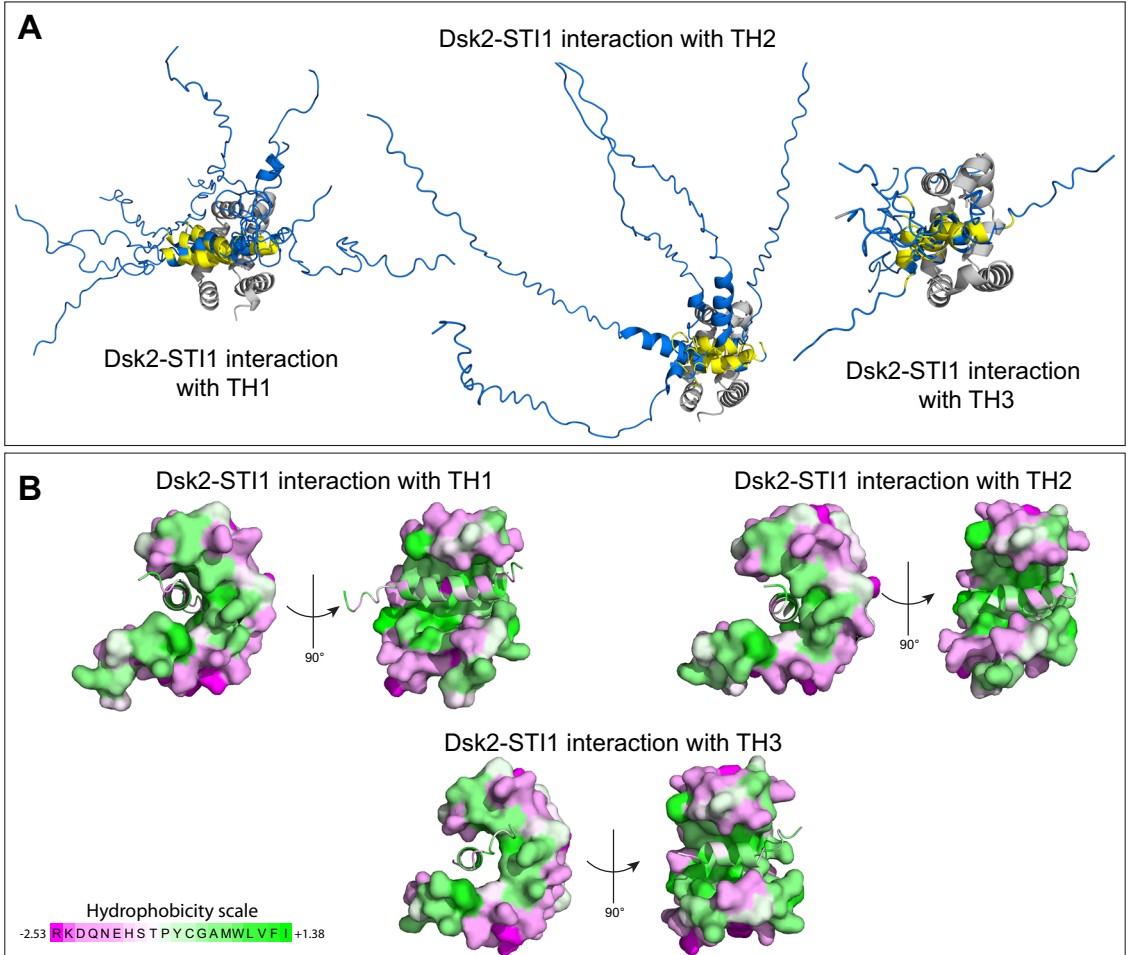

**Figure EV3. Hydrophobic interactions between amphipathic helices and STI1 domain.**

(A) Overlays of all five predicted models of AlphaFold2 multimer runs are shown for STI1-TH1, STI1-TH2, and STI1-TH3 (STI1 domain residues 147-228, TH1 region: residues 77-146, TH2 region: residues 229-291, TH3 region: residues 292-325). STI1 domain is colored gray; all TH regions (inclusive of IDRs) colored blue; TH residues with CSP > 0.04 ppm between Dsk2 FL and ΔSTI1 (from Fig. 4B) are highlighted yellow. (B) Hydrophobic interactions between the STI1 domain (surface representation) and transient helices TH1, TH2, and TH3 (cartoon representation) in Dsk2 using predicted AlphaFold models (same views of structures presented in Fig. 7). All amino acid residues in (B) are colored based on their hydrophobicity (Eisenberg et al, 1984) using Pymol.

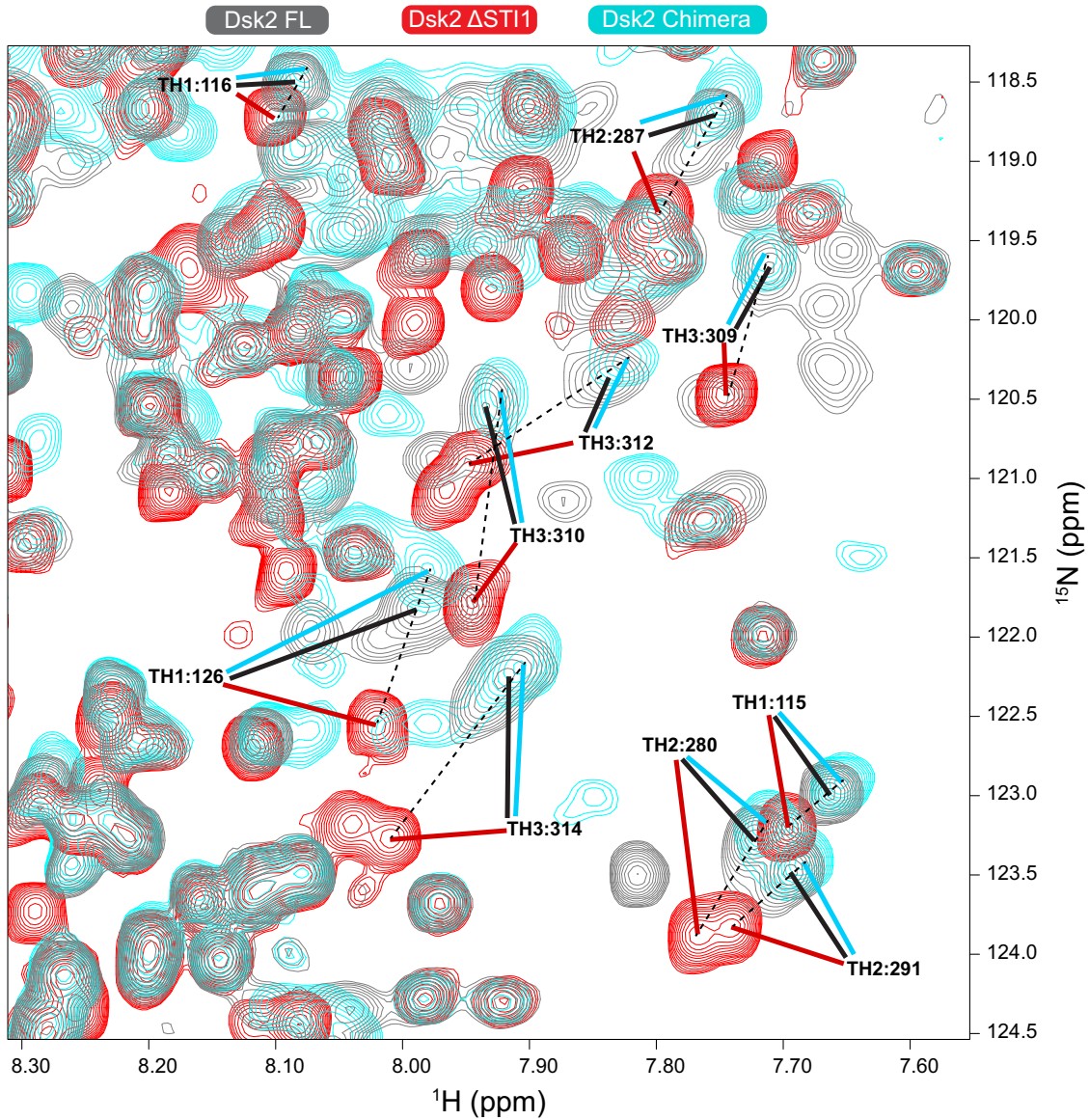

**Figure EV4.    Transient helices (THs) of Dsk2 interact with the UBQLN2 STI1-II domain.**

Overlay of $^1$H–$^{15}$N TROSY-HSQC spectra of Dsk2 FL, Dsk2 ΔSTI1, and Dsk2-Chimera (Dsk2 where STI1 is replaced with residues 379–462 (STI1-II) of UBQLN2) collected under identical conditions. Note that the backbone amide positions for Dsk2-Chimera (cyan) resonances in transient helices (annotated with "TH") nearly overlap with Dsk2 FL (gray) positions. Dotted lines are used to show trajectories of resonances across the three protein samples. The ability of UBQLN2 STI1-II domain to engage with transient helices within the Dsk2-Chimera molecule supports a general mechanism of STI1-helix interactions within UBQLNs and other STI1-containing co-chaperones.

