## [Peer Review File · The EMBO Journal]

STI1 domain engages transient helices to mediate Dsk2 phase separation and proteasome condensation

Nirbhik Acharya, Emily Daniel, Thuy Dao, Jessica Niblo, Erin Mulvey, Shahar Sukenik, Daniel Kraut, Jeroen Roelofs, and Carlos Castañeda

Corresponding author(s): Carlos Castañeda (cacastan@syr.edu)

Review Timeline:

Submission Date:	28th May 25
Editorial Decision:	8th Jul 25
Revision Received:	3rd Dec 25
Editorial Decision:	19th Dec 25
Revision Received:	29th Dec 25
Accepted:	12th Jan 26

Editor: Hartmut Vodermaier

Transaction Report:

Prof. Carlos A Castañeda
Syracuse University
Biology and Chemistry

8th Jul 2025

Re: EMBOJ-2025-121482
ST11 domain dynamically binds transient helices in disordered regions to drive Dsk2 phase separation

Dear Carlos,

Thank you again for submitting your study on Dsk2 self-association via ST11 domain interactions to The EMBO Journal. It has now been seen by three expert referees, whose comments are copied below. As you will see, the referees acknowledge the technical quality of the work and appreciate the findings as a valuable addition to the ubiquitin field. At the same time, they however also raise a number of well-taken concerns, which would need to be adequately addressed prior to publication. In particular, they all feel that the *in vivo* analyses on proteasome punctae formation would need to be strengthened and better connected to the main part of the study.

Should you be able to address this as well as the various other conceptual and technical points brought up by the referees - which also include some follow-up experimentation- then we would be interested in pursuing a revised version further for EMBO Journal publication. I should however point out that it is our policy to allow only one single round of major revision, making it important to diligently respond to each of referee points at the time of resubmission. Therefore, I would encourage you to contact me with a revision plan and preliminary point-by-point response already during the early stages of your revision work, so that we could discuss if and how the main points could be resolved; or whether a less comprehensively revised manuscript might alternatively become suitable for publication in one of our sister journals like EMBO Reports or Life Science Alliance. We would also be open to extension of the default three-months revision period if needed; our 'scooping protection' (meaning that competing work appearing elsewhere in the meantime will not affect our considerations of your study) would of course remain valid throughout the whole period.

Detailed information on preparing, formatting and uploading a revised manuscript can be found below and in our Guide to Authors, and adhering to them as closely as possible shall greatly facilitate editorial processing upon resubmission. Thank you again for the opportunity to consider this work for The EMBO Journal, and I look forward to hearing from you in due time.

With kind regards,

Hartmut

- 3) Revised manuscript text (including main tables, and figure legends for main and EV figures) has to be submitted as editable text file (e.g., .docx format). We encourage highlighting of changes (e.g., via text color) for the referees' reference.
- 4) Each main and each Expanded View (EV) figure should be uploaded as individual production-quality files (preferably in .eps, .tif, .jpg formats). For suggestions on figure preparation/layout, please refer to our Figure Preparation Guidelines: <http://bit.ly/EMBOPressFigurePreparationGuideline>
- 5) Point-by-point response letters should include the original referee comments in full together with your detailed responses to them (and to specific editor requests if applicable), and also be uploaded as editable (e.g., .docx) text files.
- 6) Please complete our Author Checklist, and make sure that information entered into the checklist is also reflected in the manuscript; the checklist will be available to readers as part of the Review Process File. A download link is found at the top of our Guide to Authors: embopress.org/page/journal/14602075/authorguide
- 7) All authors listed as (co-)corresponding need to deposit, in their respective author profiles in our submission system, a unique ORCID identifier linked to their name. Please see our Guide to Authors for detailed instructions.
- 8) Please note that supplementary information at EMBO Press has been superseded by the 'Expanded View' for inclusion of additional figures, tables, movies or datasets; with up to five EV Figures being typeset and directly accessible in the HTML version of the article. For details and guidance, please refer to: embopress.org/page/journal/14602075/authorguide#expandedview
- 9) To facilitate reproducibility and cross-laboratory adoption of methodologies, please structure the Materials & Methods section as outlined in our guide to authors, including a completed Reagents and Tools Table that can be downloaded from our author guidelines as well (<https://www.embopress.org/page/journal/14602075/authorguide#structuredmethods>).
- 10) Digital image enhancement is acceptable practice, as long as it accurately represents the original data and conforms to community standards. If a figure has been subjected to significant electronic manipulation, this must be clearly noted in the figure legend and/or the 'Materials and Methods' section. The editors reserve the right to request original versions of figures and the original images that were used to assemble the figure. Finally, we generally encourage uploading of numerical as well as gel/blot image source data; for details see: embopress.org/page/journal/14602075/authorguide#sourcedata

Revision to The EMBO Journal should be submitted online within 90 days, unless an extension has been requested and approved by the editor; please click on the link below to submit the revision online before 6th Oct 2025:
Link Not Available

If you choose to alternatively have this study further considered by another EMBO Press publication, please use the following hyperlink to directly transfer the manuscript, optionally with inclusion of referee reports and identities:
Link Not Available

Referee #1:

In this manuscript, Archarya et al. show that the yeast ubiquitin-receptor shuttle protein Dsk2 self-associates to form condensates in vitro through complex STI1 interactions with itself and newly identified transient helices in intrinsically disordered regions (IDRs) of the protein. The researchers used a combination of nuclear magnetic resonance (NMR) spectroscopy, phase-separation microscopy, and computation simulations on full-length Dsk2 to examine these interactions. Through this, three regions inside of Dsk2's IDRs were identified and determined to have propensity to form helices with transient nature, called 3TH. Interactions between STI1 and 3TH were investigated deeply using mutational analysis and deletion constructs. In alignment with 3TH's suggested role in self-association, after STI1 deletion, 3TH amide resonances and CSPs experienced large changes, which was further supported by molecular dynamics simulations. Lastly, researchers examined proteasome puncta formation in yeast cells with deletion of the STI1 domain. Overall, this manuscript helps to clarify a major question in the UBQLN field surrounding the role of the STI1 domain in binding to itself and/or co-chaperones or client proteins and reveal the identity and importance of these unique transient helices in facilitating self-interaction and phase separation. A major benefit of this study is the complementary nature of the NMR experiments and phase separation analysis, which converge on the

importance of 3TH:STI1 interactions to facilitate self-association. A weakness of the study is the lack of connection between the themes of Figure 6 and the rest of the study, which could be strengthened. Overall this is a very valuable addition to the UBQLN field and raises quite a few interesting questions that will assuredly inspire additional study from this group and others.

Major Comments:

1. Figures 1 and 6 need a stronger connection to the rest of the paper. They are clearly valuable data, and bring an interesting angle to the manuscript by highlighting how protein accumulation and condensation can be differentially affected by Dsk2 mutation, but they are disconnected from Figures 2-5. For example, this could be strengthened by looking into how deletion of 3TH influences proteasome-containing condensates in yeast. In addition, the Rad23 requirement needs some additional explanation or experimental exploration, as it is unknown why the Rad23 null background is required for strong puncta phenotypes?
2. Why does the minimal system in Figure 1 use a K63-linked substrate while Figure 6 examines K48-linked ubiquitin? It would be nice to add a K48-linked substrate to Figure 1 or blot for K63-linked ubiquitin in Figure 6.
3. The use of a ubiquitinated protein to potentiate Dsk2 phase separation is a bit confusing considering Dao et al 2018 shows that the addition of ubiquitin causes a dissipation of condensates of UBQLN2. Further, given the focus of the rest of the manuscript on STI1 domain interactions, one wonders whether the same effect would occur for a non-ubiquitinated protein known to bind UBQLNs, such as ATP5G, Omp25, or others.
4. For the CALVADOS simulations, it seems possible that TH1 has a higher probability of interacting with STI1 due to its proximity to the STI1 domain (Figure 4E). For example the probability appears high for non TH-sequences at +10 to the STI1 domain as well. Is it possible to perform simulated mutagenesis to parse this out? For example, what would happen if the TH sequences were scrambled among each other, or moved in increments of ~20 AA?

Minor Comments:

- The introduction may benefit from some additional introduction of proteasome granules to tie in Figures 1 and 6.
- Can the authors comment on the apparent 'shell-like' pattern of Proteasome staining in the condensates of Figure 1c?
- It would be helpful for the reader to have some simple way of discriminating predicted structural data compared to NMR data, for example in Figure 1B top versus bottom or Figure 1D the bottom pLDDT from AlphaFold. Could these predicted data plots be colored differently or otherwise distinguished more clearly?
- Figure 4f is not referenced in the text and it is unclear what it is meant to convey
- It would be nice to include a sequence alignment of 3TH from Dsk2 and human UBQLNs, perhaps as an addition to Figure S1. For that matter, the Discussion on page 21 implies that there may be TH sequences from other proteins that might work together with Dsk2/UBQLNs - it would be very interesting to discuss this more.
- The Discussion's third paragraph on page 22 is rather confusing
- For someone outside of the protein NMR field, phrasing and general descriptions of methods and results could be strengthened to enable a more concise takeaway for a less familiar audience.
- Rajendran and Castañeda reference is cut off and missing some detail

Referee #2:

Summary and significance.

Dsk2 is a ubiquitin shuttle factor that delivers ubiquitinated proteins to the proteasome for degradation. Dsk2 consists of three domains - a UBA, STI1, and UBL domain. Through its UBA domain, Dsk2 binds to polyubiquitinated substrates and shuttles these substrates to the proteasome via an interaction with its UBL domain. Loss of Dsk2 can lead to the accumulation of protein aggregates, upsetting cellular homeostasis, and harming the cell. The STI1 domain is the least structured of these domains and is thought to be an intrinsically disordered domain that becomes structured when bound to substrate. Other proteins containing STI1 domains include the co-chaperones SGTA, Hop, and Hip as well as ubiquitin associating proteins like UBQLNs. Acharya et al reveals that there are intrinsically disordered transient helices between the UBA and STI1 domain and the STI1 domain and UBL that interact with STI1. Additionally, the authors show in vitro that STI1 is the main driver of Dsk2 self-association and phase separation. In vivo, Dsk2 lacking STI1 in cells lacking Rad23 promotes the formation of proteasome condensates.

The authors present structural studies with various Dsk2 mutants to determine the regions responsible for phase separation. Generally, the analyses are interesting and enhance our understanding of how Dsk2 self-assembles and phase separates.

Concerns:

1. When investigating the STI1 interactions with transient helices, the authors generate Δ STI1 Dsk2 mutants. I wonder if removing a nearly 100aa section that adopts a structured conformation and is necessary for many Dsk2 functions could affect the stability of the protein. Though the authors address this with "The STI1 deletion-induced changes in 3TH backbone dynamics could result from: a) removal of STI1-3TH interactions, or b) destabilization of the secondary structure of these transient helices. However, the latter is unlikely as the measured hetNOE values (Figure 4C) and α -helix propensity (Figure 4D) for the 3TH regions in Δ STI1 remain similar when compared with data for Dsk2 FL." The argument could be strengthened by either mutating

the STI1 domain, perhaps by disrupting the hydrophobic groove formed by the Dsk2-STI1. The authors could also investigate the specificity of this interaction by replacing Dks2-STI1 with the STI1 domain from another protein like the UBQLNS, HIP, Hop, or SGTA.

2. This builds off the first point. The *in vivo* work is difficult to contextualize. Again, I have the same concerns about making a truncated complex and placing the three transient helices so much closer to each other in sequence. In WT cells, nearly all cells have at least one puncta. There is a stark decrease in the number of cells with puncta in the Δ Rad23 cells, and in cells lacking Rad23 and Dsk2- Δ STI1 there is an increase in puncta. Is Rad23 the necessary factor for proteasome puncta formation? And in the absence of Rad23 only Dsk2 without STI1 forms puncta? How does this fit with the *in vitro* work that highlights Dsk2-STI1 as the major contributor to self-association and phase separation.

Perhaps the authors can address the possibility that without its STI1 domain Dsk2 cannot perform its chaperone activity by replacing the STI1 domain with the STI1-domain of another STI1-domain containing protein.

3. The authors decision to threshold the intensity of the puncta in Fig 6A and B is not convincing. Particularly in Panel B the Dsk1 Δ STI1 construct the lower left cell looks like it has two puncta, but these puncta apparently don't meet the threshold cut-off. Perhaps if the authors set a threshold that is normalized to the intensity of each individual cell.

Major comments:

1. Throughout the Introduction the authors use the terms "STI1 domain" and "STI1-like domain". When describing the domain organization of Dsk2, the authors state that it consists of a STI1-like domain. However, Fry et al, 2021 identifies the domain within Dsk2 as a STI1-domain and the authors refer to this domain in the rest of their text as a STI1 domain.

2. The authors should introduce the three transient helices in the introduction section. This will make it clearer to the reader that the helix in the helix/STI1 interactions they observe and discussed at the end of the introduction is part of Dsk2 and not a substrate.

3. The authors discuss the concentration dependent phase separation of Dsk2 - referring to the physiological levels of Dsk2 would put their observations into perspective - especially in the discussion section.

4. Figure 5 A & B are difficult to compare. Δ 3TH appears to have a similar csat at 5C to that of Δ UBL and mutUBA, but it's difficult to see the differences between these mutants at low temperatures. Combining these plots would allow for easier comparison by the reader.

5. Figure 7 would be easier to interpret if the authors colored the helices bound to Dsk2-STI1 domain by hydrophobicity to illustrate the amphipathicity of these helices.

Minor comments:

1. In the abstract: "In vivo, STI1 removal results in an increase of proteasome-containing...". In vivo should be italicized.

2. First paragraph of the results section: "Using fluorescence microscopy," The authors do not mention what fluorophore they used or what fluorophore is attached to which component in their minimal *in vitro* system.

3. The order for Fig1 panel B is not intuitive. I suggest moving the second column to the left (200nM Substrate and 0uM Dsk2) and the left most panel to the second column. This way it will read a condition where there is no Dsk2 and only substrate and then an increasing concentration of substrate with constant Dsk2 concentration.

4. Page 21 second paragraph "Functionally, STI1 domains of co-chaperone like proteins" - these proteins are co-chaperones, not co-chaperone like.

5. Page 21: In Lin et al 2021, when determining the minimum substrate requirement for Sgt2 the authors generated artificial substrates using a mixture of leucines and alanines to control for hydrophobicity and that the identity of the residues isn't what is important.

Referee #3:

This manuscript elegantly uses NMR and biochemical assays to characterize Dsk2 intramolecular and intermolecular interactions. Some strengths are quality of the NMR experiments, with some gems including the strongly correlated experimental and predicted α values and correlation of AlphaFold pLDDT scores with NMR relaxation data - the latter might not have been shown before. The concentration-dependent data are convincing of STI1-dependent self-association and the CSP values with the STI1 deleted illustrate effects on the three identified interacting regions. These interactions are also convincingly shown to contribute to Dsk2 phase separation. The authors further find increased proteasome puncta following STI1 deletion and propose that this is an indirect effect driven by accumulation of K48-Ub proteins. This later part of the paper is less convincing, and it is not clear why loss of the STI1 domain would directly drive K48-Ub accumulation - added to this concern is the lack of an antibody to validate the presence of the edited protein. Some additional questions/suggestions are listed below.

Figure 2A: The UBL domain peaks should be labeled to make it easier to relate back to the text.

A table with the Dsk2 chemical shift assignments should be included.

Figure 3A: For this experiment and for all the NMR experiments it'd be worth listing the number of scans taken. The signals appear in the figure quite strong. Was a NOESY experiment attempted? Even amide-amide NOEs could help establish helicity in

the regions of interest.

Figure 4B: This data is convincing of changes in the three regions following STI1 deletion. It might be worth also adding in STI1 and examining whether the signals return to their state in the full length protein. Another experiment that could be done is to add in spin labeled STI1 and look for resonance broadening.

Figure 4G: This image is not a good representation of the movie provided. To summarize the simulations, distances over time should be included and average values with deviations. Grey regions also enter the red STI1 cavity and this should be represented in some way. These analyses should be provided for both simulations (with and without UBL/UBA interaction). It's not clear why deltaSTI1 would lead to increased K48-Ub in the lysates. In the study limitations section the authors note a lack of anti-Dsk2 antibodies. This is a sizeable limitation - can mass spectrometry be used instead to discern whether the Dsk2 protein levels have changed and the loss of the STI1 domain? How do these effects compare to a Dsk2 knockout? Could it be that deltaSTI1 causes loss of function or low protein levels? Assuming they are at equivalent levels, is Dsk2 binding to the proteasome or subcellular localization impacted? Validation of the edited yeast strains should be included. Along these lines, does the presence of K48-Ub chains at proteasomes change in deltaSTI1?

Response to Reviewers

Re: EMBOJ-2025-121482

STI1 domain engages transient helices to mediate Dsk2 phase separation and proteasome condensation

We thank the editor and the three reviewers for their supportive comments and experimental suggestions that undoubtedly improved this manuscript. We successfully addressed the common concerns across all three reviewers regarding *in vivo* analyses on proteasome punctae formation, and better connected these results to the rest of the paper. The summary of additions are:

- Reconstitution experiments showing that Dsk2, proteasomes, and K48-linked ubiquitinated substrates colocalize into condensates; these synergize with prior *in vivo* results in (Waite *et al*, 2024) that K48-linked polyubiquitin (via polyubiquitinated substrates) contributes to proteasome condensate formation in yeast.
- New NMR data showing that only transient helical regions are impacted when isolated STI1 domain is added in *trans*.
- New NOESY-HSQC NMR-based experiments that provide an orthogonal set of measurements that support the transient helical nature of TH2 and TH3 regions.
- Additional CALVADOS simulations (excluded volume) that establish a baseline occupancy of the STI1 groove due to steric constraints without introducing amino acid sequence effects.
- New yeast *in vivo* experiments that include:
 - Dsk2 Δ 3TH (three transient helical regions removed but STI1 intact) where removal of these helices attenuates azide stress-induced proteasome condensate formation in agreement with *in vitro* experiments
 - ALFA-tagged Dsk2 constructs (WT, Δ STI1, Δ 3TH) to obtain Dsk2 protein expression levels in yeast
 - Colocalization data of Dsk2 and proteasome condensates using fluorescently-tagged Dsk2
 - Explanation for why Dsk2 Δ STI1 showed increased proteasome condensate formation (elevated expression levels of Dsk2 and polyubiquitinated substrates)

We address all reviewer comments point-by-point in our responses below.

Referee #1:

In this manuscript, Acharya et al. show that the yeast ubiquitin-receptor shuttle protein Dsk2 self-associates to form condensates *in vitro* through complex STI1 interactions with itself and newly identified transient helices in intrinsically disordered regions (IDRs) of the protein. The researchers used a combination of nuclear magnetic resonance (NMR) spectroscopy, phase-separation microscopy, and computation simulations on full-length Dsk2 to examine these interactions. Through this, three regions inside of Dsk2's IDRs were identified and determined to have propensity to form helices with transient nature, called 3TH. Interactions between STI1 and 3TH were investigated deeply using mutational analysis and deletion constructs. In alignment with 3TH's suggested role in self-association, after STI1 deletion, 3TH amide resonances and CSPs experienced large changes, which was further supported by molecular dynamics simulations. Lastly, researchers examined proteasome puncta formation in yeast cells with deletion of the STI1 domain. Overall, this manuscript helps to clarify a major question in the UBQLN field surrounding the role of the STI1 domain in binding to itself and/or co-chaperones or client proteins and reveal the identity and importance of these unique transient helices in facilitating self-interaction and phase separation. A major benefit of this study is the complementary nature of the NMR experiments and phase separation analysis, which converge on the importance of 3TH:STI1 interactions to facilitate self-association. A weakness of the study is the lack of connection between the themes of Figure 6 and the rest of the study, which could be strengthened. Overall this is a very valuable addition to the UBQLN field and raises quite a few interesting questions that will assuredly inspire additional study from this group and others.

>> We appreciate the reviewer's positive review. As described below, we have generated additional *in vivo* data using a Dsk2 Δ 3TH construct that we believe addresses this reviewer's concerns and strengthens the connection between the *in vivo* (Figure 6) and *in vitro* experiments.

Major Comments:

1. Figures 1 and 6 need a stronger connection to the rest of the paper. They are clearly valuable data, and bring an interesting angle to the manuscript by highlighting how protein accumulation and condensation can be differentially affected by Dsk2 mutation, but they are disconnected from Figures 2-5. For example, this could be strengthened by looking into how deletion of 3TH influences proteasome-containing condensates in yeast. In addition, the Rad23 requirement needs some additional explanation or experimental exploration, as it is unknown why the Rad23 null background is required for strong puncta phenotypes?

>> We thank the reviewer for their excellent suggestions. In yeast, we successfully generated Dsk2 Δ 3TH (deletion construct of all three transient helical regions) in several yeast strains as will be explained below (Figure 6). Our new data suggest that the Dsk2 Δ 3TH substantially reduced the amount of proteasome puncta under azide stress (Figure 6A, 6B). This result synergizes with our *in vitro* data and our hypothesis that 3TH mediates Dsk2 phase separation and condensate formation. Under prolonged growth stress conditions (3d YPD), Dsk2 Δ 3TH reduces the amount of proteasome puncta formation relative to Δ STI1, but the levels of proteasome puncta formation are slightly elevated compared to WT (Figure 6G, 6H). We hypothesized that this stress-dependent effect could be due to changes in protein expression levels. To test this hypothesis, we generated additional Dsk2 yeast strains using the ALFA tag (Götzke *et al*, 2019). These strains include Dsk2 WT, Dsk2 Δ STI1, and Dsk2 Δ 3TH (Appendix Figure S7). We used the ALFA tag to monitor the protein expression levels of Dsk2 due to the unavailability of a suitable Dsk2 antibody for yeast lysates (mentioned in Study Limitations). Relative to WT Dsk2, we found that Δ 3TH expression levels were upregulated by a factor of 2.5 under prolonged growth conditions (Figure 6I) but unaffected under azide stress (Figure 6C). Furthermore, these ALFA-tag measurements help us explain why we observed elevated proteasome condensate formation for the Δ STI1 construct under **both** stress conditions, as Dsk2 Δ STI1 protein expression levels are 2.5-3X higher relative to WT (Figure 6C, 6I). Together, these data show that deletion of 3TH negatively regulates stress-induced proteasome puncta formation in line with our *in vitro* data showing that removal of 3TH negatively affects Dsk2 phase separation. However, the effects are stress-dependent (azide vs. 3d YPD) as illustrated by differences in Dsk2 protein expression levels under two stress conditions. We also refer the reviewer to our response to Reviewer #2 Major Concern #2.

The new sets of experiments described above were completed in both WT yeast strains (containing rad23) and in rad23 Δ background (rad23 removed). We have shown new results using the rad23 Δ background in Figure 6, with WT yeast strain data in Expanded View Figure EV2. The rationale for both backgrounds stems from (Waite *et al*, 2024). Stress-induced proteasome condensates require sufficient amounts of polyubiquitin, Rad23, Dsk2, and proteasomes. Rad23 and Dsk2 each contribute partially redundantly to proteasome condensate formation in yeast, as only the rad23 Δ dsk2 Δ double knockout strains are greatly inhibited in forming proteasome puncta (Figure 6B, 6H, Expanded View Figure EV2C, Expanded View Figure EV2J). This redundancy is consistent with the reported overlapping functions for Rad23 and Dsk2 in proteostasis. Deletion of Rad23 may also sensitize cells to various effects caused by Dsk2 STI1 or 3TH deletion, one of which being proteasome condensate formation.

To better connect Figure 1 to the rest of the manuscript, we repeated the reconstituted proteasome condensate experiments using K48-linked polyubiquitinated substrates. In line with our expectations, we observed that K48-linked polyubiquitinated substrates promoted Dsk2 phase separation (Figure 1B) but to a lesser extent than K63-linked polyubiquitinated substrates. We observed an identical trend using UBQLN2 and suspect this is partially due to differences to chain conformation (Valentino *et al*, 2024; Dao *et al*, 2022). Similarly, K48-linked substrates colocalize with proteasomes and Dsk2 in reconstituted condensates; this supports observations of stress-induced proteasome condensate formation made in yeast as published in (Waite *et al*, 2024).

We have copied below the new Figure 6, but please see the manuscript for the new Expanded View Figure EV2 and Figure 1.

Figure 6. Deletion of the Dsk2 STI1 domain and 3TH regions impact stress-induced proteasome puncta formation.

S. cerevisiae yeast strains (in *rad23Δ* background) were subjected to (A-F) ~24 hr azide stress or (G-L) prolonged growth stress (3 days in YPD). (A, G) Representative brightfield and extended-depth-of-field epifluorescence images (GFP channel) showing proteasome puncta (endogenous Rpn1-GFP) in strains expressing Dsk2 variants. Scale bar: 10 μ m. (B, H) Percentage of cells with ≥ 1 punctum after stress. (B) **P=0.0023, ***P=0.001, ****P<0.0001; one-way ANOVA with Dunnett's post-hoc test ($\alpha=0.05$), $n \geq 4$. (H) *P=0.0301, ***P=0.0007, ****P<0.0001; one-way ANOVA with Dunnett's post-hoc test ($\alpha=0.05$), $n \geq 4$. (C, I) Representative immunoblot of whole-cell lysates showing relative levels of endogenous Dsk2^{ALFA} variants normalized to Pgk1 and scaled to Dsk2^{ALFA} WT. $n = 3$. (D-F, J-L) Representative immunoblot of whole cell lysates from Dsk2 variant strains showing (E, K) K48-linked polyubiquitin and (F, L) Rpn1-GFP levels, normalized to Pgk1 and scaled to Dsk2 WT. $n = 4$. Each biological replicate is denoted by a single symbol wherever applicable. On plots, horizontal line and error bars represent mean and SD, respectively. Full blots in Appendix Figure S8, S9.

2. Why does the minimal system in Figure 1 use a K63-linked substrate while Figure 6 examines K48-linked ubiquitin? It would be nice to add a K48-linked substrate to Figure 1 or blot for K63-linked ubiquitin in Figure 6.

>> The reviewer raises a valid point. In our prior work, we noted that K63-linked polyUb chains promote UBQLN2 condensate formation at lower concentrations than K48-linked polyUb due to differences in chain conformation (Dao *et al*, 2022). For this reason, we initially focused on Dsk2 condensate formation with K63-linked polyubiquitinated substrates and proteasomes in Figure 1. As noted above, we decided to reconstitute Dsk2 condensates for Figure 1 using K48-linked polyubiquitinated substrates as described in our recent publication (Valentino *et al*, 2024). We observed that K48-linked polyubiquitinated substrates can also promote Dsk2 condensate formation and that proteasomes, Dsk2, and K48-linked polyubiquitinated substrates all colocalize into the same condensates. These new data have been added to revised Figure 1 in panels B and C. Both K48-linked and K63-linked chains interact with the proteasome *in vitro* but K48-linked chains preferentially interact with the proteasome in cells (Nathan *et al*, 2013; Martinez-Fonts *et al*, 2020; Saeki *et al*, 2009). Thus, of the two suggestions from the reviewer, we focused on adding the K48-linked polyubiquitinated substrates to our reconstitution experiments in Figure 1.

3. The use of a ubiquitinated protein to potentiate Dsk2 phase separation is a bit confusing considering Dao *et al* 2018 shows that the addition of ubiquitin causes a dissipation of condensates of UBQLN2. Further, given the focus of the rest of the manuscript on ST11 domain interactions, one wonders whether the same effect would

occur for a non-ubiquitinated protein known to bind UBQLNs, such as ATP5G, Omp25, or others.

>> We apologize for the confusion. The reviewer is correct in that our prior work showed that monoUb dissolves UBQLN2 droplets (Dao *et al*, 2018). However, the multivalency of a polyUb chain makes the polyUb suitable for promoting phase separation of Ub-binding proteins in a concentration-dependent manner. For example, a Ub4 chain has four Ub units that are covalently attached to each other in a 'bead-on-a-string' conformation. This makes Ub4 a scaffold on which multiple UBQLN2 proteins can bind and locally increase UBQLN2 concentration to favor phase separation. This effect is not just seen with UBQLN2 (Dao *et al*, 2022; Galagedera *et al*, 2023; Valentino *et al*, 2024) but also with other Ub-binding proteins such as p62 and NEMO (Sun *et al*, 2018; Goel *et al*, 2023; Du *et al*, 2022). To clarify this point in the text, we added this text to the first Results section: "The presence of polyUb-substrates is essential as multivalency of a polyUb chain makes it suitable for promoting phase separation of UBQLNs (Valentino *et al*, 2024; Dao *et al*, 2022; Galagedera *et al*, 2023), while monoUb dissolves UBQLN2 droplets (Dao *et al*, 2018)" with associated references.

The second part of the reviewer's question on the effects of non-ubiquitinated proteins on phase separation is very interesting. However, we think that addressing this question is outside the scope of the current paper, which is focused on the new identification of transient helical regions *within* Dsk2 that interact with the STI1 domain. We will address this reviewer's question in follow-up work as we will need to systematically identify non-ubiquitinated proteins that interact with Dsk2, and also test whether substrates that interact with other UBQLNs can also modulate Dsk2 phase separation.

4. For the CALVADOS simulations, it seems possible that TH1 has a higher probability of interacting with STI1 due to its proximity to the STI1 domain (Figure 4E). For example the probability appears high for non TH-sequences at +10 to the STI1 domain as well. Is it possible to perform simulated mutagenesis to parse this out? For example, what would happen if the TH sequences were scrambled among each other, or moved in increments of ~20 AA?

>>> This is an excellent suggestion. Indeed, proximity to the STI1 cavity does influence the probability of IDR-STI1 interactions. However, mutagenesis (or insertions/deletions) introduce new complexities due to changes in amino acid sequence, and the question then becomes what is the relevance of the now mutated sequence to the actual effect. We can circumvent this problem computationally by performing excluded volume (EV) simulations. In these simulations, all non-bonded interactions are turned off and only

hard repulsions remain. These EV simulations establish a baseline occupancy of the STI1 groove due to steric constraints without introducing amino acid sequence effects.

We have now performed these EV simulations and added a new Appendix Figure S6 where we compare the full attractive and EV simulations. As expected, the EV simulations show that the probability of STI1 occupancy decreases with increasing distance from the STI1 domain. Importantly, regions with increased probability in the normal simulations compared to the EV baseline can be attributed to specific sequence effects, and not steric constraints. To further highlight the increased probability of the THs, we have plotted the excess probability, calculated as the ratio between the probability of the full attractive system (normal simulations) to the probability of the EV system (Appendix Figure S6B). This shows that the TH regions have greater than an order of magnitude increase in probability, further suggesting that increased probability is driven by TH amino acid sequence chemistry. We have updated our Methods section to reflect these new EV simulations.

Minor Comments:

The introduction may benefit from some additional introduction of proteasome granules to tie in Figures 1 and 6.

>>> We added the following to the opening paragraph of the introduction: “The formation of distinct stress-induced proteasome-containing puncta, including cytoplasmic proteasome storage granules in yeast and nuclear proteasome condensates in mammalian cells, may be essential for proteostasis and resistance to stress (Yasuda *et al*, 2020; Uriarte *et al*, 2021; Laporte *et al*, 2008; Gu *et al*, 2017).” We also have the following text in our introduction: “Dsk2 and Rad23 are critical drivers of proteasome condensates in stressed yeast cells and deletion of the genes encoding both proteins abrogates the formation of proteasome-containing condensates in yeast (Waite *et al*, 2024).”

Can the authors comment on the apparent 'shell-like' pattern of Proteasome staining in the condensates of Figure 1c?

>>> We thank the reviewer for their observation. We agree that this is an interesting question but we do not want to overinterpret this signal given that this observation is from *in-vitro* reconstitution experiments. We do note that others have observed this shell-like pattern in other nuclear proteasome condensates containing a different Ub-binding protein, p62 (Fu *et al*, 2021). We hope to follow up on this observation in future work.

It would be helpful for the reader to have some simple way of discriminating predicted structural data compared to NMR data, for example in Figure 1B top versus bottom or

Figure 1D the bottom pLDDT from Alphafold. Could these predicted data plots be colored differently or otherwise distinguished more clearly?

>> Yes. We now colored predicted structural data in skyblue and modified the figure legend to read: 'Predicted data in panels B and E are colored skyblue for clarity'.

Figure 4f is not referenced in the text and it is unclear what it is meant to convey

>> We have reorganized Figure 4 and ensured that all subpanels are referenced in the main text.

It would be nice to include a sequence alignment of 3TH from Dsk2 and human UBQLNs, perhaps as an addition to Figure S1. For that matter, the Discussion on page 21 implies that there may be TH sequences from other proteins that might work together with Dsk2/UBQLNs - it would be very interesting to discuss this more.

>>> We appreciate the reviewer's suggestion and have added a sequence alignment comparing Dsk2 and the human UBQLNs 1, 2, and 4. We have added this as panel B to Appendix Figure S1. We completed this multiple sequence alignment using M-coffee, a multiple sequence alignment (MSA) tool that runs several MSA methods. Notably, the human UBQLNs have two STI1 domains unlike the single STI1 domain of Dsk2, which makes the identification of TH helices in UBQLNs more complicated. From our analysis, we noted some similarities in sequence for TH3 across Dsk2 and human UBQLNs, as the sequence alignment identified the UBA-adjacent (UBAA) helix in UBQLNs. In the recent preprint from the Wohlever group, the authors also tried to identify potential "placeholder" helices in UBQLN2 (we refer the reviewer to Figure S6 in (Onwunma *et al*, 2025)). We are currently in the process of systematically analyzing potential transient helical sequences in follow-up work.

The Discussion 's third paragraph on page 22 is rather confusing

>> We substantially rewrote the entire Discussion section in light of new *in vivo* yeast data. We hope this will clarify issues for the reviewer.

For someone outside of the protein NMR field, phrasing and general descriptions of methods and results could be strengthened to enable a more concise takeaway for a less familiar audience.

>> We have simplified language throughout the main manuscript to make the NMR data more digestible and hope this will address reviewer concerns and questions.

• Rajendran and Castañeda reference is cut off and missing some detail

>> We apologize for this and have corrected this mistake.

Referee #2:

Summary and significance.

Dsk2 is a ubiquitin shuttle factor that delivers ubiquitinated proteins to the proteasome for degradation. Dsk2 consists of three domains - a UBA, STI1, and UBL domain. Through its UBA domain, Dsk2 binds to polyubiquitinated substrates and shuttles these substrates to the proteasome via an interaction with its UBL domain. Loss of Dsk2 can lead to the accumulation of protein aggregates, upsetting cellular homeostasis, and harming the cell. The STI1 domain is the least structured of these domains and is thought to be an intrinsically disordered domain that becomes structured when bound to substrate.

>>> We appreciate the reviewer's comments thus far. We wanted to clarify that we are not suggesting that Dsk2's STI1 domain is intrinsically-disordered or that it becomes structured when it is bound to a substrate. Our data suggest that the STI1 domain has several well-folded helices as the secondary Ca-Cb chemical shifts are consistent with the well-folded helices in the UBA domain (Figure 2). However, many backbone amide resonances of STI1 domain are broadened beyond detection, suggestive of increased dynamics that is likely important for engaging substrates (subject of ongoing work). STI1 domains from other proteins may be disordered and/or change upon binding to substrate (Lin *et al*, 2021; Ji *et al*, 2025).

Other proteins containing STI1 domains include the co-chaperones SGTA, Hop, and Hip as well as ubiquitin associating proteins like UBQLNs. Acharya *et al* reveals that there are intrinsically disorder transient helices between the UBA and STI1 domain and the STI1 domain and UBL that interact with STI1. Additionally, the authors show *in vitro* that STI1 is the main driver of Dsk2 self-association and phase separation. *In vivo*, Dsk2 lacking STI1 in cells lacking Rad23 promotes the formation of proteasome condensates.

The authors present structural studies with various Dsk2 mutants to determine the regions responsible for phase separation. Generally, the analyses are interesting and enhance our understanding of how Dsk2 self assembles and phase separates.

>>> We appreciate the reviewer's overall summary and positive outlook on the manuscript.

Concerns:

1. When investigating the STI1 interactions with transient helices, the authors generate Δ STI1 Dsk2 mutants. I wonder if removing a nearly 100aa section that adopts a

structured conformation and is necessary for many Dsk2 function could affect the stability of the protein. Though the authors address this with "The STI1 deletion-induced changes in 3TH backbone dynamics could result from: a) removal of STI1-3TH interactions, or b) destabilization of the secondary structure of these transient helices. However, the latter is unlikely as the measured hetNOE values (Figure 4C) and α -helix propensity (Figure 4D) for the 3TH regions in Δ STI1 remain similar when compared with data for Dsk2 FL." The argument could be strengthened by either mutating the STI1 domain, perhaps by disrupting the hydrophobic groove formed by the Dsk2-STI1. The authors could also investigate the specificity of this interaction by replacing Dks2-STI1 with the STI1 domain from another protein like the UBQLNS, HIP, Hop, or SGTA.

>>> We appreciate the insightful suggestion. As the reviewer notes, we have performed some analyses already about the effects of the STI1 deletion on the rest of the Dsk2 protein. The deletion of STI1 domain apparently has negligible impact on the hetNOE values for the rest of the protein indicating that the stability of secondary structures remains unperturbed and that the stability of the UBL and UBA domains are largely unaffected (Figure 4D).

In response to this reviewer request, we have obtained preliminary NMR data where we have replaced the STI1 of Dsk2 with UBQLN2 residues 379-462 (STI1-II) in a new construct called Dsk2-Chimera (Reviewer Figure 1). We compared NMR spectra of this construct with both Dsk2 FL and Dsk2 Δ STI1. Importantly, we observed that the peak positions of 3TH residues in Dsk2-Chimera are more similar to Dsk2 FL instead of Dsk2 Δ STI1 indicating that the THs of Dsk2 can interact with STI1 domain from another protein. In the reviewer figure, many of the backbone amide resonances for TH2 and TH3 helical regions nearly overlap when comparing Dsk2 FL (gray) and Dsk2 Chimera (cyan) positions. As we have not yet fully characterized the STI1 domains of other UBQLNs and this is the subject of future work in our lab, we respectfully ask to not include these data in the manuscript.

Reviewer Figure 1: Overlay of ^1H - ^{15}N TROSY-HSQC spectra of Dsk2 FL, Dsk2 ΔSTI1 , and Dsk2-Chimera collected under identical conditions. Note that the backbone amide positions for Dsk2-Chimera (cyan) resonances in transient helices (annotated with “TH”) nearly overlap with Dsk2 FL (gray) positions. Dotted lines are used to show trajectories of resonances across the three protein samples.

2. This builds off the first point. The *in vivo* work is difficult to contextualize. Again, I have the same concerns about making a truncated complex and placing the three transient helices so much closer to each other in sequence. In WT cells, nearly all cells have at least one puncta. There is a stark decrease in the number of cells with puncta in the ΔRad23 cells, and in cells lacking Rad23 and Dsk2- ΔSTI1 there is an increase in puncta. Is Rad23 the necessary factor for proteasome puncta formation? And in the absence of Rad23 only Dsk2 without STI1 forms puncta? How does this fit with the *in vitro* work that highlights Dsk2-STI1 as the major contributor to self-association and phase separation.

Perhaps the authors can address the possibility that without its STI1 domain Dsk2 cannot perform its chaperone activity by replacing the STI1 domain with the STI1-domain of another STI1-domain containing protein.

>>> We appreciate this reviewer's suggestion. Many aspects of this question is similar to reviewer #1 major concern #1 and we refer the reviewer to that discussion in addition to the comments here.

The reviewer makes a salient point that without the STI1 domain Dsk2 might not be able to perform some of its chaperone activity. Indeed, the deletion of the STI1-domain leads to increased levels of Dsk2 (Figure 6C, 6I; determined using newly generated yeast strains with internal ALFA-tag at endogenous locus that showed similar phenotypic trends as un-tagged Dsk2 - see Appendix Figure S7). We also observed elevated levels of polyUb chains (Figure 6E, 6K). We speculate that removing the STI1 domain has dysregulated Dsk2 chaperone activity, as the reviewer suggested, and that the yeast cells are trying to compensate by upregulating Dsk2 expression levels and substrate ubiquitination.

We opted not to replace the STI1 domain of Dsk2 with another STI1 domain of a different protein because this may affect native STI1-client interactions that are important for yeast and/or introduce new STI1-STI1 interactions that could further complicate interpretations of the results of *in vivo* stress-based proteasome condensate experiments.

Instead, to address this reviewer concern, we included new experiments *in vivo* using Dsk2 Δ 3TH that were discussed above for reviewer #1 (Figure 6). In this construct, we preserve the STI1 domain (and presumably its chaperone activity) but eliminate the three transient helical regions (roughly ~35-40 AAs instead of 100 AAs) within the disordered regions of Dsk2. Our new data with Dsk2 Δ 3TH suggest that the amount of azide stress-induced proteasome puncta formation is reduced significantly compared to wild type *without* major changes in Dsk2 and polyUb levels (Figure 6A-E). These results complement our *in vitro* phase separation data and analyses. In the prolonged growth stress condition, we observed increased proteasome puncta formation for Dsk2 Δ 3TH cells relative to WT but we believe this is due to increased Dsk2 levels (2.5 -fold), elevated polyUb chain levels, and proteasome levels in prolonged growth stress (Figure 6G-L). We explain these results in detail in our results and discussion sections. In summary, we believe that inclusion of these new *in vivo* Δ 3TH data will help us eliminate the concern of what happens to the STI1 domain's functionality as the STI1 domain is preserved in the Δ 3TH construct.

Similar to reviewer 1 major concern #1 about Rad23, we have emphasized here that both Rad23 and Dsk2 play important roles in mediating proteasome condensate formation and that there is likely a redundancy between Dsk2 and Rad23 that confers their importance in mediating the formation of proteasome condensates as previously observed in (Waite *et al*, 2024). We mention that point in the opening paragraph of the Results subsection titled “Deletion of the Dsk2 STI1 domain or 3TH regions impacts proteasome condensate formation *in vivo*”: “These deletions were introduced in *rad23Δ* cells (RAD23 deletion strain) expressing GFP-tagged proteasomes (Rpn1-GFP); the *rad23Δ* background was selected to eliminate redundancy from Rad23 in proteasome condensate formation (Waite *et al*, 2024).” As mentioned in our Reviewer 1 response, only the *rad23Δdsk2Δ* double knockout strains are greatly inhibited in forming proteasome puncta (Figure 6B, 6H, Expanded View Figure EV2C, Expanded View Figure EV2J). We separated the results such that proteasome condensate formation results in *rad23Δ* yeast strains are shown in main Figure 6, while results in the WT yeast strains (containing Rad23) are shown in Expanded View Figure EV2.

3. The authors decision to threshold the intensity of the puncta in Fig 6A and B is not convincing. Particularly in Panel B the Dsk1 Δ STI1 construct the lower left cell looks like it has two puncta, but these puncta apparently don't meet the threshold cut-off. Perhaps if the authors set a threshold that is normalized to the intensity of each individual cell.

>>> We appreciate the reviewer's concern that presenting data using intensity thresholding may give the impression that some puncta were excluded from analysis. Our intention was to quantify the qualitative observation that puncta appeared brighter under certain conditions. However, we agree that quantifying the percentage of cells with one or more puncta provides a more straightforward representation of the data. We have removed the intensity thresholding data from the manuscript. Subpanels in Figure 6 and Expanded View Figure EV2 quantify the percentage of cells with one or more proteasome punctum/condensate.

Major comments:

1. Throughout the Introduction the authors use the terms "STI1 domain" and "STI1-like domain". When describing the domain organization of Dsk2, the authors state that it consists of a STI1-like domain. However, Fry *et al*, 2021 identifies the domain within Dsk2 as a STI1-domain and the authors refer to this domain in the rest of their text as a STI1 domain.

>>> We apologize for this mistake and have replaced all “STI1-like” references with STI1.

2. The authors should introduce the three transient helices in the introduction section. This will make it clearer to the reader that the helix in the helix/STI1 interactions they observe and discussed at the end of the introduction is part of Dsk2 and not a substrate.

>>> Thank you for the suggestion. We decided to reword the concluding paragraph of the introduction to explicitly state our finding about the transient helices and their interaction with the STI1 domain: “We identified three transient helical regions within the disordered regions of Dsk2 that interact with Dsk2’s own STI1 domain.”

3. The authors discuss the concentration dependent phase separation of Dsk2 - referring to the physiological levels of Dsk2 would put their observations into perspective - especially in the discussion section.

>>> We appreciate the reviewer’s suggestion here. Our reconstitution work in Figure 1 used 10 μM Dsk2, which is an order of magnitude higher than estimated endogenous concentrations of Dsk2 (Ho et al, 2018). However, we note that stress conditions alter (often increase) both protein and substrate levels (Figure 6), and increasing amounts of polyUb substrates promote Dsk2 phase separation (Figure 1). We added information to our Limitations section: “A limitation of our proteasome condensate reconstitution experiments is that the concentration of Dsk2 used is an order of magnitude more than the endogenous concentration, which is estimated at $\sim 0.3 \mu\text{M}$ (Ho et al, 2018), though stress conditions can alter protein levels.”

4. Figure 5 A & B are difficult to compare. $\Delta 3\text{TH}$ appears to have a similar c_{sat} at 5C to that of ΔUBL and mutUBA , but it’s difficult to see the differences between these mutants at low temperatures. Combining these plots would allow for easier comparison by the reader.

>>> We have addressed this suggestion by combining these two panels and showing the phase diagram for all Dsk2 deletion constructs alongside full-length Dsk2. Please see new Figure 5A in the main manuscript.

5. Figure 7 would be easier to interpret if the authors colored the helices bound to Dsk2-STI1 domain by hydrophobicity to illustrate the amphipathicity of these helices.

>>> We have adjusted this figure accordingly. Figure 7 now colors the sequence content of the helices according to the hydrophobicity scale.

Minor comments:

1. In the abstract: "In vivo, STI1 removal results in an increase of proteasome-containing...". In vivo should be italicized.

>>> We have modified the text accordingly and apologize for the oversight.

2. First paragraph of the results section: "Using fluorescence microscopy," The authors do not mention what fluorophore they used or what fluorophore is attached to which component in their minimal in vitro system.

>>> We apologize for this mistake. We have updated our legend to Figure 1 to include this information: "(B) Fluorescence microscopy showing phase separation of K63-linked or K48-linked R-Neh2Dual-sGFP polyUb-substrates (see Methods) with Dsk2 (spiked with Alexa Fluor 647-Dsk2) in a concentration-dependent manner (3% PEG 8000, 18°C, 0 or 10 μM Dsk2, 0-200 nM substrate). Scale bar, 5 μm. (C) Fluorescence microscopy of 10 μM Dsk2, 200 nM K63-linked or K48-linked polyUb-substrate, and 100 nM TagRFP-T-Rpn6 proteasome incubated at 18°C in pH 7.5 buffer with 3% PEG 8000. Scale bar, 5 μm."

This information is also provided in the Methods section.

3. The order for Fig1 panel B is not intuitive. I suggest moving the second column to the left (200nM Substrate and 0uM Dsk2) and the left most panel to the second column. This way it will read a condition where there is no Dsk2 and only substrate and then an increasing concentration of substrate with constant Dsk2 concentration.

>>> We have adjusted this figure accordingly and included results with K48-linked polyUb-substrate.

4. Page 21 second paragraph "Functionally, STI1 domains of co-chaperone like proteins" - these proteins are co-chaperones, not co-chaperone like.

>>> We have corrected this mistake.

5. Page 21: In Lin et al 2021, when determining the minimum substrate requirement for Sgt2 the authors generated artificial substrates using a mixture of leucines and alanines to control for hydrophobicity and that the identity of the residues isn't what is important.

>>> Thank you for pointing this out. We have adjusted the text in the Discussion to the following: "The STI1 domain of the co-chaperone Sgt2 interacts with transmembrane domain (TMD) client helices with varying affinities depending on helix length and distribution of hydrophobic residues along the helix (Lin et al, 2021). Highly hydrophobic helices with > 11 residues were strong binders to the STI1 domain. Similarly, the three

transient helices in Dsk2 vary in helix length, number and distribution of hydrophobic residues (Appendix Table S3), suggesting they may bind the STI1 domain with varying affinities.”

Referee #3:

This manuscript elegantly uses NMR and biochemical assays to characterize Dsk2 intramolecular and intermolecular interactions. Some strengths are quality of the NMR experiments, with some gems including the strongly correlated experimental and predicted Calpha values and correlation of AlphaFold pLDDT scores with NMR relaxation data - the latter might not have been shown before. The concentration-dependent data are convincing of STI1-dependent self-association and the CSP values with the STI1 deleted illustrate effects on the three identified interacting regions. These interactions are also convincingly shown to contribute to Dsk2 phase separation. The authors further find increased proteasome puncta following STI1 deletion and propose that this is an indirect effect driven by accumulation of K48-Ub proteins. This later part of the paper is less convincing, and it is not clear why loss of the STI1 domain would directly drive K48-Ub accumulation - added to this concern is the lack of an antibody to validate the presence of the edited protein.

>> We thank this reviewer for their positive and thoughtful assessment. We agree that the connection between the STI1 domain deletion and K48-linked polyUb accumulation was not well-described in this initial version of the manuscript as also brought up by reviewers #1 and #2. We have addressed these concerns with new *in vivo* data using the Δ 3TH deletion construct (where the STI1 domain is kept intact but the three transient helical regions are removed) and with new data that quantifies Dsk2 protein levels (using ALFA-tagged Dsk2) and ubiquitination levels. To speculate on the elevated K48-linked polyUb accumulation, we think that loss or reduction of Dsk2's chaperone activity due to STI1 deletion results in more unfolded or mislocalized proteins that are identified by ubiquitination machinery for degradation by appending polyUb chains to them. These experiments are described below (in the responses to specific concerns) but we also refer this reviewer to our responses to Reviewers 1 and 2.

Some additional questions/suggestions are listed below.

Figure 2A: The UBL domain peaks should be labeled to make it easier to relate back to the text.

>> We have modified Figure 2A as requested to include both labeled and color-coded UBL and UBA backbone amide resonances.

A table with the Dsk2 chemical shift assignments should be included.

>> We have submitted our NMR chemical shift assignments to the BMRB (Biomagnetic Resonance Bank) as part of standard practice. The accession number is: 53439. You can access the data here: https://bmrbl.io/author_view/53439_hy_kjbaarew.str

Figure 3A: For this experiment and for all the NMR experiments it'd be worth listing the number of scans taken. The signals appear in the figure quite strong. Was a NOESY experiment attempted? Even amide-amide NOEs could help establish helicity in the regions of interest.

>>> Thank you for this comment and the excellent NMR experiment suggestion. We have added information about the number of scans to figure legends. We aimed to keep experimental conditions identical across all NMR experiments to the extent possible to enable fair comparisons for all samples. We listed details within the Methods sections. Additionally, we collected a ^1H - ^{15}N HSQC-NOESY experiment using our Dsk2 STI1+IDR construct, which contains TH2 and TH3 transiently helical regions (Appendix Figure S4 and reproduced below). We observed $i:i+1$ and $i:i-1$ amide NOEs for several resonances within TH2 and TH3 regions. We observed very weak or no amide NOEs for disordered regions. These NOESY data support our Ca-Cb secondary chemical shift data that the TH2 and TH3 regions exhibit transient helicity. We added this text to the second paragraph of the results section titled "Short helices exist within intrinsically disordered regions of Dsk2": "To corroborate these observations, we collected NOESY spectra that identify through-space $\text{H}^{\text{N}}\text{-H}^{\text{N}}$ interactions. We detected strong $i\pm 1$ NOEs for some of these helical regions (Appendix Figure S4), reflecting close spatial proximity between sequential residues within the helices (Eliezer *et al*, 2000)."

Appendix Figure S4. Amide-amide NOEs reveal helical regions within Dsk2 IDRs. Strip plots from the 3D ^1H - ^{15}N HSQC-NOESY spectra are shown for selected regions of Dsk2 IDRs (see AlphaFold-predicted structure of Dsk2), highlighting amide-amide NOE cross-peaks. Each strip corresponds to a specific ^{15}N chemical shift, with the amide ^1H - ^1H peaks displayed along the horizontal axis. Within a strip, the self peak is indicated by intersection of the solid black diagonal line and a dashed vertical line. Corresponding residue (i) is labeled below each strip. Amide-amide NOE cross-peaks (inter-residue NOEs) are marked in red at their respective positions, indicating spatial proximity between backbone amide protons wherever shown. We observed strong $i\pm 1$ amide-amide NOEs within the TH2 and TH3 regions supporting the presence of helical secondary structure, while disordered regions (middle two strip plots) exhibit

very weak or no amide-amide NOEs. To reduce spectral complexity, we used the STI1+IDR construct (contains TH2 and TH3) for this HSQC-NOESY experiment.

Figure 4B: This data is convincing of changes in the three regions following STI1 deletion. It might be worth also adding in STI1 and examining whether the signals return to their state in the full length protein. Another experiment that could be done is to add in spin labeled STI1 and look for resonance broadening.

>>> We thank the reviewer for these excellent suggestions. To address this concern, we performed a NMR experiment where we titrated a 8-molar excess of unlabeled STI1 domain (400 μM) to a ^{15}N -labeled Dsk2 ΔSTI1 sample (50 μM). Strikingly, we observed that the amide resonances of the transient helical regions moved towards their Dsk2 FL positions (revised Figure 4A reproduced below and full NMR spectra in Appendix Figure S5). Resonances belonging to TH1 (115, 123, 126), TH2 (280, 282, 291), and TH3 (314) all move towards their Dsk2 FL positions. We believe that the TH resonances in the Dsk2 ΔSTI1 sample do not reach their respective Dsk2 FL positions because the STI1:TH interactions are occurring in *trans* during the titration whereas in Dsk2 FL, these STI1:TH interactions are occurring in *cis* (where local concentrations of TH for STI1 is higher). We have added this language to the results section titled “STI1 domain interacts with transient helices in Dsk2”: “To validate STI1-3TH interactions, we titrated unlabeled isolated Dsk2 STI1 domain into a ^{15}N Dsk2 ΔSTI1 sample; this experiment resulted in shifts of 3TH resonances toward their Dsk2 FL positions (Figure 4A, Appendix Figure S5).”

Figure 4A (Right) Addition of 8X isolated unlabeled STI1 domain to ^{15}N Dsk2 ΔSTI1 moved amide resonances of 3TH regions towards their respective positions in Dsk2 FL.

Figure 4G: This image is not a good representation of the movie provided. To summarize the simulations, distances over time should be included and average values with deviations. Grey regions also enter the red STI1 cavity and this should be

represented in some way. These analyses should be provided for both simulations (with and without UBL/UBA interaction).

>>> We apologize for any confusion this figure may have caused. Figure 4G (now Figure 4H) was only meant to highlight example states where the STI1 domain engages with each of the transient helical regions. We also included supplementary movies S1 (now Movie EV1) and S2 (now Movie EV2) that are the time course movies of full-length Dsk2 in both unbound and bound UBL-UBA conformations, respectively. We apologize that this information was not made clearer.

We also acknowledge that indeed many residues in the IDR occupy the STI1 groove over the duration of our simulations. As each trajectory samples different stochastic pathways, averaging distances obscures the residue-specific preferences and transient interactions that we aim to highlight. Therefore, we opted to use a probability representation, which provides what the reviewer is asking for - showing the probability of each residue occupying the STI1 groove with average and standard deviation. This is shown in Figure 4F where we plot the ensemble-averaged probability of each residue occupying the groove. We have clarified this in the main text by saying: "From these single-protein simulations, we noted that the disordered regions of Dsk2 transiently occupied the STI1 groove (Movie EV1). We quantified the ensemble-averaged probability of the disordered regions occupying the STI1 groove, and specifically observed an increased preference of all three TH regions to interact with STI1 (shaded orange regions in Figure 4F)."

We also refer the reviewer to new Appendix Figure S6, where we performed new excluded volume (EV) simulations that establish a baseline occupancy of the STI1 groove due to steric constraints without introducing amino acid sequence effects. This was in response to Reviewer #1 major concern #4.

It's not clear why Δ STI1 would lead to increased K48-Ub in the lysates.

>>> Dsk2 Δ STI1 leads to both elevated Dsk2 protein levels and ubiquitinated substrate (K48-linked polyUb) levels. We believe that the loss of STI1 domain in Dsk2 has led to significantly diminished substrate chaperone activity, leading to the accumulation of polyubiquitinated substrates. We have made extensive changes to the *in vivo* result section, including newly revised Figure 6 and Expanded View Figure EV2. Please also refer to our responses to Reviewers 1 and 2.

In the study limitations section the authors note a lack of anti-Dsk2 antibodies. This is a sizeable limitation - can mass spectrometry be used instead to discern whether the Dsk2 protein levels have changed and the loss of the STI1 domain?

>>> We thank the reviewer for these important observations. To address the Dsk2 antibody issue, we have generated ALFA-tagged Dsk2 strains (see Methods) for several Dsk2 constructs including WT Dsk2, Δ STI1, and Δ 3TH. We did this in two yeast backgrounds: WT and *rad23 Δ* (this background was selected to eliminate redundancy from Rad23 in proteasome condensate formation as described in (Waite et al, 2024)). We validated these ALFA-tagged strains in Appendix Figure S7 and show that the proteasome puncta formation trends are the same between the Dsk2 and ALFA-tagged Dsk2 constructs. Therefore, we quantified relative Dsk2 protein levels using the ALFA-specific antibody for immunoblotting. These data are now shown in revised Figure 6C and Figure 6I. These important results revealed that the protein levels for Dsk2 Δ STI1 were increased nearly three-fold relative to WT Dsk2 under both azide stress and prolonged growth conditions. Furthermore, Dsk2 Δ 3TH protein levels were increased relative to WT for prolonged growth conditions but not under azide stress. These experimental results were important to explain increased proteasome puncta formation for Dsk2 Δ STI1 relative to WT Dsk2.

How do these effects compare to a Dsk2 knockout?

>>> We have added data for *dsk2 Δ* (Dsk2 knockout) replicates alongside experiments using WT Dsk2, Δ STI1, and Δ 3TH. These data are included in revised Figure 6A-B, 6D-F, 6G-H, 6J-L in the main manuscript and Expanded View Figure EV2. The results for *dsk2 Δ* are similar to those recently published in (Waite *et al*, 2024).

Could it be that Δ STI1 causes loss of function or low protein levels?

>>> Yes, as discussed above, our revised data suggests that loss of STI1 results in elevated Dsk2 protein levels (Figure 6C, 6I), indicating that the yeast cells are compensating for loss in Dsk2 activity.

Assuming they are at equivalent levels, is Dsk2 binding to the proteasome or subcellular localization impacted?

>>> Dsk2 protein level quantification indicates that the Δ STI1 levels are elevated compared to WT, however the monitored subcellular localization of Δ STI1 condensates to proteasome condensates remains similar to WT. These data are now shown in revised Expanded View Figure EV2D and EV2K. This indicates that the observed

effects of Dsk2 domain deletions on proteasome condensate formation are likely not driven by altered interactions with the proteasome.

Validation of the edited yeast strains should be included.

>>> For validation, we have added new text to the Methods section and included details in Appendix Table S5 and S6. We also validated the ALFA-Dsk2 strains in Appendix Figure S7 as described above. Yeast knockouts have been described previously and have all been confirmed with PCRs that confirmed deletion of the endogenous genes and PCRs that confirmed integration of the selection cassette at the intended target locus. For the mutations made using CRISPR-Cas9, mutations were confirmed by sequencing the genomic region beyond the repair DNA borders to ensure genomic integration at the correct locus. The CRISPR-Cas9 plasmid was then evicted to eliminate Cas9 enzyme and guide RNA from the yeast. The information of the guide RNA and the repair DNA is provided in supplemental tables (Appendix). In our experience, this is identical or more than the customary validation that is provided. If the reviewer is seeking specific validation data we would appreciate instructions on what specific data he/she likes to see.

Methods:

“*S. cerevisiae* strains used in this work are reported in Appendix Table S5. C-terminal fluorescent tags were introduced at the endogenous locus using standard PCR-based procedures (Waite et al, 2024; Janke et al, 2004; Goldstein & McCusker, 1999). We used a CRISPR-Cas9 plasmid-based approach to delete the Dsk2 STI1 domain, delete the region encoding the three transient helices (3TH), or introduce an internal ALFA Tag between Glu246 and Gly247 of Dsk2 (Laughery et al, 2015; Götzke et al, 2019). Candidate guide RNAs were determined using the CRISPR/Cas9 target online predictor CCTop (Stemmer et al, 2015; Labuhn et al, 2018). Each guide RNA was cloned into the 2 μ Cas9- and sgRNA-expressing vector pML107, replacing the original gRNA sequence (Addgene plasmid # 67639, (Laughery et al, 2015)). The newly generated plasmids were introduced into yeast together with duplex repair DNA to generate strains with the desired genetic changes (see Appendix Tables S5-6 for details). Mutations were confirmed by sequencing the genomic region beyond the repair DNA borders and the CRISPR-Cas9 plasmid was evicted.”

Along these lines, does the presence of K48-Ub chains at proteasomes change in deltaSTI1?

>>> In an effort to address the reviewer’s concern we attempted to do Rpn1-GFP-based pulldown assays to determine changes in proteasome-polyUb associations.

Unfortunately, these experiments were unsuccessful and require optimization and controls that are beyond a reasonable rebuttal timeline in our view. Some complications are that (1) proteasomes have three intrinsic ubiquitin receptors (Rpn1, Rpn10, and Rpn13) that bind polyUb chains and these receptors are important for condensate formation (Waite *et al*, 2024) (2) our lysis conditions seem to disrupt the non-covalent multivalent interactions that hold the proteasome condensates together, as such, pulldowns following lysis likely miss some of the *in vivo* occurring interactions. We continue to optimize our lysis conditions and have attempted several (e.g., conditions with and without crosslinkers), but so far we are not confident any of these conditions retained condensate-specific interactions. (3) The polyUb levels in total lysate in different mutants change as our data indicate, so it will be hard to deduce if a deletion of the STI1 domain leads to more polyUb on the proteasome through Dsk2-dependent binding or that more polyUb leads to more polyUb binding to intrinsic receptors. We also note that there are no data indicating the STI1 domain is directly involved in interactions with either polyUb or the proteasome. It is well-documented that the Dsk2 UBL binds to proteasomes and the Dsk2 UBA binds polyubiquitin chains. Considering these constraints, we think our current conclusions and interpretation are warranted and hope to gain deeper insight in the polyUb chain interactions inside the condensates in future work.

References:

- Dao TP, Kolaitis R-M, Kim HJ, O'Donovan K, Martyniak B, Colicino E, Hehny H, Taylor JP & Castañeda CA (2018) Ubiquitin Modulates Liquid-Liquid Phase Separation of UBQLN2 via Disruption of Multivalent Interactions. *Mol Cell* 69: 965-978.e6
- Dao TP, Yang Y, Presti MF, Cosgrove MS, Hopkins JB, Ma W, Loh SN & Castañeda CA (2022) Mechanistic insights into enhancement or inhibition of phase separation by different polyubiquitin chains. *EMBO Rep* 23: e55056
- Du M, Ea C-K, Fang Y & Chen ZJ (2022) Liquid phase separation of NEMO induced by polyubiquitin chains activates NF- κ B. *Mol Cell* 82: 2415-2426.e5
- Eliezer D, Chung J, Dyson HJ & Wright PE (2000) Native and Non-native Secondary Structure and Dynamics in the pH 4 Intermediate of Apomyoglobin. *Biochemistry* 39: 2894–2901
- Fu A, Cohen-Kaplan V, Avni N, Livneh I & Ciechanover A (2021) p62-containing, proteolytically active nuclear condensates, increase the efficiency of the ubiquitin–proteasome system. *Proc Natl Acad Sci* 118: e2107321118
- Galagedera SKK, Dao TP, Enos SE, Chaudhuri A, Schmit JD & Castañeda CA (2023) Polyubiquitin ligand-induced phase transitions are optimized by spacing between ubiquitin units. *Proc Natl Acad Sci* 120: e2306638120
- Goel S, Oliva R, Jeganathan S, Bader V, Krause LJ, Kriegler S, Stender ID, Christine CW, Nakamura K, Hoffmann J-E, *et al* (2023) Linear ubiquitination induces NEMO phase separation to activate NF- κ B signaling. *Life Sci Alliance* 6
- Götzke H, Kilisch M, Martínez-Carranza M, Sograte-Idrissi S, Rajavel A, Schlichthaerle T, Engels N, Jungmann R, Stenmark P, Opazo F, *et al* (2019) The ALFA-tag is a highly versatile tool for nanobody-based bioscience applications. *Nat Commun* 10: 4403
- Gu ZC, Wu E, Sailer C, Jando J, Styles E, Eisenkolb I, Kuschel M, Bitschar K, Wang X, Huang L, *et al* (2017) Ubiquitin orchestrates proteasome dynamics between proliferation and quiescence in yeast. *Mol Biol Cell* 28: 2479–2491
- Ji T, Ge P, Zhang S, Wan C, Liu H, Qu X, Zhu F, Gong Q, Xu W, Wang C, *et al* (2025) Remote on–off switching of protein activity by intrinsically disordered region. *Nat Struct Mol Biol* 32: 2088–2098
- Laporte D, Salin B, Daignan-Fornier B & Sagot I (2008) Reversible cytoplasmic localization of the proteasome in quiescent yeast cells. *J Cell Biol* 181: 737–745
- Lin K-F, Fry MY, Saladi SM & Clemons WM (2021) Molecular basis of tail-anchored integral membrane protein recognition by the cochaperone Sgt2. *J Biol Chem* 296: 100441

Martinez-Fonts K, Davis C, Tomita T, Elsasser S, Nager AR, Shi Y, Finley D & Matouschek A (2020) The proteasome 19S cap and its ubiquitin receptors provide a versatile recognition platform for substrates. *Nat Commun* 11: 477

Nathan JA, Tae Kim H, Ting L, Gygi SP & Goldberg AL (2013) Why do cellular proteins linked to K63-polyubiquitin chains not associate with proteasomes? *EMBO J* 32: 552–565

Onwunma J, Binsabaan S, Allen SP, Sankaran B & Wohlever ML (2025) ALS mutations disrupt self-association between the Ubiquilin Sti1 hydrophobic groove and internal placeholder sequences. 2024.07.10.602902 doi:10.1101/2024.07.10.602902 [PREPRINT]

Saeki Y, Kudo T, Sone T, Kikuchi Y, Yokosawa H, Toh-e A & Tanaka K (2009) Lysine 63-linked polyubiquitin chain may serve as a targeting signal for the 26S proteasome. *EMBO J* 28: 359–371

Sun D, Wu R, Zheng J, Li P & Yu L (2018) Polyubiquitin chain-induced p62 phase separation drives autophagic cargo segregation. *Cell Res* 28: 405–415

Uriarte M, Sen Nkwe N, Tremblay R, Ahmed O, Messmer C, Mashtalir N, Barbour H, Masclef L, Voide M, Viallard C, *et al* (2021) Starvation-induced proteasome assemblies in the nucleus link amino acid supply to apoptosis. *Nat Commun* 12: 6984

Valentino IM, Llivicota-Guaman JG, Dao TP, Mulvey EO, Lehman AM, Galagedera SKK, Mallon EL, Castañeda CA & Kraut DA (2024) Phase separation of polyubiquitinated proteins in UBQLN2 condensates controls substrate fate. *Proc Natl Acad Sci* 121: e2405964121

Waite KA, Vontz G, Lee SY & Roelofs J (2024) Proteasome condensate formation is driven by multivalent interactions with shuttle factors and ubiquitin chains. *Proc Natl Acad Sci* 121: e2310756121

Yasuda S, Tsuchiya H, Kaiho A, Guo Q, Ikeuchi K, Endo A, Arai N, Ohtake F, Murata S, Inada T, *et al* (2020) Stress- and ubiquitylation-dependent phase separation of the proteasome. *Nature* 578: 296–300

Prof. Carlos A Castañeda
Syracuse University
Biology and Chemistry
Syracuse, NY 13244

19th Dec 2025

Re: EMBOJ-2025-121482R
ST11 domain engages transient helices to mediate Dsk2 phase separation and proteasome condensation

Dear Carlos,

Thank you again for submitting your revised manuscript to The EMBO Journal. Two of the original referees have now both assessed it once more, and were generally satisfied with the revisions. Referee 2 still has a few comments, although only the last one may seem to warrant additional minor modifications. In addition, there are however also still various editorial issues that I would ask you to incorporate at this stage:

- Please double-check to make sure to all relevant funding information in the manuscript is congruent with the info entered into our submission system. Currently missing in the submission system are: NSF grant MCB1935596; postdoctoral support to N.A. and J.K.N. from the Syracuse University Vice President of Research office, and a graduate fellowship from the Madison and Lila Self Graduate Programs at the University of Kansas; Alfred P. Sloan Foundation

- Please reduce the number of keywords on the abstract page to five (ideally choosing specific and general terms).

- Please remove the EV Movie legends from the main text, they should only be present in the uploaded movie ZIP archives.

- Please ensure that the deposited data are now becoming publicly accessible, and update the links in the Data Availability section accordingly (right now, some URLs may still be preliminary referee access links?).

- Please carefully go through the reference list and make sure that each reference is complete with citation year, volume, and page/locator numbers - such information is currently missing for some of them. Also, please adjust the format for citation of preprints as specified in our author guidelines:

The citation in the text should be: "(preprint: NAME1 et al, YEAR)"

The citation in the reference list: "NAME1, NAME2, ... (YEAR) (article title). bioRxiv doi: XXX" [preprint]

- Please double-checked that all Figures and Figure Panels are being referenced in the text in sequential order, e.g. Fig 1D before Fig 1E etc.

- Finally, during routine pre-acceptance checks, our data editors have raised the following queries regarding figures, data, and legends, which I would ask you to address (ideally using the Track Changes option):

- Please note that the exact p values are not provided in the legends of figures 6B,H; EV2 C,

- Please note that information related to n is missing in the legend of figure 4D

- Please note that the error bars are not defined in the legends of figures EV1 D, 4D

- Please note that the measure of center for the error bars needs to be defined in the legends of figures 1D, 2D, 5A

I am returning the manuscript to you for a final round of minor revision, to allow you to make these various modifications and to upload the revised files. Once we will have received them, we should be ready to proceed with formal acceptance and production of the manuscript!

With kind regards,

Hartmut

*** PLEASE NOTE: All revised manuscript are subject to initial checks for completeness and adherence to our formatting guidelines. Revisions may be returned to the authors and delayed in their editorial re-evaluation if they fail to comply to the

following requirements. As a first step please read our guidelines for revised submissions:
<https://link.springer.com/journal/44318/submission-guidelines#cms-Revised-submissions>

1) Every manuscript requires a Data Availability section (even if only stating that no deposited datasets are included). Primary datasets or computer code produced in the current study have to be deposited in appropriate public repositories prior to resubmission, and reviewer access details provided in case that public access is not yet allowed.

4) Each main and each Expanded View (EV) figure should be uploaded as individual production-quality files (preferably in .eps, .tif, .jpg formats). For suggestions on figure preparation/layout, please refer to our Figure Preparation Guidelines:
<https://media.springernature.com/original/springer-cms/rest/v1/content/27825798/data/v1>

6) Please complete our Author Checklist, and make sure that information entered into the checklist is also reflected in the manuscript; the checklist will be available to readers as part of the Review Process File.

8) Please note that supplementary information at EMBO Press has been superseded by the 'Expanded View' for inclusion of additional figures, tables, movies or datasets; with up to five EV Figures being typeset and directly accessible in the HTML version of the article.

9) To facilitate reproducibility and cross-laboratory adoption of methodologies, please structure the Materials & Methods section as outlined in our guide to authors, including a completed Reagents and Tools Table.

10) Digital image enhancement is acceptable practice, as long as it accurately represents the original data and conforms to community standards. If a figure has been subjected to significant electronic manipulation, this must be clearly noted in the figure legend and/or the 'Materials and Methods' section. The editors reserve the right to request original versions of figures and the original images that were used to assemble the figure. Finally, we generally encourage uploading of numerical as well as gel/blot image source data.

In the interest of ensuring the conceptual advance provided by the work, we recommend submitting a revision within 3 months (19th Mar 2026). Please discuss the revision progress ahead of this time with the editor if you require more time to complete the revisions. Use the link below to submit your revision:

Link Not Available

Referee #1:

I appreciate the additional experimentation and editing that the authors have done to address the points raised by myself and other reviewers; the authors have done a nice job improving the manuscript. I have no additional concerns.

Referee #2:

Summary and significance.

The authors describe how the ubiquitin shuttle factor, Dsk2, self-associates through ST11 domain interactions, either through interactions with ST11 domain or transient helices in linker regions between the three main domains in Dsk2 - UBA, ST11, and UBL. By combing NMR spectrometry, imaging, and computational simulations, the authors identify three transient helices, one in the linker between the UBA and ST11 domains and two in the linker between the ST11 and UBL domains. Deletion of the ST11 domain or these three helices in Dsk2 negatively affects condensate formation. Additionally, the authors show in vitro that ST11 is the main driver of Dsk2 self-association and phase separation. In vivo, Dsk2 lacking ST11 in cells lacking Rad23 promotes the formation of proteasome condensates.

The authors present structural studies with various Dsk2 mutants to determine the regions responsible for phase separation. Generally, the analyses are interesting and enhance our understanding of how Dsk2 self assembles and phase separates.

Previous Concerns:

1. When investigating the ST11 interactions with transient helices, the authors generate Δ ST11 Dsk2 mutants. I wonder if removing a nearly 100aa section that adopts a structured conformation and is necessary for many Dsk2 function could affect the stability of the protein. Though the authors address this with "The ST11 deletion-induced changes in 3TH backbone dynamics could result from: a) removal of ST11-3TH interactions, or b) destabilization of the secondary structure of these transient helices.

However, the latter is unlikely as the measured hetNOE values (Figure 4C) and α -helix propensity (Figure 4D) for the 3TH regions in Δ ST11 remain similar when compared with data for Dsk2 FL." The argument could be strengthened by either mutating the ST11 domain, perhaps by disrupting the hydrophobic groove formed by the Dsk2-ST11. The authors could also investigate the specificity of this interaction by replacing Dks2-ST11 with the ST11 domain from another protein like the UBQLNS, HIP, Hop, or SGTA.

>>> We appreciate the insightful suggestion. As the reviewer notes, we have performed some analyses already about the effects of the ST11 deletion on the rest of the Dsk2 protein. The deletion of ST11 domain apparently has negligible impact on the hetNOE values for the rest of the protein indicating that the stability of secondary structures remains unperturbed and that the stability of the UBL and UBA domains are largely unaffected (Figure 4D).

In response to this reviewer request, we have obtained preliminary NMR data where we have replaced the ST11 of Dsk2 with UBQLN2 residues 379-462 (ST11-II) in a new construct called Dsk2-Chimera (Reviewer Figure 1). We compared NMR spectra of this construct with both Dsk2 FL and Dsk2 Δ ST11. Importantly, we observed that the peak positions of 3TH residues in Dsk2-Chimera are more similar to Dsk2 FL instead of Dsk2 Δ ST11 indicating that the THs of Dsk2 can interact with ST11 domain from another protein. In the reviewer figure, many of the backbone amide resonances for TH2 and TH3 helical regions nearly overlap when comparing Dsk2 FL (gray) and Dsk2 Chimera (cyan) positions. As we have not yet fully characterized the ST11 domains of other UBQLNs and this is the subject of future work in our lab, we respectfully ask to not include these data in the manuscript.

>>> Current response: I can respect their desire to exclude this data from this study as it is a part of ongoing work, but I do think that the Dsk2-Chimera NMR spectra closely resembling the Dsk2-FL NMR spectra is an interesting observation. It indicates that their observations in Dsk2 could be applied to other ST11 domain containing proteins - namely UBQLNS. Basically, it broadens their study.

2. This builds off the first point. The in vivo work is difficult to contextualize. Again, I have the same concerns about making a truncated complex and placing the three transient helices so much closer to each other in sequence. In WT cells, nearly all cells have at least one puncta. There is a stark decrease in the number of cells with puncta in the Δ Rad23 cells, and in cells lacking Rad23 and Dsk2- Δ ST11 there is an increase in puncta. Is Rad23 the necessary factor for proteasome puncta formation? And in the absence of Rad23 only Dsk2 without ST11 forms puncta? How does this fit with the in vitro work that highlights Dsk2-ST11 as the major contributor to self-association and phase separation. Perhaps the authors can address the possibility that without its ST11 domain Dsk2 cannot perform its chaperone activity by replacing the ST11 domain with the ST11-domain of another ST11-domain containing protein.

>>> We appreciate this reviewer's suggestion. Many aspects of this question is similar to reviewer #1 major concern #1 and we refer the reviewer to that discussion in addition to the comments here.

The reviewer makes a salient point that without the ST11 domain Dsk2 might not be able to perform some of its chaperone activity. Indeed, the deletion of the ST11-domain leads to increased levels of Dsk2 (Figure 6C, 6I; determined using newly generated yeast strains with internal ALFA-tag at endogenous locus that showed similar phenotypic trends as un-tagged Dsk2 - see Appendix Figure S7). We also observed elevated levels of polyUb chains (Figure 6E, 6K). We speculate that removing the ST11 domain has dysregulated Dsk2 chaperone activity, as the reviewer suggested, and that the yeast cells are trying to compensate by upregulating Dsk2 expression levels and substrate ubiquitination.

We opted not to replace the STI1 domain of Dsk2 with another STI1 domain of a different protein because this may affect native STI1-client interactions that are important for yeast and/or introduce new STI1-STI1 interactions that could further complicate interpretations of the results of in vivo stress-based proteasome condensate experiments.

Instead, to address this reviewer concern, we included new experiments in vivo using Dsk2 Δ 3TH that were discussed above for reviewer #1 (Figure 6). In this construct, we preserve the STI1 domain (and presumably its chaperone activity) but eliminate the three transient helical regions (roughly ~35-40 AAs instead of 100 AAs) within the disordered regions of Dsk2. Our new data with Dsk2 Δ 3TH suggest that the amount of azide stress-induced proteasome puncta formation is reduced significantly compared to wild type without major changes in Dsk2 and polyUb levels (Figure 6A-E). These results complement our in vitro phase separation data and analyses. In the prolonged growth stress condition, we observed increased proteasome puncta formation for Dsk2 Δ 3TH cells relative to WT but we believe this is due to increased Dsk2 levels (2.5 -fold), elevated polyUb chain levels, and proteasome levels in prolonged growth stress (Figure 6G-L). We explain these results in detail in our results and discussion sections. In summary, we believe that inclusion of these new in vivo Δ 3TH data will help us eliminate the concern of what happens to the STI1 domain's functionality as the STI1 domain is preserved in the Δ 3TH construct.

Similar to reviewer 1 major concern #1 about Rad23, we have emphasized here that both Rad23 and Dsk2 play important roles in mediating proteasome condensate formation and that there is likely a redundancy between Dsk2 and Rad23 that confers their importance in mediating the formation of proteasome condensates as previously observed in (Waite et al, 2024). We mention that point in the opening paragraph of the Results subsection titled "Deletion of the Dsk2 STI1 domain or 3TH regions impacts proteasome condensate formation in vivo": "These deletions were introduced in *rad23 Δ* cells (RAD23 deletion strain) expressing GFP-tagged proteasomes (Rpn1-GFP); the *rad23 Δ* background was selected to eliminate redundancy from Rad23 in proteasome condensate formation (Waite et al, 2024)." As mentioned in our Reviewer 1 response, only the *rad23 Δ dsk2 Δ* double knockout strains are greatly inhibited in forming proteasome puncta (Figure 6B, 6H, Expanded View Figure EV2C, Expanded View Figure EV2J). We separated the results such that proteasome condensate formation results in *rad23 Δ* yeast strains are shown in main Figure 6, while results in the WT yeast strains (containing Rad23) are shown in Expanded View Figure EV2.

>>>Current comment: I appreciate the new Figure 6 and believe it addresses concerns about the STI1 domain functionality in Δ 3TH mutants.

3. The authors decision to threshold the intensity of the puncta in Fig 6A and B is not convincing. Particularly in Panel B the Dsk1 Δ STI1 construct the lower left cell looks like it has two puncta, but these puncta apparently don't meet the threshold cut-off. Perhaps if the authors set a threshold that is normalized to the intensity of each individual cell.

>>> We appreciate the reviewer's concern that presenting data using intensity thresholding may give the impression that some puncta were excluded from analysis. Our intention was to quantify the qualitative observation that puncta appeared brighter under certain conditions. However, we agree that quantifying the percentage of cells with one or more puncta provides a more straightforward representation of the data. We have removed the intensity thresholding data from the manuscript. Subpanels in Figure 6 and Expanded View Figure EV2 quantify the percentage of cells with one or more proteasome punctum/condensate.

>>>Current comment: I agree - this representation and analysis of the data is more convincing.

New Minor comment:

1. A new addition to the manuscript at the top of page 4:
"K48-linked or K63-linked polyUB-substrates"

While this is required to improve the manuscript, the authors need to describe the significance of these two substrates or why they were chosen.

Response to Reviewers

Re: EMBOJ-2025-121482R1

STI1 domain engages transient helices to mediate Dsk2 phase separation and proteasome condensation

We thank the editor and the reviewers for their supportive comments that continue to improve this manuscript. We have addressed the reviewer's concerns and editorial requests. We provided an updated version of our manuscript as a Word document with 'track changes' enabled.

We address all requests point-by-point in our responses below.

Referee #1:

I appreciate the additional experimentation and editing that the authors have done to address the points raised by myself and other reviewers; the authors have done a nice job improving the manuscript. I have no additional concerns.

>>> We thank the reviewer for their positive feedback and comments!

Referee #2:

1. >>> Current response: I can respect their desire to exclude this data from this study as it is a part of ongoing work, but I do think that the Dsk2-Chimera NMR spectra closely resembling the Dsk2-FL NMR spectra is an interesting observation. It indicates that their observations in Dsk2 could be applied to other STI1 domain containing proteins - namely UBQLNS. Basically, it broadens their study.

>>> We appreciate the reviewer's insights and fully agree with the reviewer. In response, we have supplied the Dsk2-Chimera NMR data in new Figure EV4. We added new text to the discussion section on pg. 14 where we comment on how this work broadens this study:

"To assess the broader applicability of STI1-transient helix interactions, we generated a chimeric construct (Dsk2-Chimera), in which the STI1 domain of Dsk2 was replaced with the STI1-II domain of human UBQLN2 (residues 379-462 of UBQLN2), and compared NMR spectra against Dsk2 FL and Dsk2 Δ STI1 (Figure EV4). Notably, the chemical shifts of 3TH residues in Dsk2-Chimera closely resemble those of Dsk2 FL rather than Dsk2 Δ STI1. This observation supports a general mechanism where the transient helices of UBQLNs engage with their STI1 domain, suggesting that such interactions may be conserved across STI1 domain-containing co-chaperone proteins."

2. >>>Current comment: I appreciate the new Figure 6 and believe it addresses concerns about the STI1 domain functionality in Δ 3TH mutants.

>>> Thank you!

3. >>>Current comment: I agree - this representation and analysis of the data is more convincing.

>>> Thank you!

New Minor comment:

1. A new addition to the manuscript at the top of page 4:

"K48-linked or K63-linked polyUB-substrates"

While this is required to improve the manuscript, the authors need to describe the significance of these two substrates or why they were chosen.

>>> We agree and added new text to the beginning of the Results section on pg. 5:

"Ub-binding shuttle proteins Dsk2 and Rad23 are required for proteasome condensate formation under certain stress conditions in yeast (Waite *et al*, 2024). To determine the role of Dsk2 in proteasome condensates, we reconstituted a minimal system *in vitro* using only Dsk2, either K48-linked or K63-linked polyUb-substrates, and yeast proteasome (Figure 1A). We chose K48-linked and K63-linked polyUb chains because they are the two most abundant polyUb chain types in cells and are both involved in maintaining proteostasis (Pickart & Fushman, 2004; Komander & Rape, 2012; Ohtake *et al*, 2016; Grumati & Dikic, 2018)."

Editorial Requests

- Please double-check to make sure to all relevant funding information in the manuscript is congruent with the info entered into our submission system. Currently missing in the submission system are: NSF grant MCB1935596; postdoctoral support to N.A. and J.K.N. from the Syracuse University Vice President of Research office, and a graduate fellowship from the Madison and Lila Self Graduate Programs at the University of Kansas; Alfred P. Sloan Foundation

>>> We have updated the funding information on the submission system as requested.

- Please reduce the number of keywords on the abstract page to five (ideally choosing specific and general terms).

>>> We have reduced the number of keywords to five.

- Please remove the EV Movie legends from the main text, they should only be present in the uploaded movie ZIP archives.

>>> We have supplied the legends inside the EV Movie ZIP archives.

- Please ensure that the deposited data are now becoming publicly accessible, and update the links in the Data Availability section accordingly (right now, some URLs may still be preliminary referee access links?).

>>> I have contacted the different data depositors. The Biostudies link will be updated with new EV Figure 4 contents after January 6, 2026. The SASBDB link has been updated. The BMRB link will be updated in January 2026 per an email from Hongyang Yao (BMRB). All other links should be available now.

- Please carefully go through the reference list and make sure that each reference is complete with citation year, volume, and page/locator numbers - such information is currently missing for some of them. Also, please adjust the format for citation of preprints as specified in our author guidelines:

The citation in the text should be: "(preprint: NAME1 et al, YEAR)"

The citation in the reference list: "NAME1, NAME2, ... (YEAR) (article title). bioRxiv doi: XXX" [preprint]

>>> We have double checked the reference list and updated citations in the main text as requested.

- Please double-checked that all Figures and Figure Panels are being referenced in the text in sequential order, e.g. Fig 1D before Fig 1E etc.

>>> We have gone through the figure call-outs and double-checked as requested.

- Finally, during routine pre-acceptance checks, our data editors have raised the following queries regarding figures, data, and legends, which I would ask you to address (ideally using the Track Changes option):

- Please note that the exact p values are not provided in the legends of figures 6B,H; EV2 C,

- Please note that information related to n is missing in the legend of figure 4D

- Please note that the error bars are not defined in the legends of figures EV1 D, 4D

- Please note that the measure of center for the error bars needs to be defined in the legends of figures 1D, 2D, 5A

>>> We have updated all requests as listed above.

Prof. Carlos A Castañeda
Syracuse University
Biology and Chemistry
Syracuse, NY 13244

12th Jan 2026

Re: EMBOJ-2025-121482R1
ST11 domain engages transient helices to mediate Dsk2 phase separation and proteasome condensation

Dear Carlos,

Thank you for submitting your final revised manuscript for our consideration. I am pleased to inform you that we have now accepted it for publication in The EMBO Journal.

You may qualify for financial assistance for your publication charges - either via a Springer Nature fully open access agreement or an EMBO initiative. Check your eligibility: <https://link.springer.com/journal/44318/how-to-publish-with-us>

With kind regards,

Hartmut

Please note that it is The EMBO Journal policy for the transcript of the editorial process (containing referee reports and your response letters) to be published as an online supplement to each paper. If you should prefer removal of any referee-only figures included in the point-by-point response(s), e.g. because they may still be used for future publication or because they have been reproduced from published work by others, please do let us know immediately via response email.

More information is available here: <https://link.springer.com/partners/embo-press/editorial-policies#Peer%20review>